# Collaborative Pure Exploration in Kernel Bandit

**Yihan Du**
Institute for Interdisciplinary Information Sciences
Tsinghua University
Beijing, China
`duyh18@mails.tsinghua.edu.cn`

**Wei Chen**
Microsoft Research
Beijing, China
`weic@microsoft.com`

**Yuko Kuroki**
The University of Tokyo & RIKEN AIP, Tokyo, Japan
CENTAI Institute, Turin, Italy
`yuko.kuroki@centai.eu`

**Longbo Huang** [*]
Institute for Interdisciplinary Information Sciences
Tsinghua University
Beijing, China
`longbohuang@tsinghua.edu.cn`

## Abstract

In this paper, we propose a novel Collaborative Pure Exploration in Kernel Bandit model (CoPE-KB), where multiple agents collaborate to complete different but related tasks with limited communication. Our model generalizes prior CoPE formulation with the single-task and classic MAB setting to allow multiple tasks and general reward structures. We propose a novel communication scheme with an efficient kernelized estimator, and design algorithms `CoKernelFC` and `CoKernelFB` for CoPE-KB with fixed-confidence and fixed-budget objectives, respectively. Sample and communication complexities are provided to demonstrate the efficiency of our algorithms. Our theoretical results explicitly quantify how task similarities influence learning speedup, and only depend on the effective dimension of feature space. Our novel techniques, such as an efficient kernelized estimator and decomposition of task similarities and arm features, which overcome the communication difficulty in high-dimensional feature space and reveal the impacts of task similarities on sample complexity, can be of independent interests.

## 1 Introduction

Pure exploration (Even-Dar et al., 2006; Kalyanakrishnan et al., 2012; Kaufmann et al., 2016) is a fundamental online learning problem in multi-armed bandits (Thompson, 1933; Lai & Robbins, 1985; Auer et al., 2002), where an agent chooses options (often called arms) and observes random feedback with the objective of identifying the best arm. This formulation has found many important applications, such as web content optimization (Agarwal et al., 2009) and online advertising (Tang et al., 2013). However, traditional pure exploration (Even-Dar et al., 2006; Kalyanakrishnan et al., 2012; Kaufmann et al., 2016) only considers single-agent decision making, and cannot be applied to prevailing distributed systems in real world, which often face a heavy computation load and require multiple parallel devices to process tasks, e.g., distributed web servers (Zhuo et al., 2003) and data centers (Liu et al., 2011).

To handle such distributed applications, prior works (Hillel et al., 2013; Tao et al., 2019; Karpov et al., 2020) have developed the Collaborative Pure Exploration (CoPE) model, where multiple agents communicate and cooperate to identify the best arm with learning speedup. Yet, existing results focus only on the classic multi-armed bandit (MAB) setting with *single* task, i.e., all agents solve a common task and the rewards of arms are *individual* values (rather than generated by a reward function). However, in many distributed applications such as multi-task neural architecture search (Gao et al., 2020), different devices can face *different but related* tasks, and there exists similar dependency of rewards on option features among different tasks. Therefore, it is important to

---

[*]Corresponding author.

develop a more general CoPE model to allow heterogeneous tasks and structured reward dependency, and further theoretically understand how task correlation impacts learning.

Motivated by the above fact, we propose a novel Collaborative Pure Exploration in Kernel Bandit (CoPE-KB) model. Specifically, in CoPE-KB, each agent is given a set of arms, and the expected reward of each arm is generated by a task-dependent reward function in a high and possibly infinite dimensional Reproducing Kernel Hilbert Space (RKHS) (Wahba, 1990; Schölkopf et al., 2002). Each agent sequentially chooses arms to sample and observes random outcomes in order to identify the best arm. Agents can broadcast and receive messages to and from others in communication rounds, so that they can collaborate and exploit the task similarity to expedite learning processes.

Our CoPE-KB model is a novel generalization of prior CoPE problem (Hillel et al., 2013; Tao et al., 2019; Karpov et al., 2020), which not only extends prior models from the single-task setting to multiple tasks, but also goes beyond classic MAB setting and allows general (linear or nonlinear) reward structures. CoPE-KB is most suitable for applications involving multiple tasks and complicated reward structures. For example, in multi-task neural architecture search (Gao et al., 2020), one wants to search for best architectures for different but related tasks on multiple devices, e.g., the object detection (Ghiasi et al., 2019) and object tracking (Yan et al., 2021) tasks in computer vision, which often use similar neural architectures. Instead of individually evaluating each possible architecture, one prefers to directly learn the relationship (reward function) between the accuracy results achieved and the features of used architectures (e.g., the type of neural networks), and exploit the similarity of reward functions among tasks to accelerate the search.

Our CoPE-KB generalization faces a unique challenge on *communication*. Specifically, in prior CoPE works with classic MAB setting (Hillel et al., 2013; Tao et al., 2019; Karpov et al., 2020), agents only need to learn *scalar* rewards, which are easy to transmit. However, under the kernel model, agents need to estimate a *high or even infinite dimensional* reward parameter, which is inefficient to directly transmit. Also, if one naively adapts existing reward estimators for kernel bandits (Srinivas et al., 2010; Camilleri et al., 2021) to learn this high-dimensional reward parameter, he/she will suffer an expensive communication cost dependent on the number of samples $N^{(r)}$, since the reward estimators there need *all raw sample outcomes* to be transmitted. To tackle this challenge, we develop an efficient kernelized estimator, which only needs *average outcomes* on $nV$ arms and reduces the required transmitted messages from $O(N^{(r)})$ to $O(nV)$. Here $V$ is the number of agents, and $n$ is the number of arms for each agent. The number of samples $N^{(r)}$ depends on the inverse of the minimum reward gap, and is often far larger than the number of arms $nV$.

Under the CoPE-KB model, we study two popular objectives, i.e., Fixed-Confidence (FC), where we aim to minimize the number of samples used under a given confidence, and Fixed-Budget (FB), where the goal is to minimize the error probability under a given sample budget. We design two algorithms `CoKernelFC` and `CoKernelFB`, which adopt an efficient kernelized estimator to simplify the required data transmission and enjoy a $O(nV)$ communication cost, instead of $O(N^{(r)})$ as in adaptions of existing kernel bandit algorithms (Srinivas et al., 2010; Camilleri et al., 2021). We provide sampling and communication guarantees, and also interpret them by standard kernel measures, e.g., maximum information gain and effective dimension. Our results rigorously quantify the influences of task similarities on learning acceleration, and hold for both finite and infinite dimensional feature space.

The contributions of this paper are summarized as follows:

- We formulate a collaborative pure exploration in kernel bandit (CoPE-KB) model, which generalizes prior single-task CoPE formulation to allow multiple tasks and general reward structures, and consider two objectives, i.e., fixed-confidence (FC) and fixed-budget (FB).

- For the FC objective, we propose algorithm `CoKernelFC`, which adopts an efficient kernelized estimator to simplify the required data transmission and enjoys only a $O(nV)$ communication cost. We derive sample complexity $\tilde{O}(\frac{\rho^*(\xi)}{V} \log \delta^{-1})$ and communication rounds $O(\log \Delta_{\min}^{-1})$. Here $\xi$ is the regularization parameter, and $\rho^*(\xi)$ is the problem hardness (see Section 4.3).

- For the FB objective, we design a novel algorithm `CoKernelFB` with error probability $\tilde{O}(\exp(-\frac{TV}{\rho^*(\xi)})n^2V)$ and communication rounds $O(\log(\omega(\xi, \tilde{\mathcal{X}})))$. Here $T$ is the sample

budget, $\tilde{\mathcal{X}}$ is the set of task-arm feature pairs, and $\omega(\xi, \tilde{\mathcal{X}})$ is the principle dimension of data projections in $\tilde{\mathcal{X}}$ to RKHS (see Section 5.1).

- Our algorithms offer an efficient communication scheme for information exchange in high dimensional feature space. Our results explicitly quantify how task similarities impact learning acceleration, and only depend on the effective dimension of feature space.

Due to the space limit, we defer all proofs to the appendix.

## 2   RELATED WORK

Below we review the most related works, and defer a complete literature review to Appendix B.

**Collaborative Pure Exploration (CoPE).** Hillel et al. (2013); Tao et al. (2019) initiate the CoPE literature with the single-task and classic MAB setting, where all agents solve a common classic best arm identification problem (without reward structures). Karpov et al. (2020) further extend the formulation in (Hillel et al., 2013; Tao et al., 2019) to best $m$ arm identification. Our CoPE-KB generalizes prior CoPE works to allow multiple tasks and general reward structures, and faces a unique challenge on communication due to the high-dimensional feature space. Recently, Wang et al. (2019) study distributed multi-armed and linear bandit problems, and He et al. (2022) investigate federated linear bandits with asynchronous communication. Their algorithms directly communicate the estimated reward parameters and cannot be applied to solve our problem, since under kernel representation, the reward parameter is high-dimensional and expensive to explicitly transmit.

**Kernel Bandits.** Srinivas et al. (2010); Valko et al. (2013); Scarlett et al. (2017); Li & Scarlett (2022) investigate the single-agent kernel bandit problem and establish regret bounds dependent on maximum information gain. Krause & Ong (2011); Deshmukh et al. (2017) study multi-task kernel bandits based on composite kernel functions. Dubey et al. (2020); Li et al. (2022) consider multi-agent kernel bandits with local-broadcast and client-server communication protocols, respectively. Camilleri et al. (2021); Zhu et al. (2021) investigate single-agent pure exploration in kernel space. The above kernel bandit works consider either regret minimization or single-agent formulation, which cannot be applied to resolve our challenges on round-speedup analysis and communication.

## 3   COLLABORATIVE PURE EXPLORATION IN KERNEL BANDIT (COPE-KB)

In this section, we define the Collaborative Pure Exploration in Kernel Bandit (CoPE-KB) problem.

**Agents and Rewards.** There are $V$ agents indexed by $[V] := \{1, \ldots, V\}$, who collaborate to solve different but possibly related instances (tasks) of the Pure Exploration in Kernel Bandit (PE-KB) problem. For each agent $v \in [V]$, she is given a set of $n$ arms $\mathcal{X}_v = \{x_{v,1}, \ldots, x_{v,n}\} \subseteq \mathbb{R}^{d_{\mathcal{X}}}$, where $x_{v,i}$ ($i \in [n]$) describes the arm feature, and $d_{\mathcal{X}}$ is the dimension of arm feature vectors. The expected reward of each arm $x \in \mathcal{X}_v$ is $f_v(x)$, where $f_v : \mathcal{X}_v \mapsto \mathbb{R}$ is an unknown reward function. Let $\mathcal{X} := \cup_{v \in [V]} \mathcal{X}_v$. At each timestep $t$, each agent $v$ pulls an arm $x_v^t \in \mathcal{X}_v$ and observes a random reward $y_v^t = f_v(x_v^t) + \eta_v^t$. Here $\eta_v^t$ is a zero-mean and 1-sub-Gaussian noise, and it is independent among different $t$ and $v$. We assume that the best arms $x_{v,*} := \operatorname{argmax}_{x \in \mathcal{X}_v} f_v(x)$ are unique for all $v \in [V]$, which is a common assumption in the pure exploration literature (Even-Dar et al., 2006; Audibert et al., 2010; Kaufmann et al., 2016).

**Multi-Task Kernel Composition.** We assume that the functions $f_v$ are parametric functionals of a global function $F : \mathcal{Z} \times \mathcal{X} \mapsto \mathbb{R}$, which satisfies that, for each agent $v \in [V]$, there exists a task feature vector $z_v \in \mathcal{Z}$ such that

$$f_v(x) = F(z_v, x), \ \forall x \in \mathcal{X}_v. \tag{1}$$

Here $\mathcal{Z}$ and $\mathcal{X}$ denote the task feature space and arm feature space, respectively. Note that Eq. (1) allows tasks to be different for agents (by having different task features), whereas prior CoPE works (Hillel et al., 2013; Tao et al., 2019; Karpov et al., 2020) restrict the tasks ($\mathcal{X}_v$ and $f_v$) to be the same for all agents $v \in [V]$.

We denote a task-arm feature pair (i.e., overall input of function $F$) by $\tilde{x} := (z_v, x)$, and denote the space of task-arm feature pairs (i.e., overall input space of function $F$) by $\tilde{\mathcal{X}} := \mathcal{Z} \times \mathcal{X}$. As a standard assumption in kernel bandits (Krause & Ong, 2011; Deshmukh et al., 2017; Dubey et al.,

2020), we assume that $F$ has a bounded norm in a high (possibly infinite) dimensional Reproducing Kernel Hilbert Space (RKHS) $\mathcal{H}$ specified by the kernel $k : \tilde{\mathcal{X}} \times \tilde{\mathcal{X}} \mapsto \mathbb{R}$, and there exists a feature mapping $\phi : \tilde{\mathcal{X}} \mapsto \mathcal{H}$ and an unknown parameter $\theta^* \in \mathcal{H}$ such that[1]

$$F(\tilde{x}) = \phi(\tilde{x})^\top \theta^*, \ \forall \tilde{x} \in \tilde{\mathcal{X}},$$

$$k(\tilde{x}, \tilde{x}') = \phi(\tilde{x})^\top \phi(\tilde{x}'), \ \forall \tilde{x}, \tilde{x}' \in \tilde{\mathcal{X}}.$$

Here $k(\cdot, \cdot)$ is a product composite kernel, which satisfies that for any $z, z' \in \mathcal{Z}$ and $x, x' \in \mathcal{X}$,

$$k((z, x), (z', x')) = k_\mathcal{Z}(z, z') \cdot k_\mathcal{X}(x, x'),$$

where $k_\mathcal{Z} : \mathcal{Z} \times \mathcal{Z} \mapsto \mathbb{R}$ is the task feature kernel which measures the similarity of functions $f_v$, and $k_\mathcal{X} : \mathcal{X} \times \mathcal{X} \mapsto \mathbb{R}$ is the arm feature kernel which depicts the feature structure of arms. The computation of kernel function $k(\cdot, \cdot)$ operates only on low-dimensional input data. By expressing all calculations via $k(\cdot, \cdot)$, it enables us to avoid maintaining high-dimensional $\phi(\cdot)$ and attain computation efficiency. In addition, the composite structure of $k(\cdot, \cdot)$ allows us to model the similarity among tasks.

Let $K_\mathcal{Z} := [k_\mathcal{Z}(z_v, z_{v'})]_{v, v' \in [V]}$ denote the kernel matrix of task features. $\texttt{rank}(K_\mathcal{Z})$ characterizes how much the tasks among agents are similar. For example, if agents solve a common task, i.e., the arm set $\mathcal{X}_v$ and reward function $f_v$ are the same for all agents, we have that the task feature $z_v$ is the same for all $v \in [V]$, and $\texttt{rank}(K_\mathcal{Z}) = 1$; If all tasks are totally different, $\texttt{rank}(K_\mathcal{Z}) = V$.

To better illustrate the model, we provide a 2-agent example in Figure 1. There are Items $1, 2, 3$ with expected rewards $3 + \sqrt{2}, 4 + 2\sqrt{2}, 5 + 3\sqrt{2}$,

$$x_{1,2} = x_{2,1} = [1, 0, 1]^\top$$
$$\phi(\tilde{x}_{1,2}) = \phi(\tilde{x}_{2,1}) = [1, 0, 1, 0, \sqrt{2}, 0]^\top$$
$$F(\tilde{x}_{1,2}) = F(\tilde{x}_{2,1}) = 4 + 2\sqrt{2}$$
Item 2, with reward $4 + 2\sqrt{2}$

Item 1, with reward $3 + \sqrt{2}$     Item 3, with reward $5 + 3\sqrt{2}$
$$x_{1,1} = [1, 1, 0]^\top \qquad\qquad x_{2,2} = [0, 1, 1]^\top$$
$$\phi(\tilde{x}_{1,1}) = [1, 1, 0, \sqrt{2}, 0, 0]^\top \quad \phi(\tilde{x}_{2,2}) = [0, 1, 1, 0, 0, \sqrt{2}]^\top$$
$$F(\tilde{x}_{1,1}) = 3 + \sqrt{2} \qquad\qquad F(\tilde{x}_{2,2}) = 5 + 3\sqrt{2}$$
Task 1   Task 2
Task feature: $z_1 = z_2 = 1$

$\forall z \in \mathcal{Z}, \ \forall x = [x_{(1)}, x_{(2)}, x_{(3)}]^\top \in \mathcal{X},$
Reward function:
$$F(\tilde{x}) = F(z, x) = z \cdot (x_{(1)}^2 + 2x_{(2)}^2 + 3x_{(3)}^2 + \sqrt{2}x_{(1)}x_{(2)} + 2\sqrt{2}x_{(1)}x_{(3)} + 3\sqrt{2}x_{(2)}x_{(3)})$$
Feature mapping:
$$\phi(\tilde{x}) = \phi(z, x) = z \cdot [x_{(1)}^2, x_{(2)}^2, x_{(3)}^2, \sqrt{2}x_{(1)}x_{(2)}, \sqrt{2}x_{(1)}x_{(3)}, \sqrt{2}x_{(2)}x_{(3)}]^\top$$
Reward parameter:
$$\theta^* = [1, 2, 3, 1, 2, 3]^\top$$
Kernel function:
$$k(\tilde{x}, \tilde{x}') = k((z, x), (z', x')) = \phi(z, x)^\top \phi(z', x') = zz' \cdot (x_{(1)}x'_{(1)} + x_{(2)}x'_{(2)} + x_{(3)}x'_{(3)})^2$$

Figure 1: Illustrating example for CoPE-KB.

respectively. Agent 1 is given Items $1, 2$, denoted by $\mathcal{X}_1 = \{x_{1,1}, x_{1,2}\}$, and Agent 2 is given Items $2, 3$, denoted by $\mathcal{X}_2 = \{x_{2,1}, x_{2,2}\}$, where both $x_{1,2}$ and $x_{2,1}$ refer to Item 2. The task feature $z_1 = z_2 = 1$, which means that the expected rewards of all items are generated by a common function. The reward function $F$ is nonlinear with respect to $x$, but can be represented in a linear form in a higher-dimensional feature space, i.e., $F(\tilde{x}) = F(z, x) = \phi(\tilde{x})^\top \theta^*$ with $\tilde{x} := (z, x)$ for any $z \in \mathcal{Z}$ and $x \in \mathcal{X}$. Here $\phi(\tilde{x})$ is the feature embedding, and $\theta^*$ is the reward parameter. The computation of kernel function $k(\tilde{x}, \tilde{x}')$ only involves low-dimensional input data $\tilde{x}$ and $\tilde{x}'$, rather than higher-dimensional feature embedding $\phi(\cdot)$. $F, \phi, \theta^*$ and $k$ are specified in Figure 1. The two agents can share the learned information on $\theta^*$ to accelerate learning.

**Communication.** Following the popular communication protocol in the CoPE literature (Hillel et al., 2013; Tao et al., 2019; Karpov et al., 2020), we allow these $V$ agents to exchange information via *communication rounds*, in which each agent can broadcast and receive messages from others. Following existing CoPE works (Hillel et al., 2013; Tao et al., 2019; Karpov et al., 2020), we restrict the length of each message within $O(n)$ bits for practicability, where $n$ is the number of arms for each agent, and we consider the number of bits for representing a real number as a constant. We want agents to cooperate and complete all tasks using as few communication rounds as possible.

**Objectives.** We consider two objectives of the CoPE-KB problem, one with Fixed-Confidence (FC) and the other with Fixed-Budget (FB). In the FC setting, given a confidence parameter $\delta \in (0, 1)$, the agents aim to identify $x_{v,*}$ for all $v \in [V]$ with probability at least $1 - \delta$, and minimize the average number of samples used per agent. In the FB setting, the agents are given an overall $TV$ sample budget ($T$ average samples per agent), and aim to use at most $TV$ samples to identify $x_{v,*}$ for all $v \in [V]$ and minimize the error probability. In both FC and FB settings, the agents are requested to minimize the number of communication rounds and control the length of each message within $O(n)$ bits, as in the CoPE literature (Hillel et al., 2013; Tao et al., 2019; Karpov et al., 2020).

---

[1]For any $h, h' \in \mathcal{H}$, we denote their inner-product by $h^\top h' := \langle h, h' \rangle_\mathcal{H}$ and denote the norm of $h$ by $\|h\| := \sqrt{h^\top h}$. For any $h \in \mathcal{H}$ and matrix $M$, we denote $\|h\|_{M^{-1}} := \sqrt{h^\top M^{-1} h}$.

---

**Algorithm 1** Collaborative Multi-agent Algorithm `CoKernelFC`: for Agent $v \in [V]$

---

1: **Input:** $\delta$, $\tilde{\mathcal{X}}$, $k(\cdot, \cdot) : \tilde{\mathcal{X}} \times \tilde{\mathcal{X}} \mapsto \mathbb{R}$, regularization parameter $\xi$, rounding procedure `ROUND`, rounding approximation parameter $\varepsilon = \frac{1}{10}$

2: **Initialization:** $\delta_r := \frac{\delta}{2r^2}$ for all $r \geq 1$. $\mathcal{B}_{v'}^{(1)} \leftarrow \tilde{\mathcal{X}}_{v'}$ for all $v' \in [V]$.

3: **for** round $r = 1, 2, \ldots$ **do**

4:   Let $\lambda_r^*$ and $\rho_r^*$ be the optimal solution and optimal value of
$$\min_{\lambda \in \triangle_{\tilde{\mathcal{X}}}} \max_{\tilde{x}_i, \tilde{x}_j \in \mathcal{B}_{v'}^{(r)}, v' \in [V]} \|\phi(\tilde{x}_i) - \phi(\tilde{x}_j)\|_{(\xi I + \sum_{\tilde{x} \in \tilde{\mathcal{X}}} \lambda_{\tilde{x}} \phi(\tilde{x}) \phi(\tilde{x})^\top)^{-1}}^2 \qquad \text{// compute the optimal sample allocation}$$

5:   $N^{(r)} \leftarrow \max\{\lceil 32(2^r)^2(1 + \varepsilon)^2 \rho_r^* \log(2n^2 V / \delta_r)\rceil, \ \tau(\xi, \lambda_r^*, \varepsilon)\}$, where $\tau(\xi, \lambda_r^*, \varepsilon)$ is the number of samples needed by `ROUND`

6:   $(\tilde{s}_1, \ldots, \tilde{s}_{N^{(r)}}) \leftarrow \text{ROUND}(\xi, \lambda_r^*, N^{(r)}, \varepsilon)$

7:   Extract a sub-sequence $\tilde{s}_v^{(r)}$ from $(\tilde{s}_1, \ldots, \tilde{s}_{N^{(r)}})$ which only contains the arms in $\tilde{\mathcal{X}}_v$

8:   Sample the arms in $\tilde{s}_v^{(r)}$ and observe random rewards $\boldsymbol{y}_v^{(r)}$

9:   Let $N_{v,i}^{(r)}$ and $\bar{y}_{v,i}^{(r)}$ be the number of samples and the average sample outcome on arm $\tilde{x}_{v,i}$

10:  Broadcast $\{(N_{v,i}^{(r)}, \bar{y}_{v,i}^{(r)})\}_{i \in [n]}$, and receive $\{(N_{v',i}^{(r)}, \bar{y}_{v',i}^{(r)})\}_{i \in [n]}$ from all other agents $v' \neq v$

11:  $k_r(\tilde{x}) \leftarrow [\sqrt{N_1^{(r)}} k(\tilde{x}, \tilde{x}_1), \ldots, \sqrt{N_{nV}^{(r)}} k(\tilde{x}, \tilde{x}_{nV})]^\top$ for any $\tilde{x} \in \tilde{\mathcal{X}}$. $K^{(r)} \leftarrow [\sqrt{N_i^{(r)} N_j^{(r)}} k(\tilde{x}_i, \tilde{x}_j)]_{i,j \in [nV]}$. $\bar{y}^{(r)} \leftarrow [\sqrt{N_1^{(r)}} \bar{y}_1^{(r)}, \ldots, \sqrt{N_{nV}^{(r)}} \bar{y}_{nV}^{(r)}]^\top$ // organize overall sample information

12:  **for** all $v' \in [V]$ **do**

13:    $\hat{\Delta}_r(\tilde{x}, \tilde{x}') \leftarrow (k_r(\tilde{x}) - k_r(\tilde{x}'))^\top (K^{(r)} + N^{(r)} \xi I)^{-1} \bar{y}^{(r)}, \ \forall \tilde{x}, \tilde{x}' \in \mathcal{B}_{v'}^{(r)}$ // use a kernelized estimator (described in Section 4.2) to estimate reward gaps

14:    $\mathcal{B}_{v'}^{(r+1)} \leftarrow \mathcal{B}_{v'}^{(r)} \setminus \{\tilde{x} \in \mathcal{B}_{v'}^{(r)} | \exists \tilde{x}' \in \mathcal{B}_{v'}^{(r)} : \hat{\Delta}_r(\tilde{x}', \tilde{x}) \geq 2^{-r}\}$ // discard sub-optimal arms

15:  **end for**

16:  **if** $\forall v' \in [V], |\mathcal{B}_{v'}^{(r+1)}| = 1$, **return** $\mathcal{B}_1^{(r+1)}, \ldots, \mathcal{B}_V^{(r+1)}$

17: **end for**

---

Different from prior CoPE-KB works (Tao et al., 2019; Karpov et al., 2020) which consider minimizing the maximum number of samples used by individual agents, we aim to minimize the average (total) number of samples used. Our objective is motivated by the fact that in many applications, obtaining a sample is expansive, e.g., clinical trials (Weninger et al., 2019), and thus it is important to minimize the average (total) number of samples required. For example, consider that a medical institution wants to conduct multiple clinical trials to identify the best treatments for different age groups of patients (different tasks), and share the obtained data to accelerate the development. Since conducting a trial can consume significant medical resources and funds (e.g., organ transplant surgeries and convalescent plasma treatments for COVID-19), the institution wants to minimize the total number of trials required. Our CoPE-KB model is most suitable for such scenarios.

To sum up, in CoPE-KB, we let agents collaborate to simultaneously complete multiple related best arm identification tasks using few communication rounds. In particular, when all agents solve the same task and $\mathcal{X}$ is the canonical basis, our CoPE-KB reduces to existing CoPE with classic MAB setting (Hillel et al., 2013; Tao et al., 2019).

## 4 Fixed-Confidence CoPE-KB

We start with the fixed-confidence setting. We present algorithm `CoKernelFC` equipped with an efficient kernelized estimator, and provide theoretical guarantees in sampling and communication.

### 4.1 Algorithm `CoKernelFC`

`CoKernelFC` (Algorithm 1) is an elimination-based multi-agent algorithm. The procedure for each agent $v$ is as follows. Agent $v$ maintains candidate arm sets $\mathcal{B}_{v'}^{(r)}$ for all $v' \in [V]$. In each round $r$, agent $v$ solves a global min-max optimization (Line 4) to find the optimal sample allocation $\lambda_r^* \in \triangle_{\tilde{\mathcal{X}}}$, which achieves the minimum estimation error. Here $\triangle_{\tilde{\mathcal{X}}}$ denotes the collection of all dis-

tributions on $\tilde{\mathcal{X}}$, and $\rho_r^*$ is the factor of optimal estimation error. In practice, the high-dimensional feature embedding $\phi(\tilde{x})$ is only implicitly maintained, and this optimization can be efficiently solved by kernelized gradient descent (Camilleri et al., 2021) (see Appendix C.2). After solving this optimization, agent $v$ uses $\rho_r^*$ to compute the number of samples $N^{(r)}$ to ensure the estimation error of reward gaps to be smaller than $2^{-(r+1)}$ (Line 5).

Next, we call a rounding procedure $\texttt{ROUND}(\xi, \lambda_r^*, N^{(r)}, \varepsilon)$ (Allen-Zhu et al., 2021; Camilleri et al., 2021), which rounds a weighted sample allocation $\lambda_r^* \in \triangle_{\tilde{\mathcal{X}}}$ into a discrete sample sequence $(\tilde{s}_1, \ldots, \tilde{s}_{N^{(r)}}) \in \tilde{\mathcal{X}}^{N^{(r)}}$, and ensures the rounding error within $\varepsilon$ (Line 6). This rounding procedure requires the number of samples $N^{(r)} \geq \tau(\xi, \lambda_r^*, \varepsilon) = O(\frac{\tilde{d}(\xi, \lambda_r^*)}{\varepsilon^2})$ (Line 5). Here $\tau(\xi, \lambda_r^*, \varepsilon)$ is the number of samples needed by $\texttt{ROUND}$. $\tilde{d}(\xi, \lambda_r^*)$ is the number of the eigenvalues of matrix $\sum_{\tilde{x} \in \tilde{\mathcal{X}}} \lambda_r^*(\tilde{x}) \phi(\tilde{x}) \phi(\tilde{x})^\top$ which are greater than $\xi$. It stands for the effective dimension of feature space (see Appendix C.1 for details of this rounding procedure).

Obtaining sample sequence $(\tilde{s}_1, \ldots, \tilde{s}_{N^{(r)}})$, agent $v$ extracts a sub-sequence $\tilde{\boldsymbol{s}}_v^{(r)}$ which only contains the arms in her arm set $\tilde{\mathcal{X}}_v$. Then, she sample the arms in $\tilde{\boldsymbol{s}}_v^{(r)}$ and observe sample outcomes $\boldsymbol{y}_v^{(r)}$. After sampling, she only communicates the number of samples $N_{v,i}^{(r)}$ and *average* sample outcome $\bar{y}_{v,i}^{(r)}$ on each arm $\tilde{x}_{v,i}$ with other agents (Line 10). Receiving overall sample information, she uses a kernelized estimator (discussed shortly) to estimate the reward gap $\hat{\Delta}_r(\tilde{x}, \tilde{x}')$ between any two arms $\tilde{x}, \tilde{x}' \in \mathcal{B}_{v'}^{(r)}$ for all $v' \in [V]$, and discards sub-optimal arms (Lines 13,14). In Line 11, for any $i \in [nV]$, we use $\tilde{x}_i, \bar{y}_i^{(r)}$ and $N_i^{(r)}$ with a single subscript to denote the $i$-th arm in $\tilde{\mathcal{X}}$, the average sample outcome on this arm and the number of samples allocated to this arm, respectively.

## 4.2 KERNELIZED ESTIMATOR, COMMUNICATION AND COMPUTATION

Now we introduce the kernelized estimator (Line 13), which boosts communication and computation efficiency of $\texttt{CoKernelFC}$. First note that, our CoPE-KB generalization faces a unique challenge on communication, i.e., how to let agents efficiently share their learned information on the high-dimensional reward parameter $\theta^*$. Naively adapting existing federated linear bandit or kernel bandit algorithms (Dubey & Pentland, 2020; Huang et al., 2021; Dubey et al., 2020), which transmit the *whole* estimated reward parameter or *all raw sample outcomes*, will suffer a $O(\dim(\mathcal{H}))$ or $O(N^{(r)})$ communication cost, respectively. Here, the number of samples $N^{(r)} = \tilde{O}(d_{\text{eff}}/\Delta_{\min}^2)$, where $d_{\text{eff}}$ is the effective dimension of feature space and $\Delta_{\min}$ is the minimum reward gap. Thus, $N^{(r)}$ is far larger than the number of arms $nV$ when $\Delta_{\min}$ is small.

**Kernelized Estimator.** To handle the communication challenge, we develop a novel kernelized estimator (Eq. (3)) to significantly simplify the required transmitted data. Specifically, we make a key observation: since the sample sequence $(\tilde{s}_1, \ldots, \tilde{s}_{N^{(r)}})$ are constituted by arms $\tilde{x}_1, \ldots, \tilde{x}_{nV}$, one can *merge repetitive computations* for same arms. Then, for all $i \in [nV]$, we merge the repetitive feature embeddings $\phi(\tilde{x}_i)$ for same arms $\tilde{x}_i$, and condense the $N^{(r)}$ raw sample outcomes to *average outcome* $\bar{y}_i^{(r)}$ on each arm $\tilde{x}_i$. As a result, we express $\hat{\theta}_r$ in a simplified kernelized form, and use it to estimate the reward gap $\hat{\Delta}_r(\tilde{x}, \tilde{x}')$ between any two arms $\tilde{x}, \tilde{x}'$ as

$$\hat{\theta}_r := \Phi_r^\top \left(N^{(r)} \xi I + K^{(r)}\right)^{-1} \bar{y}^{(r)}, \tag{2}$$

$$\hat{\Delta}_r(\tilde{x}, \tilde{x}') := (\phi(\tilde{x}) - \phi(\tilde{x}'))^\top \hat{\theta}_r = (k_r(\tilde{x}) - k_r(\tilde{x}'))^\top \left(N^{(r)} \xi I + K^{(r)}\right)^{-1} \bar{y}^{(r)}. \tag{3}$$

Here $\Phi_r := [\sqrt{N_1^{(r)}} \phi(\tilde{x}_1)^\top; \ldots; \sqrt{N_{nV}^{(r)}} \phi(\tilde{x}_{nV})^\top]$, $K^{(r)} := [\sqrt{N_i^{(r)} N_j^{(r)}} k(\tilde{x}_i, \tilde{x}_j)]_{i,j \in [nV]}$, $k_r(\tilde{x}) := \Phi_r \phi(\tilde{x}) = [\sqrt{N_1^{(r)}} k(\tilde{x}, \tilde{x}_1), \ldots, \sqrt{N_{nV}^{(r)}} k(\tilde{x}, \tilde{x}_{nV})]^\top$ and $\bar{y}^{(r)} := [\sqrt{N_1^{(r)}} \bar{y}_1^{(r)}, \ldots, \sqrt{N_{nV}^{(r)}} \bar{y}_{nV}^{(r)}]^\top$ stand for the feature embeddings, kernel matrix, correlations and average outcomes of the $nV$ arms, respectively, which *merge* repetitive information on same arms. We refer interested reader to Appendix C.2, C.3 for a detailed derivation of our kernelized estimator and a comparison with existing estimators (Dubey et al., 2020; Camilleri et al., 2021).

**Communication.** Thanks to the kernelized estimator, we only need to transmit the $nV$ average outcomes $\bar{y}_1^{(r)}, \ldots, \bar{y}_{nV}^{(r)}$ (Line 10), instead of the whole $\hat{\theta}_r$ or all $N^{(r)}$ raw outcomes as in existing federated linear bandit or kernel bandit algorithms (Dubey & Pentland, 2020; Dubey et al., 2020). This significantly reduces the number of transmission bits from $O(\dim(\mathcal{H}))$ or $O(N^{(r)})$ to only $O(nV)$, and satisfies the $O(n)$-bit per message requirement.

**Computation.** In `CoKernelFC`, $\phi(\tilde{x})$ and $\hat{\theta}_r$ are maintained implicitly, and all steps (e.g., Lines 4, 13) can be implemented efficiently by only querying kernel function $k(\cdot, \cdot)$ (see Appendix C.2 for implementation details). Thus, the computation complexity for reward estimation is only $\text{Poly}(nV)$, instead of $\text{Poly}(\dim(\mathcal{H}))$ as in prior kernel bandit algorithms (Zhou et al., 2020; Zhu et al., 2021).

## 4.3 THEORETICAL PERFORMANCE OF `CoKernelFC`

To formally state our results, we define the speedup and hardness as in the literature (Hillel et al., 2013; Tao et al., 2019; Fiez et al., 2019).

For a CoPE-KB instance $\mathcal{I}$, let $T_{\mathcal{A}_M, \mathcal{I}}$ denote the average number of samples used per agent in a multi-agent algorithm $\mathcal{A}_M$ to identify all best arms. Let $T_{\mathcal{A}_S, \mathcal{I}}$ denote the average number of samples used per agent, by replicating $V$ copies of a single-agent algorithm $\mathcal{A}_S$ to complete all tasks (without communication). Then, the speedup of $\mathcal{A}_M$ on instance $\mathcal{I}$ is defined as

$$\beta_{\mathcal{A}_M, \mathcal{I}} = \inf_{\mathcal{A}_S} \frac{T_{\mathcal{A}_S, \mathcal{I}}}{T_{\mathcal{A}_M, \mathcal{I}}}. \tag{4}$$

It holds that $1 \leq \beta_{\mathcal{A}_M, \mathcal{I}} \leq V$. When all tasks are the same, $\beta_{\mathcal{A}_M, \mathcal{I}}$ can approach $V$. When all tasks are totally different, communication brings no benefit and $\beta_{\mathcal{A}_M, \mathcal{I}} = 1$. This speedup can be similarly defined for error probability results (see Section 5.2), by taking $T_{\mathcal{A}_M, \mathcal{I}}$ and $T_{\mathcal{A}_S, \mathcal{I}}$ as the smallest numbers of samples needed to meet the confidence constraint.

The hardness for CoPE-KB is defined as

$$\rho^*(\xi) = \min_{\lambda \in \triangle_{\tilde{\mathcal{X}}}} \max_{\tilde{x} \in \tilde{\mathcal{X}}_v \setminus \{\tilde{x}_{v,*}\}, v \in [V]} \frac{\|\phi(\tilde{x}_{v,*}) - \phi(\tilde{x})\|_{A(\xi, \lambda)^{-1}}^2}{(F(\tilde{x}_{v,*}) - F(\tilde{x}))^2}, \tag{5}$$

where $\xi \geq 0$ is a regularization parameter, and $A(\xi, \lambda) := \xi I + \sum_{\tilde{x} \in \tilde{\mathcal{X}}} \lambda_{\tilde{x}} \phi(\tilde{x}) \phi(\tilde{x})^\top$. This definition of $\rho^*(\xi)$ is adapted from prior linear bandit work (Fiez et al., 2019). Here $F(\tilde{x}_{v,*}) - F(\tilde{x})$ is the reward gap between the best arm $\tilde{x}_{v,*}$ and a suboptimal arm $\tilde{x}$, and $\|\phi(\tilde{x}_{v,*}) - \phi(\tilde{x})\|_{A(\xi, \lambda)^{-1}}^2$ is a dimension-related factor of estimation error. Intuitively, $\rho^*(\xi)$ indicates how many samples it takes to make the estimation error smaller than the reward gap under regularization parameter $\xi$.

Let $\Delta_{\min} := \min_{\tilde{x} \in \tilde{\mathcal{X}}_v \setminus \{\tilde{x}_{v,*}\}, v \in [V]} (F(\tilde{x}_{v,*}) - F(\tilde{x}))$ denote the minimum reward gap between the best arm and suboptimal arms among all tasks. Let $S$ denote the average number of samples used by each agent, i.e., per-agent sample complexity. Below we present the performance of `CoKernelFC`.

**Theorem 1** (Fixed-Confidence Upper Bound). *Suppose that $\xi$ satisfies* $\sqrt{\xi} \max_{\tilde{x}_i, \tilde{x}_j \in \tilde{\mathcal{X}}_v, v \in [V]} \|\phi(\tilde{x}_i) - \phi(\tilde{x}_j)\|_{A(\xi, \lambda_1^*)^{-1}} \leq \frac{\Delta_{\min}}{32(1+\varepsilon)\|\theta^*\|}$. *With probability at least* $1 - \delta$, `CoKernelFC` *returns the best arms $\tilde{x}_{v,*}$ for all $v \in [V]$, with per-agent sample complexity*

$$S = O\left(\frac{\rho^*(\xi)}{V} \cdot \log \Delta_{\min}^{-1} \left(\log\left(\frac{nV}{\delta}\right) + \log\log \Delta_{\min}^{-1}\right) + \frac{\tilde{d}(\xi, \lambda_1^*)}{V} \cdot \log \Delta_{\min}^{-1}\right)$$

*and communication rounds $O(\log \Delta_{\min}^{-1})$.*

**Remark 1.** The condition on regularization parameter $\xi$ implies that `CoKernelFC` needs a small regularization parameter such that the bias due to regularization is smaller than $\frac{\Delta_{\min}}{2}$. Such conditions are similarly needed in prior kernel bandit work (Camilleri et al., 2021), and can be dropped in the extended PAC setting (allowing a gap between the identified best arm and true best arm). The $\tilde{d}(\xi, \lambda_1^*) \log(\Delta_{\min}^{-1})/V$ term is a cost for using rounding procedure `ROUND`. This is a second order term when the reward gaps $\Delta_{v,i} := F(\tilde{x}_{v,*}) - F(\tilde{x}_{v,i}) < 1$ for all $\tilde{x}_{v,i} \in \tilde{\mathcal{X}}_v \setminus \{\tilde{x}_{v,*}\}$ and $v \in [V]$, which is the common case in pure exploration (Fiez et al., 2019; Zhu et al., 2021).

When all tasks are the same, Theorem 1 achieves a $V$-speedup, since replicating $V$ copies of single-agent algorithms (Camilleri et al., 2021; Zhu et al., 2021) to complete all tasks without communica-

tion will cost $\tilde{O}(\rho^*(\xi))$ samples per agent. When tasks are totally different, there is no speedup in Theorem 1, since each copy of single-agent algorithm solves a task in her own sub-dimension and costs $\tilde{O}(\frac{\rho^*(\xi)}{V})$ samples ($\rho^*(\xi)$ stands for the effective dimension of *all* tasks). This result matches the restriction of speedup.

**Interpretation.** Now, we interpret Theorem 1 by standard measures in kernel bandits and a decomposition with respect to task similarities and task features. Below we first introduce the definitions of maximum information gain and effective dimension, which are adapted from kernel bandits with regret minimization (Srinivas et al., 2010; Valko et al., 2013) to the pure exploration setting.

We define the *maximum information gain* as $\Upsilon := \max_{\lambda \in \triangle_{\tilde{\mathcal{X}}}} \log \det \left( I + \xi^{-1} K_\lambda \right)$, where $K_\lambda := \left[ \sqrt{\lambda_i \lambda_j} k(\tilde{x}_i, \tilde{x}_j) \right]_{i,j \in [nV]}$ denotes the kernel matrix under sample allocation $\lambda$. $\Upsilon$ stands for the maximum information gain obtained from the samples generated according to sample allocation $\lambda$.

Let $\lambda^* := \operatorname{argmax}_{\lambda \in \triangle_{\tilde{\mathcal{X}}}} \log \det \left( I + \xi^{-1} K_\lambda \right)$ denote the sample allocation which achieves the maximum information gain, and let $\alpha_1 \geq \cdots \geq \alpha_{nV}$ denote the eigenvalues of $K_{\lambda^*}$. Then, we define the *effective dimension* as $d_{\text{eff}} := \min\{j \in [nV] : j\xi \log(nV) \geq \sum_{i=j+1}^{nV} \alpha_i\}$. $d_{\text{eff}}$ stands for the number of principle directions that data projections in RKHS spread.

Recall that $K_{\mathcal{Z}} := [k_{\mathcal{Z}}(z_v, z_{v'})]_{v,v' \in [V]}$ denotes the kernel matrix of task similarities. Let $K_{\mathcal{X}, \lambda^*} := \left[ \sqrt{\lambda_i^* \lambda_j^*} k_{\mathcal{X}}(x_i, x_j) \right]_{i,j \in [nV]}$ denote the kernel matrix of arm features under sample allocation $\lambda^*$.

**Corollary 1.** *The per-agent sample complexity $S$ of algorithm* `CoKernelFC` *can be bounded by*

$$\text{(a) } S = \tilde{O}\left( \frac{\Upsilon}{\Delta_{\min}^2 V} \right), \quad \text{(b) } S = \tilde{O}\left( \frac{d_{\text{eff}}}{\Delta_{\min}^2 V} \right), \quad \text{(c) } S = \tilde{O}\left( \frac{\operatorname{rank}(K_{\mathcal{Z}}) \cdot \operatorname{rank}(K_{\mathcal{X}, \lambda^*})}{\Delta_{\min}^2 V} \right),$$

*where $\tilde{O}(\cdot)$ omits the rounding cost term $\tilde{d}(\xi, \lambda_1^*) \log(\Delta_{\min}^{-1})/V$ and logarithmic factors.*

**Remark 2.** Corollaries 1(a),(b) show that, our sample complexity can be bounded by the maximum information gain, and only depends on the effective dimension of kernel representation. Corollary 1(c) reveals that the more tasks are similar (i.e., the smaller $\operatorname{rank}(K_{\mathcal{Z}})$ is), the fewer samples agents need. For example, when all tasks are the same, i.e., $\operatorname{rank}(K_{\mathcal{Z}}) = 1$, each agent only needs a $\frac{1}{V}$ fraction of the samples required by single-agent algorithms (Camilleri et al., 2021; Zhu et al., 2021) (which need $\tilde{O}(\operatorname{rank}(K_{\mathcal{X}, \lambda^*})/\Delta_{\min}^2)$ samples). Conversely, when all tasks are totally different, i.e., $\operatorname{rank}(K_{\mathcal{Z}}) = V$, no advantage can be obtained from communication, since the information from other agents is useless for solving local tasks. Our experimental results also reflect this relationship between task similarities and speedup, which match our theoretical bounds (see Section 6). We note that these theoretical results hold for both finite and infinite RKHS.

## 5 FIXED-BUDGET CoPE-KB

For the fixed-budget objective, we propose algorithm `CoKernelFB` and error probability guarantees. Due to space limit, we defer the pseudo-code and detailed description to Appendix C.4.

### 5.1 ALGORITHM `CoKernelFB`

`CoKernelFB` pre-determines the number of rounds and the number of samples according to the principle dimension of arms, and successively cut down candidate arms to a half based on principle dimension. `CoKernelFB` also adopts the efficient kernelized estimator (in Section 4.2) to estimate the rewards of arms, so that agents only need to transmit average outcomes rather than all raw sample outcomes or the whole estimated reward parameter. Thus, `CoKernelFB` only requires a $O(nV)$ communication cost, instead of $O(N^{(r)})$ or $O(\dim(\mathcal{H}))$ as in adaptions of prior single-agent and fixed-budget algorithm (Katz-Samuels et al., 2020).

### 5.2 THEORETICAL PERFORMANCE OF `CoKernelFB`

Define the principle dimension of $\tilde{\mathcal{X}}$ as $\omega(\xi, \tilde{\mathcal{X}}) := \min_{\lambda \in \triangle_{\tilde{\mathcal{X}}}} \max_{\tilde{x}_i, \tilde{x}_j \in \tilde{\mathcal{X}}} \|\phi(\tilde{x}_i) - \phi(\tilde{x}_j)\|_{A(\xi, \lambda)^{-1}}^2$, where $A(\xi, \lambda) := \xi I + \sum_{\tilde{x} \in \tilde{\mathcal{X}}} \lambda_{\tilde{x}} \phi(\tilde{x}) \phi(\tilde{x})^\top$. $\omega(\xi, \tilde{\mathcal{X}})$ represents the principle dimension of data projections in $\tilde{\mathcal{X}}$ to RKHS. Now we provide the error probability guarantee for `CoKernelFB`.

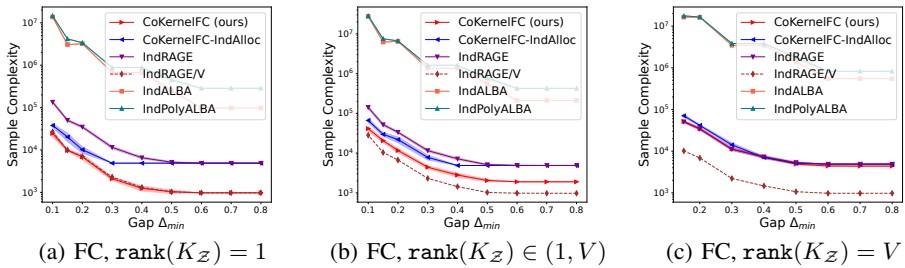

Figure 2: Experimental results for CoPE-KB in the FC setting.

**Theorem 2** (Fixed-Budget Upper Bound). *Suppose that $\xi$ satisfies $\xi \leq \frac{1}{16(1+\varepsilon)^2(\rho^*(\xi))^2\|\theta^*\|^2}$. Algorithm* CoKernelFB *uses at most $T$ samples per agent, and returns the best arms $\tilde{x}_{v,*}$ for all $v \in [V]$ with error probability*

$$O\left(\exp\left(-\frac{TV}{\rho^*(\xi)\cdot\log(\omega(\xi,\tilde{\mathcal{X}}))}\right)\cdot n^2 V \log(\omega(\xi,\tilde{\mathcal{X}}))\right)$$

*and $O(\log(\omega(\xi,\tilde{\mathcal{X}})))$ communication rounds.*

To guarantee an error probability $\delta$, CoKernelFB only requires $\tilde{O}(\frac{\rho^*(\xi)}{V}\log\delta^{-1})$ sample budget, which attains full speedup when all tasks are the same. Theorem 2 can be decomposed into components related to task similarities and arm features as in Corollary 1 (see Appendix E.2).

## 6 EXPERIMENTS

In this section, we provide the experimental results. Here we set $V = 5$, $n = 6$, $\delta = 0.005$, $\mathcal{H} = \mathbb{R}^d$, $d \in \{4, 8, 20\}$, $\theta^* = [0.1, 0.1 + \Delta_{\min}, \dots, 0.1 + (d-1)\Delta_{\min}]^\top$, $\Delta_{\min} \in [0.1, 0.8]$ and $\text{rank}(K_{\mathcal{Z}}) \in [1, V]$. We run 50 independent simulations and plot the average sample complexity with 95% confidence intervals (see Appendix A for a complete setup description and more results).

We compare our algorithm CoKernelFC with five baselines, i.e., CoKernel-IndAlloc, IndRAGE, IndRAGE/$V$, IndALBA and IndPolyALBA. CoKernel-IndAlloc is an ablation variant of CoKernelFC, where agents use locally optimal sample allocations. IndRAGE, IndALBA and IndPolyALBA are adaptions of existing single-agent algorithms RAGE (Fiez et al., 2019), ALBA (Tao et al., 2018) and PolyALBA (Du et al., 2021), respectively. IndRAGE/$V$ is a $V$-speedup baseline, which divides the sample complexity of the best single-agent adaption IndRAGE by $V$.

Figures 2(a), 2(b) show that CoKernelFC achieves the best sample complexity, which demonstrates the effectiveness of our sample allocation and cooperation schemes. Moreover, the empirical results reflect that the more tasks are similar, the higher learning speedup agents attain, which matches our theoretical analysis. Specifically, in the $\text{rank}(K_{\mathcal{Z}}) = 1$ case (Figure 2(a)), i.e., tasks are the same, CoKernelFC matches the $V$-speedup baseline IndRAGE-$V$. In the $\text{rank}(K_{\mathcal{Z}}) \in (1, V)$ case (Figure 2(b)), i.e., tasks are similar, the sample complexity of CoKernelFC lies between IndRAGE/$V$ and IndRAGE, which indicates that CoKernelFC achieves a speedup lower than $V$. In the $\text{rank}(K_{\mathcal{Z}}) = V$ case (Figure 2(c)), i.e., tasks are totally different, CoKernelFC performs similar to IndRAGE, since information sharing brings no advantage in this case.

## 7 CONCLUSION

In this paper, we propose a collaborative pure exploration in kernel bandit (CoPE-KB) model with fixed-confidence and fixed-budget objectives. CoPE-KB generalizes prior CoPE formulation from the single-task and classic MAB setting to allow multiple tasks and general reward structures. We propose novel algorithms with an efficient kernelized estimator and a novel communication scheme. Sample and communication complexities are provided to corroborate the efficiency of our algorithms. Our results explicitly quantify the influences of task similarities on learning speedup, and only depend on the effective dimension of feature space.

## ACKNOWLEDGEMENTS

The work of Yihan Du and Longbo Huang is supported by the Technology and Innovation Major Project of the Ministry of Science and Technology of China under Grant 2020AAA0108400 and 2020AAA0108403, the Tsinghua University Initiative Scientific Research Program, and Tsinghua Precision Medicine Foundation 10001020109. Yuko Kuroki was supported by Microsoft Research Asia and JST ACT-X JPMJAX200E.

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

# APPENDIX

## A  MORE EXPERIMENTS

In this section, we give a complete description of experimental setup, and present the results for the FB setting.

**Experimental Setup.** Our experiments are run on Intel Xeon E5-2660 v3 CPU at 2.60GHz. We set $V = 5$, $n = 6$, $\delta = 0.005$ and $\mathcal{H} = \mathbb{R}^d$, where $d$ is a dimension parameter that will be specified later. We consider three different cases of task similarities, i.e., $\mathrm{rank}(K_{\mathcal{Z}}) = 1$ (tasks are the same), $\mathrm{rank}(K_{\mathcal{Z}}) \in (1, V)$ (tasks are similar), and $\mathrm{rank}(K_{\mathcal{Z}}) = V$ (tasks are totally different), to show how task similarities impact learning performance in practice.

For the $\mathrm{rank}(K_{\mathcal{Z}}) = 1$ case, we set $d = 4$. For any $v \in [V]$, $\{\phi(\tilde{x})\}_{\tilde{x} \in \tilde{\mathcal{X}}_v}$ is the set of all $\binom{4}{2}$ vectors in $\mathbb{R}^4$, where each vector has two entries 0 and two entries 1. For the $\mathrm{rank}(K_{\mathcal{Z}}) \in (1, V)$ case, we set $d = 8$. For any $v \in \{1, 2\}$ and $v' \in \{3, 4, 5\}$, $\{\phi(\tilde{x})\}_{\tilde{x} \in \tilde{\mathcal{X}}_v}$ and $\{\phi(\tilde{x})\}_{\tilde{x} \in \tilde{\mathcal{X}}_{v'}}$ are the two sets of all $\binom{4}{2}$ vectors in the first and second subspaces $\mathbb{R}^4$ of the whole space $\mathbb{R}^8$, respectively, where each vector has two entries 0 and two entries 1. For the $\mathrm{rank}(K_{\mathcal{Z}}) = V$ case, we set $d = 20$. For any $v \in [V]$, $\{\phi(\tilde{x})\}_{\tilde{x} \in \tilde{\mathcal{X}}_v}$ is the set of all $\binom{4}{2}$ vectors in the $v$-th subspace $\mathbb{R}^4$ of the whole space $\mathbb{R}^{20}$, where each vector has two entries 0 and two entries 1. For all cases, we set $\theta^* = [0.1, 0.1 + \Delta_{\min}, \dots, 0.1 + (d-1)\Delta_{\min}]^\top \in \mathbb{R}^d$, where $\Delta_{\min}$ is a reward gap parameter that will be tuned in the experiments.

In the FC setting, we change the reward gap $\Delta_{\min} \in [0.1, 0.8]$ to generate different instances, and run 50 independent simulations to plot the average sample complexity with $95\%$ confidence intervals. In the FB setting, we change the sample budget $T \in [7000, 300000]$ to obtain different instances, and perform 100 independent runs to report the average error probability across runs.

**Results for the Fixed-Budget Setting.** In the FB setting (Figure 3), we compare our algorithm `CoKernelFB` with three baselines, i.e., `CoKernelFB-IndAlloc`, `IndPeaceFB` (Katz-Samuels et al., 2020) and `IndUniformFB`. `CoKernelFB-IndAlloc` is an ablation variant of `CoKernelFB`, where agents use locally optimal sample allocations, instead of a globally optimal sample allocation. `IndPeaceFB` and `IndUniformFB` run $V$ copies of existing single-agent algorithms, `PeaceFB` (Katz-Samuels et al., 2020) and the uniform sampling strategy, to independently complete the $V$ tasks, respectively.

Figures 3(a), 3(b) show that, our `CoKernelFB` enjoys a lower error probability than the baselines. Moreover, the experimental results also reflect the influences of task similarities on learning speedup, and match our theoretical bounds. Specifically, from Figures 3(a) to 3(c), the error probability of `CoKernelFB` gets closer and closer to that of the single-agent adaption `IndPeaceFB`, which shows that the learning speedup of `CoKernelFB` slows down as the task similarity decreases.

## B  RELATED WORK

In this section, we present a complete review of related works.

**Collaborative Pure Exploration (CoPE).** The collaborative pure exploration literature is initiated by (Hillel et al., 2013), where all agents solve a common classic best arm identification problem. Hillel et al. (2013) design fixed-confidence algorithms based on majority vote and provides upper bound analysis. Tao et al. (2019) further develop a fixed-budget algorithm and complete the analysis of communication round-speedup lower bounds. Karpov et al. (2020) extend the best arm identification formulation of (Hillel et al., 2013; Tao et al., 2019) to best $m$ arm identification, and show complexity separations between these two formulations. Our CoPE-KB generalizes prior CoPE works (Hillel et al., 2013; Tao et al., 2019; Karpov et al., 2020) from single-task and classic MAB setting to allow multiple tasks and general reward structures, and faces unique challenges on communication and computation due to high (possibly infinite) dimensional feature space.

**Kernel Bandits.** For kernel bandits with the regret minimization objective, Srinivas et al. (2010) and Valko et al. (2013) design algorithms from Bayesian and frequentist perspectives, respectively.

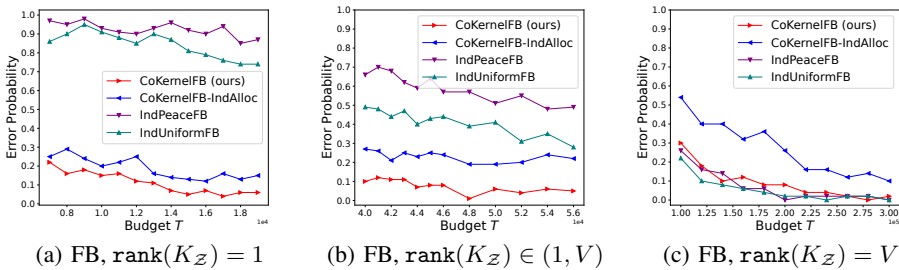

Figure 3: Experimental results for CoPE-KB in the FB setting.

Chowdhury & Gopalan (2017) improve the regret bound in (Srinivas et al., 2010) by building a new self-normalized concentration inequality for kernel bandits. Scarlett et al. (2017) develop regret lower bounds for the squared-exponential kernel and Matérn kernel, and Li & Scarlett (2022) establish a near-optimal regret upper bound. Krause & Ong (2011); Deshmukh et al. (2017) study multi-task kernel bandits, which consider a composite kernel constituted by two sub-kernels with respect to tasks and items. Dubey et al. (2020) investigate multi-agent kernel bandits with a local communication protocol, where agents can immediately share the observed data with their neighbors in a network. Li et al. (2022) study distributed kernel bandits with a client-server communication protocol, where agents only communicate with a central server. The communication costs in (Dubey et al., 2020; Li et al., 2022) depend on the total number of timesteps $T$ in the regret minimization game, while our communication costs depend only on the number of arms in the pure exploration setting.

For kernel bandits with the pure exploration objective, there are only a few related works (Scarlett et al., 2017; Vakili et al., 2021; Camilleri et al., 2021; Zhu et al., 2021), and all of them study the single-agent formulation. Scarlett et al. (2017); Vakili et al. (2021) consider the simple regret setting in pure exploration, where an agent reports an arm at each timestep and aims to minimize the suboptimality of the reported arm (simple regret) after $T$ timesteps. In contrast, Camilleri et al. (2021); Zhu et al. (2021) investigate the best arm identification setting in pure exploration, where an agent aims to identify the best arm and minimizes the number of samples used (our work also falls in the best arm identification line). Camilleri et al. (2021) propose a robust inverse propensity score estimator to save the samples needed for rounding procedures. Zhu et al. (2021) use neural networks to approximate nonlinear reward functions. The above kernel bandit works consider either the regret minimization or single-agent setting, which is different from our CoPE-KB problem and does not involve our challenges on round-speedup trade-off analysis and communication. Thus, these works cannot be applied to solve our problem.

Recently, there are several works which study federated/distributed bandits with the regret minimization objective. Wang et al. (2019) study distributed multi-armed and linear bandit problems, and He et al. (2022) investigate federated linear bandits with asynchronous communication. Wang et al. (2019); He et al. (2022); Dubey & Pentland (2020); Huang et al. (2021); Li & Wang (2022) develop algorithms for (low-dimensional) federated linear bandits, where agents directly communicate the whole vectors of reward parameters. These works cannot be adapted to resolve our challenges, because under kernel representation, the reward parameter is high-dimensional and expensive to explicitly calculate or transmit.

## C   OMITTED DISCUSSION FOR ALGORITHMS

### C.1   ROUNDING PROCEDURE

In algorithms `CoKernelFC` and `CoKernelFB`, we use a rounding procedure $\texttt{ROUND}(\xi, \lambda, N, \varepsilon)$ (Algorithm 2 in (Camilleri et al., 2021)). For any weighted sample allocation $\lambda \in \triangle_{\tilde{\mathcal{X}}}$, regularization parameter $\xi$ and approximation parameter $\varepsilon$, with $N \geq \tau(\xi, \lambda, \varepsilon) = O(\frac{\tilde{d}(\xi, \lambda)}{\varepsilon^2})$ samples,

$\text{ROUND}(\xi, \lambda, N, \varepsilon)$ returns a discrete sample allocation $(\tilde{s}_1, \ldots, \tilde{s}_N) \in \tilde{\mathcal{X}}^N$ such that

$$\max_{\tilde{x}_i, \tilde{x}_j \in \tilde{\mathcal{X}}_v, v \in [V]} \|\phi(\tilde{x}_i) - \phi(\tilde{x}_j)\|^2_{(N \xi I + \sum_{i=1}^{N} \phi(\tilde{s}_i) \phi(\tilde{s}_i)^\top)^{-1}}$$

$$\leq 2(1 + \epsilon) \max_{\tilde{x}_i, \tilde{x}_j \in \tilde{\mathcal{X}}_v, v \in [V]} \|\phi(\tilde{x}_i) - \phi(\tilde{x}_j)\|^2_{(N \xi I + \sum_{\tilde{x} \in \tilde{\mathcal{X}}} N \lambda_{\tilde{x}} \phi(\tilde{x}) \phi(\tilde{x})^\top)^{-1}}. \tag{6}$$

This rounding procedure requires the number of samples $N$ to satisfy that $N \geq \tau(\xi, \lambda, \varepsilon) = O(\frac{\tilde{d}(\xi, \lambda)}{\varepsilon^2})$. Here $\tilde{d}(\xi, \lambda)$ is the number of the eigenvalues of matrix $\sum_{\tilde{x} \in \tilde{\mathcal{X}}} \lambda_{\tilde{x}} \phi(\tilde{x}) \phi(\tilde{x})^\top$ which are greater than $\xi$. $\tilde{d}(\xi, \lambda)$ stands for the effective dimension of the feature space spanned by data projections of $\tilde{\mathcal{X}}$ under regularization parameter $\xi$.

## C.2 KERNELIZED COMPUTATION IN ALGORITHM CoKernelFC

**Kernelized Estimator.** Below we present a derivation of our kernelized estimator (Eq. (3)), which plays an important role in boosting the communication and computation efficiency.

Let $\hat{\theta}_r$ denote the minimizer of the following regularized least square loss function:

$$\mathcal{L}(\theta) = N^{(r)} \xi \|\theta\|^2 + \sum_{j=1}^{N^{(r)}} (y_j - \phi(\tilde{s}_j)^\top \theta)^2.$$

Letting the derivative of $\mathcal{L}(\theta)$ equal to zero, we have

$$N^{(r)} \xi \hat{\theta}_r + \sum_{j=1}^{N^{(r)}} \phi(\tilde{x}_j) \phi(\tilde{x}_j)^\top \hat{\theta}_r = \sum_{j=1}^{N^{(r)}} \phi(\tilde{x}_j) y_j.$$

Merging repetitive computations for the same arms, we can obtain

$$N^{(r)} \xi \hat{\theta}_r + \left( \sum_{i=1}^{nV} N_i^{(r)} \phi(\tilde{x}_i) \phi(\tilde{x}_i)^\top \right) \hat{\theta}_r = \sum_{i=1}^{nV} N_i^{(r)} \phi(\tilde{x}_i) \bar{y}_i^{(r)}, \tag{7}$$

where $N_i^{(r)}$ is the number of samples and $\bar{y}_i^{(r)}$ is the average observation on arm $\tilde{x}_i$ for any $i \in [nV]$. Let $\Phi_r = [\sqrt{N_1^{(r)}} \phi(\tilde{x}_1)^\top; \ldots; \sqrt{N_{nV}^{(r)}} \phi(\tilde{x}_{nV})^\top]$, $K^{(r)} = \Phi_r \Phi_r^\top = [\sqrt{N_i^{(r)} N_j^{(r)}} k(\tilde{x}_i, \tilde{x}_j)]_{i,j \in [nV]}$ and $\bar{y}^{(r)} = [\sqrt{N_1^{(r)}} \bar{y}_1^{(r)}, \ldots, \sqrt{N_{nV}^{(r)}} \bar{y}_{nV}^{(r)}]^\top$. Then, we can write Eq. (7) as

$$\left( N^{(r)} \xi I + \Phi_r^\top \Phi_r \right) \hat{\theta}_r = \Phi_r^\top \bar{y}^{(r)}.$$

Since $\left( N^{(r)} \xi I + \Phi_r^\top \Phi_r \right) \succ 0$ and $\left( N^{(r)} \xi I + \Phi_r \Phi_r^\top \right) \succ 0$,

$$\hat{\theta}_r = \left( N^{(r)} \xi I + \Phi_r^\top \Phi_r \right)^{-1} \Phi_r^\top \bar{y}^{(r)}$$

$$= \Phi_r^\top \left( N^{(r)} \xi I + \Phi_r \Phi_r^\top \right)^{-1} \bar{y}^{(r)}$$

$$= \Phi_r^\top \left( N^{(r)} \xi I + K^{(r)} \right)^{-1} \bar{y}^{(r)}.$$

Let $k_r(\tilde{x}) = \Phi_r \phi(\tilde{x}) = [\sqrt{N_1^{(r)}} k(\tilde{x}, \tilde{x}_1), \ldots, \sqrt{N_{nV}^{(r)}} k(\tilde{x}, \tilde{x}_{nV})]^\top$ for any $\tilde{x} \in \tilde{\mathcal{X}}$. Then, for any arms $\tilde{x}_i, \tilde{x}_j \in \tilde{\mathcal{X}}$, we obtain the efficient kernelized estimators of the expected reward $F(\tilde{x}_i)$ and expected reward gap $F(\tilde{x}_i) - F(\tilde{x}_j)$ as

$$\hat{f}_r(\tilde{x}_i) = \phi(\tilde{x}_i)^\top \hat{\theta}_r = k_r(\tilde{x}_i)^\top \left( N^{(r)} \xi I + K^{(r)} \right)^{-1} \bar{y}^{(r)},$$

$$\hat{\Delta}_r(\tilde{x}_i, \tilde{x}_j) = (k_r(\tilde{x}_i) - k_r(\tilde{x}_j))^\top \left( N^{(r)} \xi I + K^{(r)} \right)^{-1} \bar{y}^{(r)}.$$

**Kernelized Optimization Solver.** Following (Camilleri et al., 2021), we use a kernelized optimization solver for the min-max optimization in Line 4 of Algorithm 1. The optimization problem is as follows.

$$\min_{\lambda \in \triangle_{\tilde{\mathcal{X}}}} \max_{\tilde{x}_i, \tilde{x}_j \in \mathcal{B}_v^{(r)}, v \in [V]} \|\phi(\tilde{x}_i) - \phi(\tilde{x}_j)\|^2_{A(\xi, \lambda)^{-1}}, \tag{8}$$

where $A(\xi, \lambda) := \xi I + \sum_{\tilde{x} \in \tilde{\mathcal{X}}} \lambda_{\tilde{x}} \phi(\tilde{x}) \phi(\tilde{x})^\top$ for any $\xi \geq 0$ and $\lambda \in \triangle_{\tilde{\mathcal{X}}}$.

Define function $h(\lambda) = \max_{\tilde{x}_i, \tilde{x}_j \in \mathcal{B}_v^{(r)}, v \in [V]} \|\phi(\tilde{x}_i) - \phi(\tilde{x}_j)\|_{A(\xi, \lambda)^{-1}}^2$, and define $\tilde{x}_i^*(\lambda)$, $\tilde{x}_j^*(\lambda)$ as the optimal solution of $h(\lambda)$. Then, the gradient of $h(\lambda)$ with respect to $\lambda$ is

$$[\nabla_\lambda h(\lambda)]_{\tilde{x}} = - \left( \left( \phi(\tilde{x}_i^*(\lambda)) - \phi(\tilde{x}_j^*(\lambda)) \right)^\top A(\xi, \lambda)^{-1} \phi(\tilde{x}) \right)^2, \forall \tilde{x} \in \tilde{\mathcal{X}}. \tag{9}$$

Next, we show how to efficiently compute gradient $[\nabla_\lambda h(\lambda)]_{\tilde{x}}$ with kernel function $k(\cdot, \cdot)$.

Since $\left( \xi I + \Phi_\lambda^\top \Phi_\lambda \right) \phi(\tilde{x}) = \xi \phi(\tilde{x}) + \Phi_\lambda^\top k_\lambda(\tilde{x})$ for any $\tilde{x} \in \tilde{\mathcal{X}}$, we have

$$\phi(\tilde{x}) = \xi \left( \xi I + \Phi_\lambda^\top \Phi_\lambda \right)^{-1} \phi(\tilde{x}) + \left( \xi I + \Phi_\lambda^\top \Phi_\lambda \right)^{-1} \Phi_\lambda^\top k_\lambda(\tilde{x})$$
$$= \xi \left( \xi I + \Phi_\lambda^\top \Phi_\lambda \right)^{-1} \phi(\tilde{x}) + \Phi_\lambda^\top \left( \xi I + K_\lambda \right)^{-1} k_\lambda(\tilde{x})$$

Multiplying $\left( \phi(\tilde{x}_i^*(\lambda)) - \phi(\tilde{x}_j^*(\lambda)) \right)^\top$ on both sides, we have

$$\left( \phi(\tilde{x}_i^*(\lambda)) - \phi(\tilde{x}_j^*(\lambda)) \right)^\top \phi(\tilde{x})$$
$$= \xi \left( \phi(\tilde{x}_i^*(\lambda)) - \phi(\tilde{x}_j^*(\lambda)) \right)^\top \left( \xi I + \Phi_\lambda^\top \Phi_\lambda \right)^{-1} \phi(\tilde{x})$$
$$+ \left( k_\lambda(\tilde{x}_i^*(\lambda)) - k_\lambda(\tilde{x}_j^*(\lambda)) \right)^\top \left( \xi I + K_\lambda \right)^{-1} k_\lambda(\tilde{x})$$

Then,

$$\left( \phi(\tilde{x}_i^*(\lambda)) - \phi(\tilde{x}_j^*(\lambda)) \right)^\top \left( \xi I + \Phi_\lambda^\top \Phi_\lambda \right)^{-1} \phi(\tilde{x})$$
$$= \xi^{-1} \left( \phi(\tilde{x}_i^*(\lambda)) - \phi(\tilde{x}_j^*(\lambda)) \right)^\top \phi(\tilde{x}) - \xi^{-1} \left( k_\lambda(\tilde{x}_i^*(\lambda)) - k_\lambda(\tilde{x}_j^*(\lambda)) \right)^\top \left( \xi I + K_\lambda \right)^{-1} k_\lambda(\tilde{x})$$
$$= \xi^{-1} \left( k(\tilde{x}_i^*(\lambda), \tilde{x}) - k(\tilde{x}_j^*(\lambda), \tilde{x}) - \left( k_\lambda(\tilde{x}_i^*(\lambda)) - k_\lambda(\tilde{x}_j^*(\lambda)) \right)^\top \left( \xi I + K_\lambda \right)^{-1} k_\lambda(\tilde{x}) \right) \tag{10}$$

Therefore, we can compute gradient $\nabla_\lambda h(\lambda)$ (Eq. (9)) using the equivalent kernelized expression Eq. (10), and then the optimization (Eq. (8)) can be efficiently solved by projected gradient descent.

## C.3 COMPARISON OF ESTIMATORS

In the following, we compare our kernelized estimator (Eq. (3)) to those in prior kernel bandit works (Zhou et al., 2020; Zhu et al., 2021; Dubey et al., 2020; Camilleri et al., 2021) and adaptions of federated linear bandits (Dubey & Pentland, 2020; Huang et al., 2021; Li & Wang, 2022).

First, prior kernel bandit works (Zhou et al., 2020; Zhu et al., 2021) or adaptions of federated linear bandits (Dubey & Pentland, 2020; Huang et al., 2021; Li & Wang, 2022) explicitly maintain the regularized least square estimator of reward parameter $\theta^*$ as

$$\hat{\theta}_r = \left( N^{(r)} \xi I + \sum_{j=1}^{N^{(r)}} \phi(\tilde{s}_j) \phi(\tilde{s}_j)^\top \right)^{-1} \sum_{j=1}^{N^{(r)}} \phi(\tilde{s}_j) y_j. \tag{11}$$

Since both $\hat{\theta}_r$ and $\phi(\tilde{s}_j)$ lie in the high-dimensional feature space $\mathcal{H}$, this estimator will incur $O(\dim(\mathcal{H}))$ computation and communication costs.

Second, (Dubey et al., 2020) and Algorithm 6 in (Camilleri et al., 2021) use a redundant kernelized form of $\hat{\theta}_r$ as

$$\hat{\theta}_r = (\Psi^\top \Psi + N^{(r)} \xi I)^{-1} \Psi^\top Y,$$

where $\Psi := [\phi(\tilde{s}_1)^\top; \ldots; \phi(\tilde{s}_{N^{(r)}})^\top] \in \mathbb{R}^{N^{(r)} \times \dim(\mathcal{H})}$, $Y := [y_1, \ldots, y_{N^{(r)}}]^\top \in \mathbb{R}^{N^{(r)}}$, and $y_1, \ldots, y_{N^{(r)}}$ are the observed raw outcomes of sample sequence $(\tilde{s}_1, \ldots, \tilde{s}_{N^{(r)}})$. This estimator needs all $N^{(r)}$ raw sample outcomes $y_1, \ldots, y_{N^{(r)}}$ as inputs, rather than only $nV$ average outcomes $\bar{y}_1^{(r)}, \ldots, \bar{y}_{nV}^{(r)}$ as our estimator (Eq. (2)), which will incur a $O(N^{(r)})$ communication cost. Here the number of samples $N^{(r)}$ is often far larger than the number of arms $nV$.

Finally, Algorithm 4 in (Camilleri et al., 2021) adopts a robust inverse propensity score estimator as

$$\hat{\theta}_r := \arg\min_\theta \max_{\tilde{x}_i, \tilde{x}_{i'} \in \mathcal{B}_v^{(r)}, v \in [V]} \frac{\left| (\phi(\tilde{x}_i) - \phi(\tilde{x}_{i'}))^\top \theta - W^{i,i'} \right|}{\|\phi(\tilde{x}_i) - \phi(\tilde{x}_{i'})\|_{A(\xi, \lambda_r^*)^{-1}}},$$

---

**Algorithm 2** Collaborative Multi-agent Algorithm `CoKernelFB`: for Agent $v \in [V]$

1: **Input:** regularization parameter $\xi \leq \frac{1}{16(1+\varepsilon)^2(\rho^*(\xi))^2\|\theta^*\|^2}$, per-agent budget $T \geq \frac{\max\{\rho^*(\xi),\ \tilde{d}(\xi,\lambda_1^*)\}}{V}\log(\omega(\xi,\tilde{\mathcal{X}}))$, arm set $\tilde{\mathcal{X}}$ which satisfies $\omega(\xi,\{\tilde{x}_{v,*},\tilde{x}\}) \geq 1$ for all $\tilde{x} \in \tilde{\mathcal{X}}_v \setminus \{\tilde{x}_{v,*}\}$ and $v \in [V]$, $k(\cdot,\cdot): \tilde{\mathcal{X}} \times \tilde{\mathcal{X}} \mapsto \mathbb{R}$, rounding procedure ROUND, rounding approximation parameter $\varepsilon = \frac{1}{10}$.

2: **Initialization:** $R \leftarrow \lceil \log_2(\omega(\xi,\tilde{\mathcal{X}})) \rceil$. $N \leftarrow \lfloor TV/R \rfloor$. $\mathcal{B}_{v'}^{(1)} \leftarrow \tilde{\mathcal{X}}_{v'}$ for all $v' \in [V]$. $r \leftarrow 1$.   // pre-determine the number of rounds and samples

3: **while** $r \leq R$ **and** $\exists v' \in [V], |\mathcal{B}_{v'}^{(r)}| > 1$ **do**

4:     Let $\lambda_r^*$ and $\rho_r^*$ be the optimal solution and optimal value of
$$\min_{\lambda \in \triangle_{\tilde{\mathcal{X}}}} \max_{\tilde{x}_i,\tilde{x}_j \in \mathcal{B}_{v'}^{(r)}, v' \in [V]} \|\phi(\tilde{x}_i) - \phi(\tilde{x}_j)\|^2_{(\xi I + \sum_{\tilde{x} \in \tilde{\mathcal{X}}} \lambda_{\tilde{x}} \phi(\tilde{x})\phi(\tilde{x})^\top)^{-1}}$$   // compute the optimal sample allocation

5:     $(\tilde{s}_1,\ldots,\tilde{s}_N) \leftarrow \text{ROUND}(\xi,\lambda_r^*,N,\varepsilon)$

6:     Extract a sub-sequence $\tilde{s}_v^{(r)}$ from $(\tilde{s}_1,\ldots,\tilde{s}_N)$ which only contains arms in $\tilde{\mathcal{X}}_v$

7:     Sample $\tilde{s}_v^{(r)}$ and observe random rewards $\boldsymbol{y}_v^{(r)}$

8:     Broadcast $\{(N_{v,i}^{(r)}, \bar{y}_{v,i}^{(r)})\}_{i\in[n]}$, and receive $\{(N_{v',i}^{(r)}, \bar{y}_{v',i}^{(r)})\}_{i\in[n]}$ from all other agents $v' \in [V] \setminus \{v\}$

9:     **for all** $v' \in [V]$ **do**

10:        $\hat{F}_r(\tilde{x}) \leftarrow k_r(\tilde{x})^\top (K^{(r)} + N^{(r)}\xi I)^{-1} \bar{\boldsymbol{y}}^{(r)}$ for all $\tilde{x} \in \mathcal{B}_{v'}^{(r)}$   // estimate the rewards of alive arms

11:        Sort all $\tilde{x} \in \mathcal{B}_{v'}^{(r)}$ by $\hat{F}_r(\tilde{x})$ in decreasing order, and denote the sorted sequence by $\tilde{x}_{(1)},\ldots,\tilde{x}_{(|\mathcal{B}_{v'}^{(r)}|)}$

12:        Let $i_{r+1}$ be the largest index such that $\omega(\xi,\{\tilde{x}_{(1)},\ldots,\tilde{x}_{(i_{r+1})}\}) \leq \omega(\xi,\mathcal{B}_{v'}^{(r)})/2$

13:        $\mathcal{B}_{v'}^{(r+1)} \leftarrow \{\tilde{x}_{(1)},\ldots,\tilde{x}_{(i_{r+1})}\}$   // cut down the alive arm set to half dimension

14:     **end for**

15:     $r \leftarrow r + 1$

16: **end while**

17: **return** $\mathcal{B}_1^{(r)},\ldots,\mathcal{B}_V^{(r)}$

---

$$W^{i,i'} := \hat{\mu}\left(\left\{(\phi(\tilde{x}_i) - \phi(\tilde{x}_{i'}))^\top A(\xi,\lambda_r^*)^{-1}\phi(\tilde{s}_j)y_j\right\}_{j=1}^{N^{(r)}}\right),$$

Here $A(\xi,\lambda) := \xi I + \sum_{\tilde{x} \in \tilde{\mathcal{X}}} \lambda_{\tilde{x}} \phi(\tilde{x})\phi(\tilde{x})^\top$ for any $\xi \geq 0$ and $\lambda \in \triangle_{\tilde{\mathcal{X}}}$. $\lambda_r^*$ is the optimal sample allocation defined in Line 4 of algorithm `CoKernelFC` (Algorithm 1). $\hat{\mu}(\cdot)$ is the median-of-means estimator or Catoni's estimator (Lugosi & Mendelson, 2019). This estimator also requires all $N^{(r)}$ raw sample outcomes $y_1,\ldots,y_{N^{(r)}}$, and will cause a $O(N^{(r)})$ communication cost.

## C.4   Pseudo-code and Description of Algorithm `CoKernelFB`

In this section, we present the algorithm pseudo-code of `CoKernelFB` (in Algorithm 2), and give a detailed algorithm description.

The procedure of `CoKernelFB` is as follows. During initialization (Line 2), we pre-determine the number of rounds $R$ and the number of samples $N$ for each round according to data dimension $\omega(\xi,\tilde{\mathcal{X}})$, which is formally defined as

$$\omega(\xi,\tilde{\mathcal{S}}) := \min_{\lambda \in \triangle_{\tilde{\mathcal{X}}}} \max_{\tilde{x}_i,\tilde{x}_j \in \tilde{\mathcal{S}}} \|\phi(\tilde{x}_i) - \phi(\tilde{x}_j)\|^2_{A(\xi,\lambda)^{-1}}, \ \forall \tilde{\mathcal{S}} \subseteq \tilde{\mathcal{X}},$$

where $A(\xi,\lambda) := \xi I + \sum_{\tilde{x} \in \tilde{\mathcal{X}}} \lambda_{\tilde{x}} \phi(\tilde{x})\phi(\tilde{x})^\top$. $\omega(\xi,\tilde{\mathcal{S}})$ represents the *principle dimension* of data projections in $\tilde{\mathcal{S}}$ to the RKHS under regularization parameter $\xi$. Each agent $v$ maintains alive arm sets $\mathcal{B}_{v'}^{(r)}$ for all agents $v' \in [V]$, and calculates a global optimal sample allocation $\lambda_r^*$ (Line 4). Then, she generates a sample sequence $(\tilde{s}_1^{(r)},\ldots,\tilde{s}_N^{(r)})$ according to $\lambda_r^*$, and selects the sub-sequence $\tilde{\boldsymbol{s}}_v^{(r)}$ that only contains her available arms to perform sampling (Lines 5,6). Similar to `CoKernelFC`, she only communicates the number of samples $N_{v,i}^{(r)}$ and average observed reward $\bar{y}_{v,i}^{(r)}$ for each arm $\tilde{x}_{v,i}$

with other agents (Line 8). With the shared information, she estimates rewards of alive arms and only keeps the best half of them in the dimension sense (Lines 10-13).

Regarding the conditions on the input parameters (Line 1), the condition on regularization parameter $\xi$ implies that `CoKernelFC` needs a small regularization parameter such that the bias due to regularization is small. Such conditions are similarly needed in prior kernel bandit work (Camilleri et al., 2021), and can be dropped in the extended PAC setting (allowing a gap between the identified best arm and true best arm). The condition on $T$ is to ensure that the given sample budget is larger than the number of samples required by rounding procedure `ROUND`. In addition, the condition on $\omega(\cdot, \cdot)$ is to guarantee that the number of rounds is bounded by $O(\log(\omega(\xi, \tilde{\mathcal{X}})))$ rather than an uncontrollable variable. Such conditions are also needed by prior fixed-budget pure exploration algorithm (Katz-Samuels et al., 2020).

**Communication and Computation.** `CoKernelFB` also adopts the kernelized estimator in Section 4.2 to estimate the rewards of alive arms (Line 10). Specifically, using Eq. (2), we can estimate the reward for arm $\tilde{x}$ by

$$\hat{F}_r(\tilde{x}) = \phi(\tilde{x})^\top \hat{\theta}_r = k_r(\tilde{x})^\top \left( N^{(r)}\xi I + K^{(r)} \right)^{-1} \bar{y}^{(r)}, \tag{12}$$

where $\bar{y}^{(r)} := [\sqrt{N_1^{(r)}}\bar{y}_1^{(r)}, \ldots, \sqrt{N_{nV}^{(r)}}\bar{y}_{nV}^{(r)}]^\top$, $K^{(r)} := [\sqrt{N_i^{(r)}N_j^{(r)}}k(\tilde{x}_i, \tilde{x}_j)]_{i,j \in [nV]}$ and $k_r(\tilde{x}) := [\sqrt{N_1^{(r)}}k(\tilde{x}, \tilde{x}_1), \ldots, \sqrt{N_{nV}^{(r)}}k(\tilde{x}, \tilde{x}_{nV})]^\top$ are the average outcomes, kernel matrix and correlations of the $nV$ arms, respectively, which *merge* repetitive information on same arms.

Using the kernelized estimator (Eq. (12)), `CoKernelFB` only requires a $O(nV)$ communication cost, instead of $O(N^{(r)})$ or $O(\dim(\mathcal{H}))$ as in adaptions of existing single-agent algorithm (Katz-Samuels et al., 2020).

# D PROOFS FOR THE FIXED-CONFIDENCE SETTING

## D.1 PROOF OF THEOREM 1

Our proof of Theorem 1 adapts the analytical procedure of (Fiez et al., 2019; Katz-Samuels et al., 2020) to the multi-agent setting.

Let $r^* := \lceil \log_2 \frac{2}{\Delta_{\min}} \rceil + 1$. Intuitively, $r^*$ is the upper bound of the number of rounds used by algorithm `CoKernelFC`. For any $\lambda \in \triangle_{\tilde{\mathcal{X}}}$, let $\Phi_\lambda := [\sqrt{\lambda_1}\phi(\tilde{x}_1)^\top; \ldots; \sqrt{\lambda_{nV}}\phi(\tilde{x}_{nV})^\top]$, where $\lambda_i$ denotes the weight allocated to arm $\tilde{x}_i$ for any $i \in [nV]$. For any $\xi \geq 0$ and $\lambda \in \triangle_{\tilde{\mathcal{X}}}$, $A(\xi, \lambda) := \xi I + \sum_{\tilde{x} \in \tilde{\mathcal{X}}} \lambda_{\tilde{x}}\phi(\tilde{x})\phi(\tilde{x})^\top = \xi I + \Phi_\lambda^\top \Phi_\lambda$.

The regularization parameter $\xi$ in algorithm `CoKernelFC` satisfies $\sqrt{\xi} \max_{\tilde{x}_i, \tilde{x}_j \in \tilde{\mathcal{X}}_v, v \in [V]} \|\phi(\tilde{x}_i) - \phi(\tilde{x}_j)\|_{A(\xi, \lambda_1^*)^{-1}} \leq \frac{\Delta_{\min}}{32(1+\varepsilon)\|\theta^*\|}$. Since $\max_{\tilde{x}_i, \tilde{x}_j \in \tilde{\mathcal{B}}_v^{(r)}, v \in [V]} \|\phi(\tilde{x}_i) - \phi(\tilde{x}_j)\|_{A(\xi, \lambda_r^*)^{-1}}$ is non-increasing with respect to $r$ (from Line 4 in algorithm `CoKernelFC`), we have $2(1 + \varepsilon)\sqrt{\xi} \max_{\tilde{x}_i, \tilde{x}_j \in \tilde{\mathcal{B}}_v^{(r)}, v \in [V]} \|\phi(\tilde{x}_i) - \phi(\tilde{x}_j)\|_{A(\xi, \lambda_r^*)^{-1}}\|\theta^*\| \leq \frac{\Delta_{\min}}{16} \leq \frac{1}{2^{r+1}}$ for any $1 \leq r \leq r^*$. In order to prove Theorem 1, we first introduce several important lemmas.

**Lemma 1** (Concentration). *Defining event*

$$\mathcal{G} = \left\{ \left| \left( \hat{F}_r(\tilde{x}_i) - \hat{F}_r(\tilde{x}_j) \right) - (F(\tilde{x}_i) - F(\tilde{x}_j)) \right| < 2(1 + \varepsilon) \cdot \|\phi(\tilde{x}_i) - \phi(\tilde{x}_j)\|_{A(\xi, \lambda_r^*)^{-1}} \cdot \right.$$
$$\left. \left( \sqrt{\frac{2\log(2n^2V/\delta_r)}{N^{(r)}}} + \sqrt{\xi}\|\theta^*\|_2 \right) \leq 2^{-r}, \forall \tilde{x}_i, \tilde{x}_j \in \mathcal{B}_v^{(r)}, \forall v \in [V], \forall 1 \leq r \leq r^* \right\},$$

*we have*

$$\Pr[\mathcal{G}] \geq 1 - \delta.$$

*Proof of Lemma 1.* Let $\gamma_r := N^{(r)}\xi$. Recall that $\hat{\theta}_r := \left(\gamma_r I + \Phi_r^\top \Phi_r\right)^{-1} \Phi_r^\top \bar{y}^{(r)}$, $\Phi_r := [\sqrt{N_1^{(r)}}\phi(\tilde{x}_1)^\top; \ldots; \sqrt{N_{nV}^{(r)}}\phi(\tilde{x}_{nV})^\top]$ and $\bar{y}^{(r)} := [\sqrt{N_1^{(r)}}\bar{y}_1^{(r)}, \ldots, \sqrt{N_{nV}^{(r)}}\bar{y}_{nV}^{(r)}]^\top$.

Let $\bar{\eta}^{(r)} := [\sqrt{N_1^{(r)}}\bar{\eta}_1^{(r)}, \ldots, \sqrt{N_{nV}^{(r)}}\bar{\eta}_{nV}^{(r)}]^\top$, where $\bar{\eta}_i^{(r)} := \bar{y}_i^{(r)} - \phi(\tilde{x}_i)^\top \theta^*$ denotes the average noise of the $N_i^{(r)}$ samples on arm $\tilde{x}_i$ for any $i \in [nV]$.

Then, for any fixed round $1 \le r \le r^*$, $\tilde{x}_i, \tilde{x}_j \in \tilde{\mathcal{B}}_v^{(r)}$ and $v \in [V]$, we have

$$\left(\hat{F}_r(\tilde{x}_i) - \hat{F}_r(\tilde{x}_j)\right) - (F(\tilde{x}_i) - F(\tilde{x}_j))$$

$$= (\phi(\tilde{x}_i) - \phi(\tilde{x}_j))^\top \left(\hat{\theta}_r - \theta^*\right)$$

$$= (\phi(\tilde{x}_i) - \phi(\tilde{x}_j))^\top \left(\left(\gamma_r I + \Phi_r^\top \Phi_r\right)^{-1} \Phi_r^\top \bar{y}^{(r)} - \theta^*\right)$$

$$= (\phi(\tilde{x}_i) - \phi(\tilde{x}_j))^\top \left(\left(\gamma_r I + \Phi_r^\top \Phi_r\right)^{-1} \Phi_r^\top \left(\Phi_r \theta^* + \bar{\eta}^{(r)}\right) - \theta^*\right)$$

$$= (\phi(\tilde{x}_i) - \phi(\tilde{x}_j))^\top \left(\left(\gamma_r I + \Phi_r^\top \Phi_r\right)^{-1} \Phi_r^\top \Phi_r \theta^* + \left(\gamma_r I + \Phi_r^\top \Phi_r\right)^{-1} \Phi_r^\top \bar{\eta}^{(r)} - \theta^*\right)$$

$$= (\phi(\tilde{x}_i) - \phi(\tilde{x}_j))^\top \left(\left(\gamma_r I + \Phi_r^\top \Phi_r\right)^{-1} \left(\Phi_r^\top \Phi_r + \gamma_r I\right) \theta^* + \left(\gamma_r I + \Phi_r^\top \Phi_r\right)^{-1} \Phi_r^\top \bar{\eta}^{(r)}\right.$$

$$\left. - \theta^* - \gamma_r \left(\gamma_r I + \Phi_r^\top \Phi_r\right)^{-1} \theta^*\right)$$

$$= \underbrace{(\phi(\tilde{x}_i) - \phi(\tilde{x}_j))^\top \left(\gamma_r I + \Phi_r^\top \Phi_r\right)^{-1} \Phi_r^\top \bar{\eta}^{(r)}}_{\text{Term 1}} - \gamma_r (\phi(\tilde{x}_i) - \phi(\tilde{x}_j))^\top \left(\gamma_r I + \Phi_r^\top \Phi_r\right)^{-1} \theta^* \quad (13)$$

In Eq. (13), the expectation of Term 1 is zero, and the variance of Term 1 is bounded by

$$(\phi(\tilde{x}_i) - \phi(\tilde{x}_j))^\top \left(\gamma_r I + \Phi_r^\top \Phi_r\right)^{-1} \Phi_r^\top \Phi_r \left(\gamma_r I + \Phi_r^\top \Phi_r\right)^{-1} (\phi(\tilde{x}_i) - \phi(\tilde{x}_j))$$

$$\le (\phi(\tilde{x}_i) - \phi(\tilde{x}_j))^\top \left(\gamma_r I + \Phi_r^\top \Phi_r\right)^{-1} \left(\gamma_r I + \Phi_r^\top \Phi_r\right) \left(\gamma_r I + \Phi_r^\top \Phi_r\right)^{-1} (\phi(\tilde{x}_i) - \phi(\tilde{x}_j))$$

$$= (\phi(\tilde{x}_i) - \phi(\tilde{x}_j))^\top \left(\gamma_r I + \Phi_r^\top \Phi_r\right)^{-1} (\phi(\tilde{x}_i) - \phi(\tilde{x}_j))$$

$$= \|\phi(\tilde{x}_i) - \phi(\tilde{x}_j)\|_{(\gamma_r I + \Phi_r^\top \Phi_r)^{-1}}^2.$$

Using the Hoeffding inequality, we have that with probability at least $1 - \frac{\delta_r}{n^2 V}$,

$$\left|(\phi(\tilde{x}_i) - \phi(\tilde{x}_j))^\top \left(\gamma_r I + \Phi_r^\top \Phi_r\right)^{-1} \Phi_r^\top \bar{\eta}^{(r)}\right| < \|\phi(\tilde{x}_i) - \phi(\tilde{x}_j)\|_{(\gamma_r I + \Phi_r^\top \Phi_r)^{-1}} \sqrt{2\log\left(\frac{2n^2 V}{\delta_r}\right)}$$

$$(14)$$

Thus, taking the absolute value on both sides of Eq. (13) and using Eq. (14) and the Cauchy–Schwarz inequality, we have that for any fixed round $1 \le r \le r^*$, $\tilde{x}_i, \tilde{x}_j \in \tilde{\mathcal{B}}_v^{(r)}$ and $v \in [V]$, with probability at least $1 - \frac{\delta_r}{n^2 V}$,

$$\left|\left(\hat{F}_r(\tilde{x}_i) - \hat{F}_r(\tilde{x}_j)\right) - (F(\tilde{x}_i) - F(\tilde{x}_j))\right|$$

$$< \|\phi(\tilde{x}_i) - \phi(\tilde{x}_j)\|_{(\gamma_r I + \Phi_r^\top \Phi_r)^{-1}} \sqrt{2\log\left(\frac{2n^2 V}{\delta_r}\right)}$$

$$+ \gamma_r \|\phi(\tilde{x}_i) - \phi(\tilde{x}_j)\|_{(\gamma_r I + \Phi_r^\top \Phi_r)^{-1}} \|\theta^*\|_{(\gamma_r I + \Phi_r^\top \Phi_r)^{-1}}$$

$$\le \|\phi(\tilde{x}_i) - \phi(\tilde{x}_j)\|_{(\gamma_r I + \Phi_r^\top \Phi_r)^{-1}} \sqrt{2\log\left(\frac{2n^2 V}{\delta_r}\right)} + \sqrt{\gamma_r} \cdot \|\phi(\tilde{x}_i) - \phi(\tilde{x}_j)\|_{(\gamma_r I + \Phi_r^\top \Phi_r)^{-1}} \|\theta^*\|_2$$

$$\overset{(a)}{\le} \frac{2(1+\varepsilon) \cdot \|\phi(\tilde{x}_i) - \phi(\tilde{x}_j)\|_{(\xi I + \Phi_{\lambda_r^*}^\top \Phi_{\lambda_r^*})^{-1}}}{\sqrt{N^{(r)}}} \sqrt{2\log\left(\frac{2n^2 V}{\delta_r}\right)}$$

$$+ \sqrt{\xi N^{(r)}} \cdot \frac{2(1+\varepsilon) \cdot \|\phi(\tilde{x}_i) - \phi(\tilde{x}_j)\|_{(\xi I + \Phi_{\lambda_r^*}^\top \Phi_{\lambda_r^*})^{-1}}}{\sqrt{N^{(r)}}} \cdot \|\theta^*\|_2$$

$$=2(1+\varepsilon)\cdot\|\phi(\tilde{x}_i)-\phi(\tilde{x}_j)\|_{(\xi I+\Phi_{\lambda_r^*}^\top\Phi_{\lambda_r^*})^{-1}}\sqrt{\frac{2\log(2n^2V/\delta_r)}{N^{(r)}}}$$

$$+2(1+\varepsilon)\sqrt{\xi}\|\phi(\tilde{x}_i)-\phi(\tilde{x}_j)\|_{(\xi I+\Phi_{\lambda_r^*}^\top\Phi_{\lambda_r^*})^{-1}}\|\theta^*\|_2$$

$$\leq 2(1+\varepsilon)\max_{\tilde{x}_i,\tilde{x}_j\in\tilde{\mathcal{B}}_v^{(r)},v\in[V]}\|\phi(\tilde{x}_i)-\phi(\tilde{x}_j)\|_{(\xi I+\Phi_{\lambda_r^*}^\top\Phi_{\lambda_r^*})^{-1}}\sqrt{\frac{2\log(2n^2V/\delta_r)}{N^{(r)}}}$$

$$+2(1+\varepsilon)\sqrt{\xi}\max_{\tilde{x}_i,\tilde{x}_j\in\tilde{\mathcal{B}}_v^{(r)},v\in[V]}\|\phi(\tilde{x}_i)-\phi(\tilde{x}_j)\|_{(\xi I+\Phi_{\lambda_r^*}^\top\Phi_{\lambda_r^*})^{-1}}\|\theta^*\|_2,$$

where (a) follows from the guarantee of rounding procedure (Eq. (6)) and $\gamma_r := N^{(r)}\xi$.

According to the condition of $\xi$, it holds that

$$2(1+\varepsilon)\sqrt{\xi}\max_{\tilde{x}_i,\tilde{x}_j\in\tilde{\mathcal{B}}_v^{(r)},v\in[V]}\|\phi(\tilde{x}_i)-\phi(\tilde{x}_j)\|_{(\xi I+\Phi_{\lambda_r^*}^\top\Phi_{\lambda_r^*})^{-1}}\|\theta^*\|_2\leq\frac{1}{2^{r+1}}.$$

Thus, with probability at least $1-\frac{\delta_r}{n^2V}$,

$$\left|\left(\hat{F}_r(\tilde{x}_i)-\hat{F}_r(\tilde{x}_j)\right)-(F(\tilde{x}_i)-F(\tilde{x}_j))\right|$$

$$<2(1+\varepsilon)\max_{\tilde{x}_i,\tilde{x}_j\in\tilde{\mathcal{B}}_v^{(r)},v\in[V]}\|\phi(\tilde{x}_i)-\phi(\tilde{x}_j)\|_{(\xi I+\Phi_{\lambda_r^*}^\top\Phi_{\lambda_r^*})^{-1}}\sqrt{\frac{2\log(2n^2V/\delta_r)}{N^{(r)}}}+\frac{1}{2^{r+1}}$$

$$=\sqrt{\frac{8(1+\varepsilon)^2\rho_r^*\log(2n^2V/\delta_r)}{N^{(r)}}}+\frac{1}{2^{r+1}}$$

$$\leq\frac{1}{2^{r+1}}+\frac{1}{2^{r+1}}$$

$$=\frac{1}{2^r}$$

By a union bound over arms $\tilde{x}_i,\tilde{x}_j$, agent $v$ and round $r$, we have that
$$\Pr[\mathcal{G}]\geq 1-\delta.$$

$\square$

For any $1<r\leq r^*$ and $v\in[V]$, let $\mathcal{S}_v^{(r)}=\{\tilde{x}\in\tilde{\mathcal{X}}_v:F(\tilde{x}_{v,*})-F(\tilde{x})\leq 4\cdot 2^{-r}\}$.

**Lemma 2.** *Assume that event $\mathcal{G}$ occurs. Then, for any round $1<r\leq r^*+1$ and agent $v\in[V]$, we have that $\tilde{x}_{v,*}\in\mathcal{B}_v^{(r)}$ and $\mathcal{B}_v^{(r)}\subseteq\mathcal{S}_v^{(r)}$.*

*Proof of Lemma 2.* We prove the first statement by induction.

To begin, for any $v\in[V]$, $\tilde{x}_{v,*}\in\mathcal{B}_v^{(1)}$ trivially holds.

Suppose that $\tilde{x}_{v,*}\in\mathcal{B}_v^{(r)}$ ($1\leq r\leq r^*$) holds for any $v\in[V]$, and there exists some $v'\in[V]$ such that $\tilde{x}_{v'}^*\notin\mathcal{B}_{v'}^{(r+1)}$. According to the elimination rule of algorithm CoKernelFC, we have that these exists some $\tilde{x}'\in\mathcal{B}_v^{(r)}$ such that
$$\hat{F}_r(\tilde{x}')-\hat{F}_r(\tilde{x}_{v'}^*)\geq 2^{-r}.$$
Using Lemma 1, we have
$$F(\tilde{x}')-F(\tilde{x}_{v'}^*)>\hat{F}_r(\tilde{x}')-\hat{F}_r(\tilde{x}_{v'}^*)-2^{-r}\geq 0,$$
which contradicts the definition of $\tilde{x}_{v'}^*$. Thus, we have that for any $v\in[V]$, $\tilde{x}_{v,*}\in\mathcal{B}_v^{(r+1)}$, which completes the proof of the first statement.

Now, we prove the second statement also by induction.

To begin, we prove that for any $v\in[V]$, $\mathcal{B}_v^{(2)}\subseteq\mathcal{S}_v^{(2)}$. Suppose that there exists some $v'\in[V]$ such that $\mathcal{B}_{v'}^{(2)}\subsetneq\mathcal{S}_{v'}^{(2)}$. Then, there exists some $\tilde{x}'\in\mathcal{B}_{v'}^{(2)}$ such that $F(\tilde{x}_{v'}^*)-F(\tilde{x}')>4\cdot 2^{-2}=1$. Using Lemma 1, we have that at the round $r=1$,
$$\hat{F}_r(\tilde{x}_{v'}^*)-\hat{F}_r(\tilde{x}')\geq F(\tilde{x}_{v'}^*)-F(\tilde{x}')-2^{-1}>1-2^{-1}=2^{-1},$$
which implies that $\tilde{x}'$ should have been eliminated in round $r=1$ and gives a contradiction.

Suppose that $\mathcal{B}_v^{(r)} \subseteq \mathcal{S}_v^{(r)}$ ($1 < r \leq r^*$) holds for any $v \in [V]$, and there exists some $v' \in [V]$ such that $\mathcal{B}_{v'}^{(r+1)} \subsetneq \mathcal{S}_{v'}^{(r+1)}$. Then, there exists some $\tilde{x}' \in \mathcal{B}_{v'}^{(r+1)}$ such that $F(\tilde{x}_{v'}^*) - F(\tilde{x}') > 4 \cdot 2^{-(r+1)} = 2 \cdot 2^{-r}$. Using Lemma 1, we have that at the round $r$,

$$\hat{F}_r(\tilde{x}_{v'}^*) - \hat{F}_r(\tilde{x}') \geq F(\tilde{x}_{v'}^*) - F(\tilde{x}') - 2^{-r} > 2 \cdot 2^{-r} - 2^{-r} = 2^{-r},$$

which implies that $\tilde{x}'$ should have been eliminated in round $r$ and gives a contradiction. Thus, we complete the proof of Lemma 2. $\qquad\square$

Now we prove Theorem 1.

*Proof of Theorem 1.* Assume that event $\mathcal{G}$ occurs.

We first prove the correctness.

Recall that $r^* := \lceil \log_2(\frac{2}{\Delta_{\min}}) \rceil + 1$. According to Lemma 2, for any $v \in [V]$ and $\tilde{x} \in \mathcal{B}_v^{(r^*+1)}$, we have $F(\tilde{x}_{v,*}) - F(\tilde{x}) \leq 4 \cdot 2^{-(r^*+1)} = 2 \cdot 2^{-(\lceil \log_2(\frac{2}{\Delta_{\min}}) \rceil + 1)} < 2 \cdot 2^{-\log_2(\frac{2}{\Delta_{\min}})} = \Delta_{\min}$, and thus $\mathcal{B}_v^{(r^*+1)} = \{\tilde{x}_{v,*}\}$. Therefore, algorithm CoKernelFC will return the correct answer $\tilde{x}_{v,*}$ for all $v \in [V]$ and use at most $r^* := \lceil \log_2(\frac{2}{\Delta_{\min}}) \rceil + 1$ rounds.

Next, we prove the sample complexity.

In algorithm CoKernelFC, the computation of $\lambda_r^*$, $\rho_r^*$ and $N^{(r)}$ is the same for all agents, and each agent $v$ just generates partial samples that belong to her arm set $\tilde{\mathcal{X}}_v$ from the total $N^{(r)}$ samples. Thus, to bound the overall sample complexity, it suffices to bound $\sum_{r=1}^{r^*} N^{(r)}$, and then we can obtain the per-agent sample complexity by dividing $V$.

Let $\varepsilon = \frac{1}{10}$. Then, we have

$$\sum_{r=1}^{r^*} N^{(r)}$$

$$\leq \sum_{r=1}^{r^*} \left( 32(2^r)^2(1+\varepsilon)^2 \rho_r^* \log\left(\frac{2n^2 V}{\delta_r}\right) + 1 + \tau(\xi, \lambda_r^*, \varepsilon) \right)$$

$$= \sum_{t=2}^{r^*} 32(2^r)^2 \left(4 \cdot 2^{-r}\right)^2 (1+\varepsilon)^2 \frac{\min_{\lambda \in \triangle_{\tilde{\mathcal{X}}}} \max_{\tilde{x}_i, \tilde{x}_j \in \mathcal{B}_v^{(r)}, v \in [V]} \|\phi(\tilde{x}_i) - \phi(\tilde{x}_j)\|_{A(\xi, \lambda)^{-1}}^2}{\left(4 \cdot 2^{-r}\right)^2} \cdot$$
$$\log\left(\frac{4Vn^2 r^2}{\delta}\right) + N_1 + 2\tau(\xi, \lambda_1^*, \varepsilon) \cdot r^*$$

$$= \sum_{t=2}^{r^*} \left( 512(1+\varepsilon)^2 \frac{\min_{\lambda \in \triangle_{\tilde{\mathcal{X}}}} \max_{\tilde{x}_i, \tilde{x}_j \in \mathcal{B}_v^{(r)}, v \in [V]} \|\phi(\tilde{x}_i) - \phi(\tilde{x}_j)\|_{A(\xi, \lambda)^{-1}}^2}{\left(4 \cdot 2^{-r}\right)^2} \log\left(\frac{4Vn^2 (r^*)^2}{\delta}\right) \right)$$
$$+ N_1 + 2\tau(\xi, \lambda_1^*, \varepsilon) \cdot r^*$$

$$\leq \sum_{t=2}^{r^*} \left( 2048(1+\varepsilon)^2 \frac{\min_{\lambda \in \triangle_{\tilde{\mathcal{X}}}} \max_{\tilde{x} \in \mathcal{B}_v^{(r)} \setminus \{\tilde{x}_{v,*}\}, v \in [V]} \|\phi(\tilde{x}_{v,*}) - \phi(\tilde{x})\|_{A(\xi, \lambda)^{-1}}^2}{\left(4 \cdot 2^{-r}\right)^2} \cdot$$
$$\log\left(\frac{4Vn^2 (r^*)^2}{\delta}\right) \right) + N_1 + 2\tau(\xi, \lambda_1^*, \varepsilon) \cdot r^*$$

$$\leq \sum_{t=2}^{r^*} \left( 2048(1+\varepsilon)^2 \min_{\lambda \in \triangle_{\tilde{\mathcal{X}}}} \max_{\tilde{x} \in \mathcal{B}_v^{(r)} \setminus \{\tilde{x}_{v,*}\}, v \in [V]} \frac{\|\phi(\tilde{x}_{v,*}) - \phi(\tilde{x})\|_{A(\xi, \lambda)^{-1}}^2}{\left(F(\tilde{x}_{v,*}) - F(\tilde{x})\right)^2} \log\left(\frac{4Vn^2 (r^*)^2}{\delta}\right) \right)$$
$$+ N_1 + 2\tau(\xi, \lambda_1^*, \varepsilon) \cdot r^*$$

$$\leq \sum_{t=2}^{r^*} \left( 2048(1+\varepsilon)^2 \min_{\lambda \in \triangle_{\tilde{\mathcal{X}}}} \max_{\tilde{x} \in \tilde{\mathcal{X}}_v \setminus \{\tilde{x}_{v,*}\}, v \in [V]} \frac{\|\phi(\tilde{x}_{v,*}) - \phi(\tilde{x})\|_{A(\xi, \lambda)^{-1}}^2}{\left(F(\tilde{x}_{v,*}) - F(\tilde{x})\right)^2} \log\left(\frac{4Vn^2 (r^*)^2}{\delta}\right) \right)$$
$$+ N_1 + 2\tau(\xi, \lambda_1^*, \varepsilon) \cdot r^*$$

$$\leq r^* \cdot \left( 2048(1+\varepsilon)^2 \cdot \rho^*(\xi) \cdot \log\left( \frac{4Vn^2(r^*)^2}{\delta} \right) \right) + N_1 + 2\tau(\xi, \lambda_1^*, \varepsilon) \cdot r^*$$

$$= O\left( \rho^*(\xi) \cdot \log \Delta_{\min}^{-1} \cdot \left( \log\left( \frac{Vn}{\delta} \right) + \log\log \Delta_{\min}^{-1} \right) + \tilde{d}(\xi, \lambda_1^*) \cdot \log \Delta_{\min}^{-1} \right)$$

Thus, the per-agent sample complexity is bounded by

$$O\left( \frac{\rho^*(\xi)}{V} \cdot \log \Delta_{\min}^{-1} \cdot \left( \log\left( \frac{Vn}{\delta} \right) + \log\log \Delta_{\min}^{-1} \right) + \frac{\tilde{d}(\xi, \lambda_1^*)}{V} \cdot \log \Delta_{\min}^{-1} \right).$$

Since algorithm `CoKernelFC` has at most $r^* := \lceil \log_2(\frac{2}{\Delta_{\min}}) \rceil + 1$ rounds, the number of communication rounds is bounded by $O(\log \Delta_{\min}^{-1})$. $\qquad\square$

## D.2 INTERPRETATION OF THEOREM 1

*Proof of Corollary 1.* Recall that for any $\lambda \in \triangle_{\tilde{\mathcal{X}}}$, $\Phi_\lambda := [\sqrt{\lambda_1}\phi(\tilde{x}_1)^\top; \dots; \sqrt{\lambda_{nV}}\phi(\tilde{x}_{nV})^\top]$ and $K_\lambda := [\sqrt{\lambda_i \lambda_j} k(\tilde{x}_i, \tilde{x}_j)]_{i,j\in[nV]} = \Phi_\lambda \Phi_\lambda^\top$. Let $\lambda^* := \text{argmax}_{\lambda \in \triangle_{\tilde{\mathcal{X}}}} \log\det\left( I + \xi^{-1}K_\lambda \right) = \text{argmax}_{\lambda \in \triangle_{\tilde{\mathcal{X}}}} \log\det\left( I + \xi^{-1}\Phi_\lambda \Phi_\lambda^\top \right)$. For any $\xi \geq 0$ and $\lambda \in \triangle_{\tilde{\mathcal{X}}}$, $A(\xi, \lambda) := \xi I + \sum_{\tilde{x}\in\tilde{\mathcal{X}}} \lambda_{\tilde{x}} \phi(\tilde{x})\phi(\tilde{x})^\top = \xi I + \Phi_\lambda^\top \Phi_\lambda$.

Then, we have

$$\rho^*(\xi) = \min_{\lambda\in\triangle_{\tilde{\mathcal{X}}}} \max_{\tilde{x}\in\tilde{\mathcal{X}}_v\setminus\{\tilde{x}_{v,*}\}, v\in[V]} \frac{\|\phi(\tilde{x}_{v,*}) - \phi(\tilde{x})\|^2_{(\xi I + \sum_{\tilde{x}'\in\tilde{\mathcal{X}}} \lambda_{\tilde{x}'}\phi(\tilde{x}')\phi(\tilde{x}')^\top)^{-1}}}{(F(\tilde{x}_{v,*}) - F(\tilde{x}))^2}$$

$$\leq \min_{\lambda\in\triangle_{\tilde{\mathcal{X}}}} \max_{\tilde{x}\in\tilde{\mathcal{X}}_v, v\in[V]} \frac{\|\phi(\tilde{x}_{v,*}) - \phi(\tilde{x})\|^2_{(\xi I + \sum_{\tilde{x}'\in\tilde{\mathcal{X}}} \lambda_{\tilde{x}'}\phi(\tilde{x}')\phi(\tilde{x}')^\top)^{-1}}}{\Delta_{\min}^2}$$

$$= \frac{1}{\Delta_{\min}^2} \cdot \min_{\lambda\in\triangle_{\tilde{\mathcal{X}}}} \max_{\tilde{x}\in\tilde{\mathcal{X}}_v, v\in[V]} \|\phi(\tilde{x}_{v,*}) - \phi(\tilde{x})\|^2_{(\xi I + \sum_{\tilde{x}'\in\tilde{\mathcal{X}}} \lambda_{\tilde{x}'}\phi(\tilde{x}')\phi(\tilde{x}')^\top)^{-1}}$$

$$\leq \frac{1}{\Delta_{\min}^2} \cdot \min_{\lambda\in\triangle_{\tilde{\mathcal{X}}}} \left( 2 \max_{\tilde{x}\in\tilde{\mathcal{X}}} \|\phi(\tilde{x})\|_{(\xi I + \sum_{\tilde{x}'\in\tilde{\mathcal{X}}} \lambda_{\tilde{x}'}\phi(\tilde{x}')\phi(\tilde{x}')^\top)^{-1}} \right)^2$$

$$= \frac{4}{\Delta_{\min}^2} \cdot \min_{\lambda\in\triangle_{\tilde{\mathcal{X}}}} \max_{\tilde{x}\in\tilde{\mathcal{X}}} \|\phi(\tilde{x})\|^2_{(\xi I + \sum_{\tilde{x}'\in\tilde{\mathcal{X}}} \lambda_{\tilde{x}'}\phi(\tilde{x}')\phi(\tilde{x}')^\top)^{-1}}$$

$$\leq \frac{4}{\Delta_{\min}^2} \cdot \max_{\tilde{x}\in\tilde{\mathcal{X}}} \|\phi(\tilde{x})\|^2_{(\xi I + \sum_{\tilde{x}'\in\tilde{\mathcal{X}}} \lambda_{\tilde{x}'}^*\phi(\tilde{x}')\phi(\tilde{x}')^\top)^{-1}}$$

$$\overset{(b)}{=} \frac{4}{\Delta_{\min}^2} \cdot \sum_{\tilde{x}\in\tilde{\mathcal{X}}} \lambda_{\tilde{x}}^* \|\phi(\tilde{x})\|^2_{(\xi I + \sum_{\tilde{x}'\in\tilde{\mathcal{X}}} \lambda_{\tilde{x}'}^*\phi(\tilde{x}')\phi(\tilde{x}')^\top)^{-1}},$$

where (b) is due to Lemma 3.

Since $\lambda_{\tilde{x}}^* \|\phi(\tilde{x})\|^2_{(\xi I + \sum_{\tilde{x}'\in\tilde{\mathcal{X}}} \lambda_{\tilde{x}'}^*\phi(\tilde{x}')\phi(\tilde{x}')^\top)^{-1}} \leq 1$ for any $\tilde{x} \in \tilde{\mathcal{X}}$,

$$\sum_{\tilde{x}\in\tilde{\mathcal{X}}} \lambda_{\tilde{x}}^* \|\phi(\tilde{x})\|^2_{(\xi I + \sum_{\tilde{x}'\in\tilde{\mathcal{X}}} \lambda_{\tilde{x}'}^*\phi(\tilde{x}')\phi(\tilde{x}')^\top)^{-1}}$$

$$\leq 2 \sum_{\tilde{x}\in\tilde{\mathcal{X}}} \log\left( 1 + \lambda_{\tilde{x}}^* \|\phi(\tilde{x})\|^2_{(\xi I + \sum_{\tilde{x}'\in\tilde{\mathcal{X}}} \lambda_{\tilde{x}'}^*\phi(\tilde{x}')\phi(\tilde{x}')^\top)^{-1}} \right)$$

$$\overset{(c)}{\leq} 2 \log \frac{\det\left( \xi I + \sum_{\tilde{x}\in\tilde{\mathcal{X}}} \lambda_{\tilde{x}}^*\phi(\tilde{x})\phi(\tilde{x})^\top \right)}{\det\left( \xi I \right)}$$

$$= 2 \log\det\left( I + \xi^{-1} \sum_{\tilde{x}\in\tilde{\mathcal{X}}} \lambda_{\tilde{x}}^*\phi(\tilde{x})\phi(\tilde{x})^\top \right)$$

$$= 2 \log\det\left( I + \xi^{-1} \Phi_\lambda^\top \Phi_\lambda \right)$$

$$= 2 \log\det\left( I + \xi^{-1} \Phi_\lambda \Phi_\lambda^\top \right)$$

$$=2\log\det\left(I+\xi^{-1}K_{\lambda^*}\right),$$

where (c) comes from Lemma 4.

Thus, we have

$$\rho^*(\xi) \leq \frac{8}{\Delta_{\min}^2} \cdot \log\det\left(I+\xi^{-1}K_{\lambda^*}\right). \tag{15}$$

In the following, we interpret the term $\log\det\left(I+\xi^{-1}K_{\lambda^*}\right)$ by maximum information gain and effective dimension, and then decompose it into components from task similarities and arm features.

**Maximum Information Gain.** Recall that the maximum information is defined as

$$\Upsilon = \max_{\lambda\in\triangle_{\tilde{\mathcal{X}}}} \log\det\left(I+\xi^{-1}K_{\lambda}\right) = \log\det\left(I+\xi^{-1}K_{\lambda^*}\right).$$

Then, using Eq. (15), we have

$$\rho^*(\xi) \leq \frac{8}{\Delta_{\min}^2} \cdot \Upsilon. \tag{16}$$

Combining the bound in Theorem 1 and Eq. (16), the per-agent sample complexity is bounded by

$$S = O\left(\frac{\rho^*(\xi)}{V} \cdot \log\Delta_{\min}^{-1}\left(\log\left(\frac{Vn}{\delta}\right) + \log\log\Delta_{\min}^{-1}\right) + \frac{\tilde{d}(\xi,\lambda_1^*)}{V} \cdot \log\Delta_{\min}^{-1}\right)$$

$$= O\left(\frac{\Upsilon}{\Delta_{\min}^2 V} \cdot \log\Delta_{\min}^{-1}\left(\log\left(\frac{Vn}{\delta}\right) + \log\log\Delta_{\min}^{-1}\right) + \frac{\tilde{d}(\xi,\lambda_1^*)}{V} \cdot \log\Delta_{\min}^{-1}\right)$$

**Effective Dimension.** Recall that $\alpha_1 \geq \cdots \geq \alpha_{nV}$ denote the eigenvalues of $K_{\lambda^*}$ in decreasing order, and the effective dimension is defined as

$$d_{\text{eff}} = \min\left\{j : j\xi\log(nV) \geq \sum_{i=j+1}^{nV}\alpha_i\right\}.$$

Then, it holds that $d_{\text{eff}}\xi\log(nV) \geq \sum_{i=d_{\text{eff}}+1}^{nV}\alpha_i$.

Let $\varepsilon = d_{\text{eff}}\xi\log(nV) - \sum_{i=d_{\text{eff}}+1}^{nV}\alpha_i$, and thus $\varepsilon \leq d_{\text{eff}}\xi\log(nV)$. Then, we have $\sum_{i=1}^{d_{\text{eff}}}\alpha_i = \texttt{Trace}(K_{\lambda^*}) - \sum_{i=d_{\text{eff}}+1}^{nV}\alpha_i = \texttt{Trace}(K_{\lambda^*}) - d_{\text{eff}}\xi\log(nV) + \varepsilon$ and $\sum_{i=d_{\text{eff}}+1}^{nV}\alpha_i = d_{\text{eff}}\xi\log(nV) - \varepsilon$.

$$\log\det\left(I+\xi^{-1}K_{\lambda^*}\right)$$

$$=\log\left(\Pi_{i=1}^{nV}\left(1+\xi^{-1}\alpha_i\right)\right)$$

$$=\log\left(\Pi_{i=1}^{d_{\text{eff}}}\left(1+\xi^{-1}\alpha_i\right) \cdot \Pi_{i=d_{\text{eff}}+1}^{nV}\left(1+\xi^{-1}\alpha_i\right)\right)$$

$$\leq\log\left(\left(1+\xi^{-1}\cdot\frac{\texttt{Trace}(K_{\lambda^*})-d_{\text{eff}}\xi\log(nV)+\varepsilon}{d_{\text{eff}}}\right)^{d_{\text{eff}}}\left(1+\xi^{-1}\cdot\frac{d_{\text{eff}}\xi\log(nV)-\varepsilon}{nV-d_{\text{eff}}}\right)^{nV-d_{\text{eff}}}\right)$$

$$\leq d_{\text{eff}}\log\left(1+\xi^{-1}\cdot\frac{\texttt{Trace}(K_{\lambda^*})-d_{\text{eff}}\xi\log(nV)+\varepsilon}{d_{\text{eff}}}\right) + \log\left(1+\frac{d_{\text{eff}}\log(nV)}{nV-d_{\text{eff}}}\right)^{nV-d_{\text{eff}}}$$

$$=d_{\text{eff}}\log\left(1+\xi^{-1}\cdot\frac{\texttt{Trace}(K_{\lambda^*})-d_{\text{eff}}\xi\log(nV)+\varepsilon}{d_{\text{eff}}}\right)$$

$$\quad +\log\left(1+\frac{d_{\text{eff}}\log(nV-d_{\text{eff}}+d_{\text{eff}})}{nV-d_{\text{eff}}}\right)^{nV-d_{\text{eff}}}$$

$$\overset{(d)}{\leq} d_{\text{eff}}\log\left(1+\xi^{-1}\cdot\frac{\texttt{Trace}(K_{\lambda^*})-d_{\text{eff}}\xi\log(nV)+\varepsilon}{d_{\text{eff}}}\right) + \log\left(1+\frac{d_{\text{eff}}\log(nV+d_{\text{eff}})}{nV}\right)^{nV}$$

$$=d_{\text{eff}}\log\left(1+\xi^{-1}\cdot\frac{\texttt{Trace}(K_{\lambda^*})-d_{\text{eff}}\xi\log(nV)+\varepsilon}{d_{\text{eff}}}\right) + nV\log\left(1+\frac{d_{\text{eff}}\log(nV+d_{\text{eff}})}{nV}\right)$$

$$\leq d_{\text{eff}}\log\left(1+\frac{\texttt{Trace}(K_{\lambda^*})}{\xi d_{\text{eff}}}\right) + d_{\text{eff}}\log(nV+d_{\text{eff}})$$

$$\leq d_{\text{eff}} \log \left( 2nV \cdot \left( 1 + \frac{\texttt{Trace}(K_{\lambda^*})}{\xi d_{\text{eff}}} \right) \right),$$

where inequality (d) is due to that $\left( 1 + \frac{d_{\text{eff}} \log(x + d_{\text{eff}})}{x} \right)^x$ is monotonically increasing with respect to $x \geq 1$.

Then, using Eq. (15), we have

$$\rho^*(\xi) \leq \frac{8}{\Delta_{\min}^2} \cdot d_{\text{eff}} \log \left( 2nV \cdot \left( 1 + \frac{\texttt{Trace}(K_{\lambda^*})}{\xi d_{\text{eff}}} \right) \right). \tag{17}$$

Combining the bound in Theorem 1 and Eq. (17), the per-agent sample complexity is bounded by

$$
\begin{aligned}
S =& O \left( \frac{\rho^*(\xi)}{V} \cdot \log \Delta_{\min}^{-1} \left( \log \left( \frac{Vn}{\delta} \right) + \log \log \Delta_{\min}^{-1} \right) + \frac{\tilde{d}(\xi, \lambda_1^*)}{V} \cdot \log \Delta_{\min}^{-1} \right) \\
=& O \left( \frac{d_{\text{eff}}}{\Delta_{\min}^2 V} \cdot \log \left( nV \cdot \left( 1 + \frac{\texttt{Trace}(K_{\lambda^*})}{\xi d_{\text{eff}}} \right) \right) \cdot \log \Delta_{\min}^{-1} \left( \log \left( \frac{Vn}{\delta} \right) + \log \log \Delta_{\min}^{-1} \right) \right. \\
& \left. + \frac{\tilde{d}(\xi, \lambda_1^*)}{V} \cdot \log \Delta_{\min}^{-1} \right)
\end{aligned}
$$

**Decomposition.** Recall that $K_{\lambda^*} = [\sqrt{\lambda_i^* \lambda_j^*} k(\tilde{x}_i, \tilde{x}_j)]_{i,j \in [nV]}$, $K_{\mathcal{Z}} = [k_{\mathcal{Z}}(z_v, z_{v'})]_{v,v' \in [V]}$ and $K_{\mathcal{X}, \lambda^*} = [\sqrt{\lambda_i^* \lambda_j^*} k_{\mathcal{X}}(\tilde{x}_i, \tilde{x}_j)]_{i,j \in [nV]}$. Let $\tilde{K}_{\mathcal{Z}} = [k_{\mathcal{Z}}(z_{v_i}, z_{v_j})]_{i,j \in [nV]}$, where for any $i \in [nV]$, $v_i$ denotes the index of the task for the $i$-th arm $\tilde{x}_i$ in the arm set $\tilde{\mathcal{X}}$ and $z_{v_i}$ denotes its task feature. It holds that $\texttt{rank}(\tilde{K}_{\mathcal{Z}}) = \texttt{rank}(K_{\mathcal{Z}})$.

Since $K_{\lambda^*}$ is a Hadamard composition of $\tilde{K}_{\mathcal{Z}}$ and $K_{\mathcal{X}, \lambda^*}$, we have that $\texttt{rank}(K_{\lambda^*}) \leq \texttt{rank}(\tilde{K}_{\mathcal{Z}}) \cdot \texttt{rank}(K_{\mathcal{X}, \lambda^*}) = \texttt{rank}(K_{\mathcal{Z}}) \cdot \texttt{rank}(K_{\mathcal{X}, \lambda^*})$.

$$
\begin{aligned}
& \log \det \left( I + \xi^{-1} K_{\lambda^*} \right) \\
=& \log \left( \Pi_{i=1}^{nV} \left( 1 + \xi^{-1} \alpha_i \right) \right) \\
=& \log \left( \Pi_{i=1}^{\texttt{rank}(K_{\lambda^*})} \left( 1 + \xi^{-1} \alpha_i \right) \right) \\
\leq& \log \left( \frac{\sum_{i=1}^{\texttt{rank}(K_{\lambda^*})} \left( 1 + \xi^{-1} \alpha_i \right)}{\texttt{rank}(K_{\lambda^*})} \right)^{\texttt{rank}(K_{\lambda^*})} \\
=& \texttt{rank}(K_{\lambda^*}) \log \left( \frac{\sum_{i=1}^{\texttt{rank}(K_{\lambda^*})} \left( 1 + \xi^{-1} \alpha_i \right)}{\texttt{rank}(K_{\lambda^*})} \right) \\
\leq& \texttt{rank}(K_{\mathcal{Z}}) \cdot \texttt{rank}(K_{\mathcal{X}, \lambda^*}) \log \left( \frac{\texttt{Trace} \left( I + \xi^{-1} K_{\lambda^*} \right)}{\texttt{rank}(K_{\lambda^*})} \right)
\end{aligned}
$$

Then, using Eq. (15), we have

$$\rho^*(\xi) \leq \frac{8}{\Delta_{\min}^2} \cdot \texttt{rank}(K_{\mathcal{Z}}) \cdot \texttt{rank}(K_{\mathcal{X}, \lambda^*}) \log \left( \frac{\texttt{Trace} \left( I + \xi^{-1} K_{\lambda^*} \right)}{\texttt{rank}(K_{\lambda^*})} \right). \tag{18}$$

Combining the bound in Theorem 1 and Eq. (18), the per-agent sample complexity is bounded by

$$
\begin{aligned}
& O \left( \frac{\rho^*(\xi)}{V} \cdot \log \Delta_{\min}^{-1} \left( \log \left( \frac{Vn}{\delta} \right) + \log \log \Delta_{\min}^{-1} \right) + \frac{\tilde{d}(\xi, \lambda_1^*)}{V} \cdot \log \Delta_{\min}^{-1} \right) \\
=& O \left( \frac{\texttt{rank}(K_{\mathcal{Z}}) \cdot \texttt{rank}(K_{\mathcal{X}, \lambda^*})}{\Delta_{\min}^2 V} \cdot \log \left( \frac{\texttt{Trace} \left( I + \xi^{-1} K_{\lambda^*} \right)}{\texttt{rank}(K_{\lambda^*})} \right) \right).
\end{aligned}
$$

$$\log \Delta_{\min}^{-1} \left( \log \left( \frac{Vn}{\delta} \right) + \log \log \Delta_{\min}^{-1} \right) + \frac{\tilde{d}(\xi, \lambda_1^*)}{V} \cdot \log \Delta_{\min}^{-1} \right)$$

Hence, we complete the proof of Corollary 1. □

# E  PROOFS FOR THE FIXED-BUDGET SETTING

## E.1  PROOF OF THEOREM 2

*Proof of Theorem 2.* Our proof of Theorem 2 adapts the error probability analysis in (Katz-Samuels et al., 2020) to the multi-agent setting.

Since the number of samples used over all agents in each round is $N = \lfloor TV/R \rfloor$, the total number of samples used by algorithm CoKernelFB is at most $TV$ and the total number of samples used per agent is at most $T$.

Now we prove the error probability upper bound.

Recall that for any $\xi \geq 0$ and $\lambda \in \triangle_{\tilde{\mathcal{X}}}$, $A(\xi, \lambda) := \xi I + \sum_{\tilde{x} \in \tilde{\mathcal{X}}} \lambda_{\tilde{x}} \phi(\tilde{x}) \phi(\tilde{x})^\top = \xi I + \Phi_\lambda^\top \Phi_\lambda$. For any $\lambda \in \triangle_{\tilde{\mathcal{X}}}$, $\Phi_\lambda = [\sqrt{\lambda_1} \phi(\tilde{x}_1)^\top; \ldots; \sqrt{\lambda_{nV}} \phi(\tilde{x}_{nV})^\top]$. The regularization parameter $\xi$ in algorithm CoKernelFB satisfies $\xi \leq \frac{1}{16(1+\varepsilon)^2 (\rho^*(\xi))^2 \|\theta^*\|^2}$.

For any $r \in [R]$, $\tilde{x}_i, \tilde{x}_j \in \mathcal{B}_v^{(r)}$ and $v \in [V]$, define reward gap

$$\Delta_{r, \tilde{x}_i, \tilde{x}_j} = \inf_{\Delta > 0} \left\{ \frac{\|\phi(\tilde{x}_i) - \phi(\tilde{x}_j)\|_{(\xi I + \Phi_{\lambda_r^*}^\top \Phi_{\lambda_r^*})^{-1}}^2}{\Delta^2} \leq 8 \rho^*(\xi) \right\},$$

and event

$$\mathcal{J}_{r, \tilde{x}_i, \tilde{x}_j} = \left\{ \left| \left( \hat{F}_r(\tilde{x}_i) - \hat{F}_r(\tilde{x}_j) \right) - (F(\tilde{x}_i) - F(\tilde{x}_j)) \right| < \Delta_{r, \tilde{x}_i, \tilde{x}_j} \right\}.$$

In the following, we prove $\Pr\left[ \neg \mathcal{J}_{r, \tilde{x}_i, \tilde{x}_j} \right] \leq 2 \exp\left( -\frac{N}{32(1+\varepsilon)\rho^*(\xi)} \right).$

Similar to the analysis for Eq. (13) in the proof of Lemma 1, we have that for any $r \in [R]$, $\tilde{x}_i, \tilde{x}_j \in \mathcal{B}_v^{(r)}$ and $v \in [V]$,

$$\left( \hat{F}_r(\tilde{x}_i) - \hat{F}_r(\tilde{x}_j) \right) - (F(\tilde{x}_i) - F(\tilde{x}_j))$$
$$= \underbrace{(\phi(\tilde{x}_i) - \phi(\tilde{x}_j))^\top \left( N\xi I + \Phi_r^\top \Phi_r \right)^{-1} \Phi_r^\top \bar{\eta}^{(r)}}_{\text{Term 1}} - \underbrace{\xi N \left( \phi(\tilde{x}_i) - \phi(\tilde{x}_j) \right)^\top \left( N\xi I + \Phi_r^\top \Phi_r \right)^{-1} \theta^*}_{\text{Term 2}}.$$

$$\tag{19}$$

Here, the expectation of Term 1 in Eq. (19) is zero, and the variance of Term 1 is bounded by

$$\|\phi(\tilde{x}_i) - \phi(\tilde{x}_j)\|_{(N\xi I + \Phi_r^\top \Phi_r)^{-1}}^2 \overset{(a)}{\leq} \frac{2(1+\varepsilon) \cdot \|\phi(\tilde{x}_i) - \phi(\tilde{x}_j)\|_{(\xi I + \Phi_{\lambda_r^*}^\top \Phi_{\lambda_r^*})^{-1}}^2}{N},$$

where (a) comes from the guarantee of rounding procedure (Eq. (6)).

Using the Hoeffding inequality, we have

$$\Pr\left[ \left| (\phi(\tilde{x}_i) - \phi(\tilde{x}_j))^\top \left( N\xi I + \Phi_r^\top \Phi_r \right)^{-1} \Phi_r^\top \bar{\eta}^{(r)} \right| \geq \frac{1}{2} \Delta_{r, \tilde{x}_i, \tilde{x}_j} \right]$$

$$\leq 2 \exp\left( -2 \cdot \frac{\frac{1}{4} \Delta_{r, \tilde{x}_i, \tilde{x}_j}^2}{\frac{2(1+\varepsilon) \cdot \|\phi(\tilde{x}_i) - \phi(\tilde{x}_j)\|_{(\xi I + \Phi_{\lambda_r^*}^\top \Phi_{\lambda_r^*})^{-1}}^2}{N}} \right)$$

$$\leq 2 \exp\left( -\frac{1}{2} \cdot \frac{N}{\frac{2(1+\varepsilon) \cdot \|\phi(\tilde{x}_i) - \phi(\tilde{x}_j)\|_{(\xi I + \Phi_{\lambda_r^*}^\top \Phi_{\lambda_r^*})^{-1}}^2}{\Delta_{r, \tilde{x}_i, \tilde{x}_j}^2}} \right)$$

$$\leq 2 \exp\left(-\frac{N}{32(1+\varepsilon)\rho^*(\xi)}\right)$$

Thus, with probability at least $1 - 2\exp\left(-\frac{N}{32(1+\varepsilon)\rho^*(\xi)}\right)$, we have

$$\left|(\phi(\tilde{x}_i) - \phi(\tilde{x}_j))^\top \left(N\xi I + \Phi_r^\top \Phi_r\right)^{-1} \Phi_r^\top \eta_v^{(r)}\right| < \frac{1}{2}\Delta_{r,\tilde{x}_i,\tilde{x}_j}. \tag{20}$$

Since $\xi$ satisfies $\xi \leq \frac{1}{16(1+\varepsilon)^2(\rho^*(\xi))^2\|\theta^*\|^2}$, we have $4(1+\varepsilon)\sqrt{\xi}\rho^*(\xi)\|\theta^*\| \leq 1$. Then, for any $r \in [R]$, $\tilde{x}_i, \tilde{x}_j \in \mathcal{B}_v^{(r)}$ and $v \in [V]$, we have

$$4(1+\varepsilon)\sqrt{\xi} \cdot \frac{\|\phi(\tilde{x}_i) - \phi(\tilde{x}_j)\|_{(\xi I + \Phi_{\lambda_r^*}^\top \Phi_{\lambda_r^*})^{-1}}}{\Delta_{r,\tilde{x}_i,\tilde{x}_j}} \cdot \|\theta^*\| \leq 1,$$

and thus,

$$2(1+\varepsilon)\sqrt{\xi}\|\phi(\tilde{x}_i) - \phi(\tilde{x}_j)\|_{(\xi I + \Phi_{\lambda_r^*}^\top \Phi_{\lambda_r^*})^{-1}}\|\theta^*\| \leq \frac{1}{2}\Delta_{r,\tilde{x}_i,\tilde{x}_j}.$$

Then, we can bound the bias term (Term 2 in Eq . (19)) as

$$\left|\xi N \left(\phi(\tilde{x}_i) - \phi(\tilde{x}_j)\right)^\top \left(N\xi I + \Phi_r^\top \Phi_r\right)^{-1} \theta^*\right|$$

$$\leq \xi N \|\phi(\tilde{x}_i) - \phi(\tilde{x}_j)\|_{(N\xi I + \Phi_r^\top \Phi_r)^{-1}} \|\theta^*\|_{(N\xi I + \Phi_r^\top \Phi_r)^{-1}}$$

$$\leq \sqrt{\xi N} \cdot \|\phi(\tilde{x}_i) - \phi(\tilde{x}_j)\|_{(N\xi I + \Phi_r^\top \Phi_r)^{-1}} \|\theta^*\|_2$$

$$\leq \sqrt{\xi N} \cdot \frac{2(1+\varepsilon) \cdot \|\phi(\tilde{x}_i) - \phi(\tilde{x}_j)\|_{(\xi I + \Phi_{\lambda_r^*}^\top \Phi_{\lambda_r^*})^{-1}}}{\sqrt{N}} \cdot \|\theta^*\|_2$$

$$= 2(1+\varepsilon)\sqrt{\xi}\|\phi(\tilde{x}_i) - \phi(\tilde{x}_j)\|_{(\xi I + \Phi_{\lambda_r^*}^\top \Phi_{\lambda_r^*})^{-1}}\|\theta^*\|_2$$

$$\leq \frac{1}{2}\Delta_{r,\tilde{x}_i,\tilde{x}_j} \tag{21}$$

Plugging Eqs. (20) and (21) into Eq. (19), we have that with probability at least $1 - 2\exp\left(-\frac{N}{32(1+\varepsilon)\rho^*(\xi)}\right)$, for any $r \in [R]$, $\tilde{x}_i, \tilde{x}_j \in \mathcal{B}_v^{(r)}$ and $v \in [V]$,

$$\left|\left(\hat{F}_r(\tilde{x}_i) - \hat{F}_r(\tilde{x}_j)\right) - (F(\tilde{x}_i) - F(\tilde{x}_j))\right|$$

$$\leq \left|(\phi(\tilde{x}_i) - \phi(\tilde{x}_j))^\top \left(N\xi I + \Phi_r^\top \Phi_r\right)^{-1} \Phi_r^\top \eta_v^{(r)}\right| + \left|\xi N \left(\phi(\tilde{x}_i) - \phi(\tilde{x}_j)\right)^\top \left(N\xi I + \Phi_r^\top \Phi_r\right)^{-1} \theta^*\right|$$

$$< \Delta_{r,\tilde{x}_i,\tilde{x}_j},$$

which completes the proof of $\Pr\left[\neg \mathcal{J}_{r,\tilde{x}_i,\tilde{x}_j}\right] \leq 2\exp\left(-\frac{N}{32(1+\varepsilon)\rho^*(\xi)}\right)$.

Define event

$$\mathcal{J} = \left\{\left|\left(\hat{F}_r(\tilde{x}_i) - \hat{F}_r(\tilde{x}_j)\right) - (F(\tilde{x}_i) - F(\tilde{x}_j))\right| < \Delta_{r,\tilde{x}_i,\tilde{x}_j}, \ \forall \tilde{x}_i, \tilde{x}_j \in \mathcal{B}_v^{(r)}, \forall v \in [V], \forall r \in [R]\right\},$$

Let $\varepsilon = \frac{1}{10}$. By a union bound over $\tilde{x}_i, \tilde{x}_j \in \mathcal{B}_v^{(r)}$, $v \in [V]$ and $r \in [R]$, we have

$$\Pr\left[\neg \mathcal{J}\right] \leq 2n^2 V \log(\omega(\xi, \tilde{\mathcal{X}})) \cdot \exp\left(-\frac{N}{32(1+\varepsilon)\rho^*(\xi)}\right)$$

$$= O\left(n^2 V \log(\omega(\xi, \tilde{\mathcal{X}})) \cdot \exp\left(-\frac{TV}{\rho^*(\xi) \cdot \log(\omega(\xi, \tilde{\mathcal{X}}))}\right)\right)$$

In order to prove Theorem 2, it suffices to prove that conditioning on event $\mathcal{J}$, algorithm CoKernelFB returns the correct answers $\tilde{x}_{v,*}$ for all $v \in [V]$.

Suppose that there exist some $r \in [R]$ and some $v \in [V]$ such that $\tilde{x}_{v,*}$ is eliminated in round $r$. Define

$$\mathcal{B}_v'^{(r)} = \{\tilde{x} \in \mathcal{B}_v^{(r)} : \hat{F}_r(\tilde{x}) > \hat{F}_r(\tilde{x}_{v,*})\},$$

which denotes the set of arms in $\mathcal{B}_v^{(r)}$ which are ranked before $\tilde{x}_{v,*}$ according to the estimated rewards. According to the elimination rule (Line 12 in Algorithm 2), we have

$$\omega(\xi, \mathcal{B}_v'^{(r)} \cup \{\tilde{x}_{v,*}\}) > \frac{1}{2}\omega(\xi, \mathcal{B}_v^{(r)}) = \frac{1}{2}\max_{\tilde{x}_i, \tilde{x}_j \in \mathcal{B}_v^{(r)}} \|\phi(\tilde{x}_i) - \phi(\tilde{x}_j)\|_{(\xi I + \Phi_{\lambda_r^*}^\top \Phi_{\lambda_r^*})^{-1}} \quad (22)$$

Define $\tilde{x}_0 = \mathrm{argmax}_{\tilde{x} \in \mathcal{B}_v'^{(r)}} \Delta_{\tilde{x}}$. We have

$$\frac{1}{2}\frac{\|\phi(\tilde{x}_{v,*}) - \phi(\tilde{x}_0)\|^2_{(\xi I + \Phi_{\lambda_r^*}^\top \Phi_{\lambda_r^*})^{-1}}}{\Delta_{\tilde{x}_0}^2}$$

$$\le \frac{1}{2}\max_{\tilde{x}_i, \tilde{x}_j \in \mathcal{B}_v^{(r)}} \frac{\|\phi(\tilde{x}_i) - \phi(\tilde{x}_j)\|^2_{(\xi I + \Phi_{\lambda_r^*}^\top \Phi_{\lambda_r^*})^{-1}}}{\Delta_{\tilde{x}_0}^2}$$

$$\overset{(a)}{<} \frac{\omega(\xi, \mathcal{B}_v'^{(r)} \cup \{\tilde{x}_{v,*}\})}{\Delta_{\tilde{x}_0}^2}$$

$$\overset{(b)}{=} \min_{\lambda \in \triangle \tilde{\mathcal{X}}} \max_{\tilde{x}_i, \tilde{x}_j \in \mathcal{B}_v'^{(r)} \cup \{\tilde{x}_{v,*}\}} \frac{\|\phi(\tilde{x}_i) - \phi(\tilde{x}_j)\|^2_{(\xi I + \Phi_\lambda^\top \Phi_\lambda)^{-1}}}{\Delta_{\tilde{x}_0}^2}$$

$$\overset{(c)}{\le} 4 \min_{\lambda \in \triangle \tilde{\mathcal{X}}} \max_{\tilde{x} \in \mathcal{B}_v'^{(r)}} \frac{\|\phi(\tilde{x}_{v,*}) - \phi(\tilde{x})\|^2_{(\xi I + \Phi_\lambda^\top \Phi_\lambda)^{-1}}}{\Delta_{\tilde{x}}^2}$$

$$\le 4 \min_{\lambda \in \triangle \tilde{\mathcal{X}}} \max_{\tilde{x} \in \tilde{\mathcal{X}}_v \setminus \{\tilde{x}_{v,*}\}, v \in [V]} \frac{\|\phi(\tilde{x}_{v,*}) - \phi(\tilde{x})\|^2_{(\xi I + \Phi_\lambda^\top \Phi_\lambda)^{-1}}}{\Delta_{\tilde{x}}^2}$$

$$= 4\rho^*(\xi),$$

where (a) follows from Eq. (22), (b) comes from the definition of $\omega(\cdot, \cdot)$, and (c) is due to the definition of $\tilde{x}_0$.

According to the definition

$$\Delta_{r, \tilde{x}_{v,*}, \tilde{x}_0} = \inf_{\Delta > 0} \left\{ \frac{\|\phi(\tilde{x}_{v,*}) - \phi(\tilde{x}_0)\|^2_{(\xi I + \Phi_{\lambda_r^*}^\top \Phi_{\lambda_r^*})^{-1}}}{\Delta^2} \le 8\rho^*(\xi) \right\},$$

we have $\Delta_{r, \tilde{x}_{v,*}, \tilde{x}_0} \le \Delta_{\tilde{x}_0}$.

Conditioning on $\mathcal{J}$, we have $\left|\left(\hat{F}_r(\tilde{x}_{v,*}) - \hat{F}_r(\tilde{x}_0)\right) - (F(\tilde{x}_{v,*}) - F(\tilde{x}_0))\right| < \Delta_{r, \tilde{x}_{v,*}, \tilde{x}_0} \le \Delta_{\tilde{x}_0}$. Then, we have

$$\hat{F}_r(\tilde{x}_{v,*}) - \hat{F}_r(\tilde{x}_0) > (F(\tilde{x}_{v,*}) - F(\tilde{x}_0)) - \Delta_{\tilde{x}_0} = 0,$$

which contradicts the definition of $\tilde{x}_0$ that satisfies $\hat{F}_r(\tilde{x}_0) > \hat{F}_r(\tilde{x}_{v,*})$. Thus, for any round $r \in [R]$ and task $v \in [V]$, $\tilde{x}_{v,*}$ will never be eliminated.

Since $\omega(\xi, \{\tilde{x}_{v,*}, \tilde{x}\}) \ge 1$ for any $\tilde{x} \in \tilde{\mathcal{X}}_v \setminus \{\tilde{x}_{v,*}\}, v \in [V]$ and $R = \lceil \log_2(\omega(\xi, \tilde{\mathcal{X}})) \rceil \ge \lceil \log_2(\omega(\xi, \tilde{\mathcal{X}}_v)) \rceil$ for any $v \in [V]$, we have that conditioning on $\mathcal{J}$, algorithm `CoKernelFB` will return the correct answers $\tilde{x}_{v,*}$ for all $v \in [V]$ using at most $R$ rounds. Therefore, we complete the proof of the error probability guarantee.

For communication rounds, since algorithm `CoKernelFB` has at most $R := \lceil \log_2(\omega(\xi, \tilde{\mathcal{X}})) \rceil$ rounds, the number of communication rounds is bounded by $O(\log(\omega(\xi, \tilde{\mathcal{X}})))$. $\qquad\square$

### E.2 INTERPRETATION OF THEOREM 2

In the following, we interpret the error probability for algorithm `CoKernelFB` with maximum information gain and effective dimension, and decompose it into two parts with respect to task similarities and arm features.

**Corollary 2.** *The error probability of algorithm* CoKernelFC, *denoted by* $Err$, *can also be bounded as follows:*

(a) $Err = O\left(\exp\left(-\dfrac{TV\Delta_{\min}^2}{\Upsilon \log(\omega(\xi, \tilde{\mathcal{X}}))}\right) \cdot n^2 V \log(\omega(\xi, \tilde{\mathcal{X}}))\right),$

  *where $\Upsilon$ is the maximum information gain.*

(b) $Err = O\left(\exp\left(-\dfrac{TV\Delta_{\min}^2}{d_{\text{eff}} \log\left(nV \cdot \left(1 + \frac{\text{Trace}(K_{\lambda^*})}{\xi d_{\text{eff}}}\right)\right) \log(\omega(\xi, \tilde{\mathcal{X}}))}\right) \cdot n^2 V \log(\omega(\xi, \tilde{\mathcal{X}}))\right),$

  *where $d_{\text{eff}}$ is the effective dimension.*

(c) $Err = O\Bigg(\exp\left(-\dfrac{TV\Delta_{\min}^2}{\text{rank}(K_{\mathcal{Z}}) \cdot \text{rank}(K_{\mathcal{X}, \lambda^*}) \log\left(\frac{\text{Trace}(I + \xi^{-1}K_{\lambda^*})}{\text{rank}(K_{\lambda^*})}\right) \log(\omega(\xi, \tilde{\mathcal{X}}))}\right) \cdot$

  $n^2 V \log(\omega(\xi, \tilde{\mathcal{X}}))\Bigg).$

Corollaries 2(a),(b) bound the error probability by maximum information gain and effective dimension, respectively, which capture essential structures of tasks and arm features and only depend on the effective dimension of kernel space.

Corollary 2(c) reveals how task similarities impact the error probability performance. Specifically, when tasks are the same, i.e., $\text{rank}(K_{\mathcal{Z}}) = 1$, the error probability enjoys an exponential decay factor of $V$ compared to single-agent algorithm (Katz-Samuels et al., 2020), and achieves the maximum $V$-speedup. Conversely, when tasks are totally different, i.e., $\text{rank}(K_{\mathcal{Z}}) = V$, the error probability is similar to single-agent results (Katz-Samuels et al., 2020), since in this case information sharing brings no benefit.

*Proof of Corollary 2.* Recall that $\lambda^* := \text{argmax}_{\lambda \in \triangle_{\tilde{\mathcal{X}}}} \log \det\left(I + \xi^{-1}K_\lambda\right)$.

Recalling Eq. (15), we have

$$\rho^*(\xi) \le \frac{8}{\Delta_{\min}^2} \cdot \log \det\left(I + \xi^{-1}K_{\lambda^*}\right).$$

**Maximum Information Gain.** Recall that the maximum information gain over all sample allocation $\lambda \in \triangle_{\tilde{\mathcal{X}}}$ is defined as

$$\Upsilon = \max_{\lambda \in \triangle_{\tilde{\mathcal{X}}}} \log \det\left(I + \xi^{-1}K_\lambda\right) = \log \det\left(I + \xi^{-1}K_{\lambda^*}\right).$$

Recall Eq (16), we have

$$\rho^*(\xi) \le \frac{8}{\Delta_{\min}^2} \cdot \Upsilon.$$

Combining the bound in Theorem 2 and the above equation, the error probability is bounded by

$$Err = O\left(n^2 V \log(\omega(\xi, \tilde{\mathcal{X}})) \cdot \exp\left(-\frac{TV}{\rho^*(\xi) \cdot \log(\omega(\xi, \tilde{\mathcal{X}}))}\right)\right)$$

$$= O\left(n^2 V \log(\omega(\xi, \tilde{\mathcal{X}})) \cdot \exp\left(-\frac{TV\Delta_{\min}^2}{\Upsilon \cdot \log(\omega(\xi, \tilde{\mathcal{X}}))}\right)\right)$$

**Effective Dimension.** Recall that $\alpha_1 \ge \cdots \ge \alpha_{nV}$ denote the eigenvalues of $K_{\lambda^*}$ in decreasing order. The effective dimension of $K_{\lambda^*}$ is defined as

$$d_{\text{eff}} = \min\left\{j : j\xi \log(nV) \ge \sum_{i=j+1}^{nV} \alpha_i\right\}.$$

Recalling Eq. (17), we have

$$\rho^*(\xi) \le \frac{8}{\Delta_{\min}^2} \cdot d_{\text{eff}} \log\left(2nV \cdot \left(1 + \frac{\text{Trace}(K_{\lambda^*})}{\xi d_{\text{eff}}}\right)\right).$$

Combining the bound in Theorem 2 and the above equation, the error probability is bounded by

$$
\begin{aligned}
Err =& O\left(n^2 V \log(\omega(\xi, \tilde{\mathcal{X}})) \cdot \exp\left(-\frac{TV}{\rho^*(\xi) \cdot \log(\omega(\xi, \tilde{\mathcal{X}}))}\right)\right) \\
=& O\left(n^2 V \log(\omega(\xi, \tilde{\mathcal{X}})) \cdot \exp\left(-\frac{TV\Delta_{\min}^2}{d_{\text{eff}} \cdot \log\left(nV \cdot \left(1 + \frac{\text{Trace}(K_{\lambda^*})}{\xi d_{\text{eff}}}\right)\right) \cdot \log(\omega(\xi, \tilde{\mathcal{X}}))}\right)\right)
\end{aligned}
$$

**Decomposition.** Recall that $K_{\lambda^*} = [\sqrt{\lambda_i^* \lambda_j^*} k(\tilde{x}_i, \tilde{x}_j)]_{i,j \in [nV]}$, $K_{\mathcal{Z}} = [k_{\mathcal{Z}}(z_v, z_{v'})]_{v,v' \in [V]}$ and $K_{\mathcal{X},\lambda^*} = [\sqrt{\lambda_i^* \lambda_j^*} k_{\mathcal{X}}(\tilde{x}_i, \tilde{x}_j)]_{i,j \in [nV]}$. Let $\tilde{K}_{\mathcal{Z}} = [k_{\mathcal{Z}}(z_{v_i}, z_{v_j})]_{i,j \in [nV]}$, where for any $i \in [nV]$, $v_i$ denotes the index of the task for the $i$-th arm $\tilde{x}_i$ in the arm set $\tilde{\mathcal{X}}$ and $z_{v_i}$ denotes its task feature. It holds that $\text{rank}(\tilde{K}_{\mathcal{Z}}) = \text{rank}(K_{\mathcal{Z}})$.

Since $K_{\lambda^*}$ is a Hadamard composition of $\tilde{K}_{\mathcal{Z}}$ and $K_{\mathcal{X},\lambda^*}$, we have that $\text{rank}(K_{\lambda^*}) \le \text{rank}(\tilde{K}_{\mathcal{Z}}) \cdot \text{rank}(K_{\mathcal{X},\lambda^*}) = \text{rank}(K_{\mathcal{Z}}) \cdot \text{rank}(K_{\mathcal{X},\lambda^*})$.

Recalling Eq. (18), we have

$$\rho^*(\xi) \le \frac{8}{\Delta_{\min}^2} \cdot \text{rank}(K_{\mathcal{Z}}) \cdot \text{rank}(K_{\mathcal{X},\lambda^*}) \log\left(\frac{\text{Trace}\left(I + \xi^{-1} K_{\lambda^*}\right)}{\text{rank}(K_{\lambda^*})}\right).$$

Combining the bound in Theorem 2 and the above equation, the error probability is bounded by

$$
\begin{aligned}
Err =& O\left(n^2 V \log(\omega(\xi, \tilde{\mathcal{X}})) \cdot \exp\left(-\frac{TV}{\rho^*(\xi) \cdot \log(\omega(\xi, \tilde{\mathcal{X}}))}\right)\right) \\
=& O\Bigg(n^2 V \log(\omega(\xi, \tilde{\mathcal{X}})) \cdot \\
& \exp\left(-\frac{TV\Delta_{\min}^2}{\text{rank}(K_{\mathcal{Z}}) \cdot \text{rank}(K_{\mathcal{X},\lambda^*}) \cdot \log\left(\frac{\text{Trace}(I + \xi^{-1} K_{\lambda^*})}{\text{rank}(K_{\lambda^*})}\right) \cdot \log(\omega(\xi, \tilde{\mathcal{X}}))}\right)\Bigg)
\end{aligned}
$$

Therefore, we complete the proof of Corollary 2. $\qquad\square$

## F  TECHNICAL TOOLS

In this section, we provide some useful technical tools.

**Lemma 3** (Lemma 15 in (Camilleri et al., 2021)). *For* $\lambda^* = \text{argmax}_{\lambda \in \triangle_{\tilde{\mathcal{X}}}} \log \det\left(I + \xi^{-1} \sum_{\tilde{x}' \in \tilde{\mathcal{X}}} \lambda_{\tilde{x}'} \phi(\tilde{x}') \phi(\tilde{x}')^\top\right)$*, we have*

$$\max_{\tilde{x} \in \tilde{\mathcal{X}}} \|\phi(\tilde{x})\|^2_{\left(\xi I + \sum_{\tilde{x}' \in \tilde{\mathcal{X}}} \lambda_{\tilde{x}'}^* \phi(\tilde{x}') \phi(\tilde{x}')^\top\right)^{-1}} = \sum_{\tilde{x} \in \tilde{\mathcal{X}}} \lambda_{\tilde{x}}^* \|\phi(\tilde{x})\|^2_{\left(\xi I + \sum_{\tilde{x}' \in \tilde{\mathcal{X}}} \lambda_{\tilde{x}'}^* \phi(\tilde{x}') \phi(\tilde{x}')^\top\right)^{-1}}$$

**Lemma 4.** *For* $\lambda^* = \text{argmax}_{\lambda \in \triangle_{\tilde{\mathcal{X}}}} \log \det\left(I + \xi^{-1} \sum_{\tilde{x}' \in \tilde{\mathcal{X}}} \lambda_{\tilde{x}'} \phi(\tilde{x}') \phi(\tilde{x}')^\top\right)$*, we have*

$$\sum_{\tilde{x} \in \tilde{\mathcal{X}}} \log\left(1 + \lambda_{\tilde{x}}^* \|\phi(\tilde{x})\|^2_{\left(\xi I + \sum_{\tilde{x}' \in \tilde{\mathcal{X}}} \lambda_{\tilde{x}'}^* \phi(\tilde{x}') \phi(\tilde{x}')^\top\right)^{-1}}\right) \le \log \frac{\det\left(\xi I + \sum_{\tilde{x} \in \tilde{\mathcal{X}}} \lambda_{\tilde{x}}^* \phi(\tilde{x}) \phi(\tilde{x})^\top\right)}{\det\left(\xi I\right)}$$

*Proof of Lemma 4.* This proof is inspired by Lemma 11 in (Abbasi-Yadkori et al., 2011), and uses a similar analytical procedure as that of Lemma 16 in (Camilleri et al., 2021). However, different from the analysis of Lemma 16 in (Camilleri et al., 2021), we do not include the number of samples $N^{(r)}$ in the $\det(\cdot)$ operator in this proof.

For any $j \in [nV]$, let $M_j = \det \left( \xi I + \sum_{i \in [j]} \lambda_i^* \phi(\tilde{x}_i) \phi(\tilde{x}_i)^\top \right)$.

$$\det \left( \xi I + \sum_{\tilde{x} \in \tilde{\mathcal{X}}} \lambda_{\tilde{x}}^* \phi(\tilde{x}) \phi(\tilde{x})^\top \right)$$

$$= \det \left( \xi I + \sum_{i \in [nV-1]} \lambda_i^* \phi(\tilde{x}_i) \phi(\tilde{x}_i)^\top + \lambda_{nV}^* \phi(\tilde{x}_{nV}) \phi(\tilde{x}_{nV})^\top \right)$$

$$= \det \left( M_{nV-1} \right) \det \left( I + \lambda_{nV}^* \cdot M_{nV-1}^{-\frac{1}{2}} \phi(\tilde{x}_{nV}) \left( M_{nV-1}^{-\frac{1}{2}} \phi(\tilde{x}_{nV}) \right)^\top \right)$$

$$= \det \left( M_{nV-1} \right) \det \left( I + \lambda_{nV}^* \cdot \phi(\tilde{x}_{nV})^\top M_{nV-1}^{-1} \phi(\tilde{x}_{nV}) \right)$$

$$= \det \left( M_{nV-1} \right) \left( 1 + \lambda_{nV}^* \| \phi(\tilde{x}_{nV}) \|_{M_{nV-1}^{-1}}^2 \right)$$

$$= \det \left( \xi I \right) \prod_{i=1}^{nV} \left( 1 + \lambda_i^* \| \phi(\tilde{x}_i) \|_{M_{i-1}^{-1}}^2 \right)$$

Thus,

$$\frac{\det \left( \xi I + \sum_{\tilde{x} \in \tilde{\mathcal{X}}} \lambda_{\tilde{x}}^* \phi(\tilde{x}) \phi(\tilde{x})^\top \right)}{\det \left( \xi I \right)} = \prod_{i=1}^{nV} \left( 1 + \lambda_i^* \| \phi(\tilde{x}_i) \|_{M_{i-1}^{-1}}^2 \right)$$

Taking logarithm on both sides, we have

$$\log \frac{\det \left( \xi I + \sum_{\tilde{x} \in \tilde{\mathcal{X}}} \lambda_{\tilde{x}}^* \phi(\tilde{x}) \phi(\tilde{x})^\top \right)}{\det \left( \xi I \right)}$$

$$= \sum_{i=1}^{nV} \log \left( 1 + \lambda_i^* \| \phi(\tilde{x}_i) \|_{M_{i-1}^{-1}}^2 \right)$$

$$\geq \sum_{i=1}^{nV} \log \left( 1 + \lambda_i^* \| \phi(\tilde{x}_i) \|_{M_{nV}^{-1}}^2 \right),$$

which completes the proof of Lemma 4. $\qquad\square$

