# OpenReview forum: "Collaborative Pure Exploration in Kernel Bandit"
_ICLR.cc/2023/Conference — ICLR 2023 poster_

### Official Review · Reviewer_D4Wd · 2022-10-20

**Confidence:** 3
**Correctness:** 4
**Technical Novelty And Significance:** 3
**Empirical Novelty And Significance:** Not applicable
**Recommendation:** 8

**Clarity, Quality, Novelty And Reproducibility:**

On the novelty side, the paper generalizes some well-known problems. The idea of communication rounds is not new, but the authors add a specific and new kernelized estimator to reduce the dimension of sent messages. The communication lower bound for structured arms is new and general (it also holds at least for multi-armed bandits and linear bandits).

**Strength And Weaknesses:**

The studied problem is very general. The paper is technically sound. The analytical results are convincing.

The paper is quite difficult to read. This is largely due to the complexity of the studied problem.
However, sometimes some notations are not well introduced. This complicates the task of the reader. For instance at the end of page 2 \tilde{X} is the set of arms, while in page 3 X_v is also the set of arms for agent v. The reader learns at the end of page 3, what is the difference between \tilde{X} and X, \tilde{X} = Z x X, where Z is the task feature space…
In Algorithm 1, the authors should emphasizes that t is not the standard time index, but the index of communication rounds. May be by simply writing line 3, for round r = 1,2,… do, could help the reader to understand the algorithm.

This paper has a lot of results, including some experiments, but only the analysis of fixed confidence exploration problem is in the main paper, without any experiment.


**Summary Of The Paper:**

In this paper the authors study collaborative exploration in kernel bandits. This problem is very general, and it generalizes a lot of works done in collaborative exploration. Due to the use of kernel, which allows non-linear structure of arms, it generalizes the works done on collaborative exploration in MAB and in linear bandits. Moreover, this work allows agents to collaborate on different tasks: the set of arms and the best arm are not necessary the same for each agent, as well as the features of arms.
The authors study the both exploration problems: fixed confidence and fixed budget. They propose and analyze an algorithm for each problem setting. They notably prove near optimal sample complexity and a near optimal number of communication rounds in the fixed confidence setting. Last but not least for the confidence setting, they provide a lower bound of the near optimal number of communication rounds that generalizes (Tao et al 2019) to structured rewards. Finally the same job is done for the fixed budget setting.


**Summary Of The Review:**

It is a theoretical paper that generalizes some well-known problems and results. The paper is quite hard to read. The authors could help a little bit more the reader. That being said, I clearly vote for its acceptance.

________________________
I read the rebuttal and I thank the authors for their answers.

---

> ### Author Response · Authors · 2022-11-15
> **Response to Reviewer D4Wd - Part 1/2**
>
> Thank you very much for your positive comments and valuable suggestions! We have improved the writing and organization of our paper in the rebuttal version according to your suggestions.
>
> **1. Writing and Notations.**
>
> Following the reviewer's suggestion, we have changed unclear notations and added more clarifications to facilitate readers' understanding.
>
> Our main modifications are summarized as follows.
>
> - We clarified that $\tilde{\mathcal{X}}$ denotes the set of task-arm feature pairs in Section 1, and added explanations on notations $\tilde{\mathcal{X}}$ and $\tilde{\mathcal{x}}$ in Section 3.
> - We revised Line 3 of Algorithm 1 to ``for round $r=1,2,\dots$'', and changed the notation of the round index from $t$ to $r$ throughout the paper to avoid confusion.
> - We modified the organization of the paper, and included the experiments for our algorithms and discussion on experimental results in the main text (Section 6) of our revision.
>
>
> **2. Organization and Experiments.**
>
> According to the reviewer's suggestion, we have incorporated the experiments in Section 6 of our revision. The code and implementation instructions have been provided in our supplementary materials for reproducibility (we have uploaded them when we submitted this work).
>
> For ease of the reviewer's reading, we also present our experimental results here.
>
> In our experiments, we set $V = 5$, $n = 6$, $\delta = 0.005$ and $\mathcal{H}=\mathbb{R}^{d}$, where $d$ is a dimension parameter that will be specified later. We consider three
> different cases of task similarities in Tables 1-3, i.e., $rank(K_{\mathcal{Z}}) = 1$ (all tasks are the same), $rank(K_{\mathcal{Z}}) \in (1,V)$ (tasks are similar), and $rank(K_{\mathcal{Z}}) = V$ (tasks are totally different),
> to show how task similarities impact learning performance in practice.
>
> For the $rank(K_{\mathcal{Z}})=1$ case (Table 1), we set
> $d=4$. For any $v \in [V]$,  $\{\phi(\tilde{x})\}_{\tilde{x} \in \tilde{\mathcal{X}}_v}$ is the set of all $\binom{4}{2}$ vectors in $\mathbb{R}^{4}$, where each vector has two entries $0$ and two entries $1$.
>
> For the $rank(K_{\mathcal{Z}})\in(1,V)$ case (Table 2), we set $d=8$. For any $v \in \\{1,2\\}$, $\{\phi(\tilde{x})\}_{\tilde{x} \in \tilde{\mathcal{X}}_v}$ is the set of all $\binom{4}{2}$ vectors in the first subspaces $\mathbb{R}^{4}$ of the whole space $\mathbb{R}^{8}$, where each vector has two entries $0$ and two entries $1$.
>
> For any $v \in \\{3,4,5\\}$, $\{\phi(\tilde{x})\}_{\tilde{x} \in \tilde{\mathcal{X}}_v}$ is the set of all $\binom{4}{2}$ vectors in the second subspaces $\mathbb{R}^{4}$ of the whole space $\mathbb{R}^{8}$, where each vector has two entries $0$ and two entries $1$.
>
> For the $rank(K_{\mathcal{Z}})=V$ case (Table 3), we set $d=20$. For any $v \in [V]$, $\{\phi(\tilde{x})\}_{\tilde{x} \in \tilde{\mathcal{X}}_v}$ is the set of all $\binom{4}{2}$ vectors in the $v$-th subspace $\mathbb{R}^{4}$ of the whole space $\mathbb{R}^{20}$, where each vector has two entries $0$ and two entries $1$.
>
>
> For all cases, we set $\theta^*=[0.1,0.1+\Delta_{min},\dots,0.1+(d-1)\Delta_{min}]^\top \in \mathbb{R}^{d}$, where $\Delta_{min}$ is a reward gap parameter.
> We change $\Delta_{min} \in [0.15,0.6]$ to generate different instances, and show the sample complexities of algorithms on these different instances. We perform $50$ independent runs, and report the average sample complexities across runs.
>
> We compare  our algorithm CoKernelFC with four baseline algorithms, i.e., CoKernel-IndAlloc, IndRAGE, IndALBA and IndPolyALBA. CoKernel-IndAlloc is an ablation variant of CoKernelFC, where agents use locally optimal sample allocations, instead of a globally optimal sample allocation. IndRAGE, IndALBA and IndPolyALBA are adaptions of single-agent algorithms, which use $V$ copies of existing single-agent algorithms RAGE [Fiez et al., 2019], ALBA [Tao et al., 2018] and PolyALBA [Du et al., 2021] to independently solve the $V$ tasks, respectively. Since all these algorithms determine the number of required samples at the beginning of each round, the sample complexities for different instances can be the same (i.e., use the same number of rounds).
>
>
> **Table 1. $rank(\mathcal{Z})=1$, i.e., all tasks are the same.**
>
> | $\Delta_{min}$  | CoKernelFC (ours)  | CoKernelFC-IndAlloc  | IndRAGE  |  IndALBA | IndPolyALBA  |
> | :------------: | :------------: | :------------: | :------------: | :------------: | :------------: |
> | 0.15  |  9739.28 | 20281.00  | 50185.04  |  3065285.00 |  4161535.32 |
> | 0.2  | 6923.42  | 10140.90  |  34881.62 | 3204695.00  | 3390960.00  |
> |  0.3 |  2109.40 | 4885.00  | 11528.06  | 672555.00  | 881190.00  |
> | 0.4  | 1245.20  | 4885.00 |  6553.80  | 672555.00   |  881190.00 |
> |  0.5 |  1036.60 |  4885.00 |  5153.20 | 251573.76  | 449422.04  |
> |  0.6 | 977.00  | 4885.00  | 4885.00  |  97665.00 |  285120.00 |

---

> > ### Author Response · Authors · 2022-11-15
> > **Response to Reviewer D4Wd - Part 2/2**
> >
> > **Table 2. $rank(\mathcal{Z}) \in (1,V)$, i.e., tasks are similar.**
> >
> > | $\Delta_{min}$  | CoKernelFC (ours)  | CoKernelFC-IndAlloc  | IndRAGE  |  IndALBA | IndPolyALBA  |
> > | :------------: | :------------: | :------------: | :------------: | :------------: | :------------: |
> > | 0.15  |  20529.76 | 29666.72  | 51920.66  |  6207575.00 | 7514873.80  |
> > | 0.2  |  11647.86 |  21923.64 | 33481.30  | 6451960.00  |  6650110.00 |
> > |  0.3 |  4417.44 |  7808.32 | 11572.34  | 1364845.00  |  1619735.00 |
> > | 0.4  |  2815.82 | 4885.00 |  7179.60 |  1364845.00  |  1619735.00 |
> > |  0.5 |  2049.00 |  4885.00 |  5123.40 | 553702.32  |  770530.44 |
> > |  0.6 |  1900.00 |  4885.00 |  4914.80 | 211170.00  |  425285.00 |
> >
> >
> >
> > **Table 3. $rank(\mathcal{Z}) = V$, i.e., tasks are totally different.**
> >
> > | $\Delta_{min}$  | CoKernelFC (ours)  | CoKernelFC-IndAlloc  | IndRAGE  |  IndALBA | IndPolyALBA  |
> > | :------------: | :------------: | :------------: | :------------: | :------------: | :------------: |
> > | 0.15  | 51407.42  | 70460.00 | 50676.80 |  16312385.00 |  17558480.32 |
> > | 0.2  |  34374.84 | 41445.72  | 34252.42  | 16267755.00 | 16479720.00  |
> > |  0.3 |  11208.60 |  14166.10 |  11078.14 |  3454540.00  | 3828730.00  |
> > | 0.4  |  7242.20 | 7060.40 |  7328.60 |   3454540.00  |   3828730.00 |
> > |  0.5 | 4885.60 |  5123.40 | 5332.00  | 1374857.30 | 1785804.80 |
> > |  0.6 | 4408.80 | 4885.00  |  4885.00 | 548700.00  | 823350.00 |
> >
> >
> >
> > The experimental results show that, our  CoKernelFC achieves the best sample complexity when tasks are the same or similar (in which case cooperation among agents can bring benefits). This demonstrates the superiority of our globally optimal sampling strategy and the effectiveness of our cooperation scheme.
> >
> > Moreover, the experimental results reflect that, the more the tasks are similar, the higher learning speedup the agents can attain.
> > Specifically, in the $rank(K_{\mathcal{Z}})=1$ case (Table 1), the sample complexity of CoKernelFC is about $\frac{1}{V}$ of that of the best single-agent adaption IndRAGE, which shows that CoKernelFC achieves a $V$-speedup.
> > In the $rank(K_{\mathcal{Z}})\in(1,V)$ case (Table 2), the ratio of the sample complexity of CoKernelFC over that of IndRAGE is larger than $\frac{1}{V}$, which implies that CoKernelFC achieves a speedup lower than $V$.
> > In the $rank(K_{\mathcal{Z}})=V$ case (Table 3), CoKernelFC performs similar to IndRAGE, since information sharing among agents brings no advantage in this case.
> > These empirical results  match and validate our theoretical analysis.
> >
> >
> >
> > ---
> >
> > References:
> > Tanner Fiez, Lalit Jain, Kevin G. Jamieson,  Lillian Ratliff. Sequential experimental design
> > for transductive linear bandits. NeurIPS, 2019.
> > Chao Tao, Saúl Blanco, Yuan Zhou. Best arm identification in linear bandits with linear dimension dependency. ICML, 2018.
> > Yihan Du, Yuko Kuroki, Wei Chen. Combinatorial pure exploration with full-bandit or partial
> > linear feedback. AAAI, 2021.

---

### Official Review · Reviewer_3Dpt · 2022-10-24

**Confidence:** 3
**Correctness:** 3
**Technical Novelty And Significance:** 3
**Empirical Novelty And Significance:** Not applicable
**Recommendation:** 8

**Clarity, Quality, Novelty And Reproducibility:**

The paper is in general clear and is with novelty. I believe that it makes a contribution to the bandit study.

**Strength And Weaknesses:**

Strength:
1. The paper is easy to follow and provides a thorough overview of existing works.
2. The motivation of this work is convincing: some tasks such as objective detection and object tracking are related but different, and they can be viewed as a possible application of the proposed model.
3. The explanation and analysis of the proposed algorithm are detailed.

Weaknesses:
1. Is it possible to provide some numerical results?

**Summary Of The Paper:**

This work focuses on a new problem: collaboration pure exploration with multiple tasks and general reward structures. It explores both fixed-confidence and fixed-budget settings. Nearly-matching upper and lower bounds on sampling and communication complexity are derived.

**Summary Of The Review:**

This work clarifies its motivation and formulation in the beginning. After that, it studies the pure exploration problem for both fixed-confidence and fixed-budget settings. The nearly-matching bounds convince the efficacy of the proposed algorithms. However, numerical results are appreciated to even improve the work.

========================

After rebuttal: thanks for your reply. You may consider to involve the experiment results in the first version next time.

---

> ### Author Response · Authors · 2022-11-15
> **Response to Reviewer 3Dpt - Part 1/2**
>
> Thank you very much for your time and efforts in reviewing our paper! We have incorporated your comments in our revised version.
>
> **1. Experiments.**
>
> In our original submission, we have provided the experiments in Appendix A and uploaded the code and implementation instructions in supplementary materials.
>
> Following the reviewer's suggestion, we further put the experiments in the main text (Section 6) of our revision to improve the organization of this paper.
>
> For ease of the reviewer's reading, we also present our experimental results here.
>
> In our experiments, we set $V = 5$, $n = 6$, $\delta = 0.005$ and $\mathcal{H}=\mathbb{R}^{d}$, where $d$ is a dimension parameter that will be specified later. We consider three
> different cases of task similarities in Tables 1-3, i.e., $rank(K_{\mathcal{Z}}) = 1$ (all tasks are the same), $rank(K_{\mathcal{Z}}) \in (1,V)$ (tasks are similar), and $rank(K_{\mathcal{Z}}) = V$ (tasks are totally different),
> to show how task similarities impact learning performance in practice.
>
> For the $rank(K_{\mathcal{Z}})=1$ case (Table 1), we set
> $d=4$. For any $v \in [V]$,  $\{\phi(\tilde{x})\}_{\tilde{x} \in \tilde{\mathcal{X}}_v}$ is the set of all $\binom{4}{2}$ vectors in $\mathbb{R}^{4}$, where each vector has two entries $0$ and two entries $1$.
>
> For the $rank(K_{\mathcal{Z}})\in(1,V)$ case (Table 2), we set $d=8$. For any $v \in \\{1,2\\}$, $\{\phi(\tilde{x})\}_{\tilde{x} \in \tilde{\mathcal{X}}_v}$ is the set of all $\binom{4}{2}$ vectors in the first subspaces $\mathbb{R}^{4}$ of the whole space $\mathbb{R}^{8}$, where each vector has two entries $0$ and two entries $1$.
>
> For any $v \in \\{3,4,5\\}$, $\{\phi(\tilde{x})\}_{\tilde{x} \in \tilde{\mathcal{X}}_v}$ is the set of all $\binom{4}{2}$ vectors in the second subspaces $\mathbb{R}^{4}$ of the whole space $\mathbb{R}^{8}$, where each vector has two entries $0$ and two entries $1$.
>
> For the $rank(K_{\mathcal{Z}})=V$ case (Table 3), we set $d=20$. For any $v \in [V]$, $\{\phi(\tilde{x})\}_{\tilde{x} \in \tilde{\mathcal{X}}_v}$ is the set of all $\binom{4}{2}$ vectors in the $v$-th subspace $\mathbb{R}^{4}$ of the whole space $\mathbb{R}^{20}$, where each vector has two entries $0$ and two entries $1$.
>
>
> For all cases, we set $\theta^*=[0.1,0.1+\Delta_{min},\dots,0.1+(d-1)\Delta_{min}]^\top \in \mathbb{R}^{d}$, where $\Delta_{min}$ is a reward gap parameter.
> We change $\Delta_{min} \in [0.15,0.6]$ to generate different instances, and show the sample complexities of algorithms on these different instances. We perform $50$ independent runs, and report the average sample complexities across runs.
>
> We compare  our algorithm CoKernelFC with four baseline algorithms, i.e., CoKernel-IndAlloc, IndRAGE, IndALBA and IndPolyALBA. CoKernel-IndAlloc is an ablation variant of CoKernelFC, where agents use locally optimal sample allocations, instead of a globally optimal sample allocation. IndRAGE, IndALBA and IndPolyALBA are adaptions of single-agent algorithms, which use $V$ copies of existing single-agent algorithms RAGE [Fiez et al., 2019], ALBA [Tao et al., 2018] and PolyALBA [Du et al., 2021] to independently solve the $V$ tasks, respectively. Since all these algorithms determine the number of required samples at the beginning of each round, the sample complexities for different instances can be the same (i.e., use the same number of rounds).
>
>
> **Table 1. $rank(\mathcal{Z})=1$, i.e., all tasks are the same.**
>
> | $\Delta_{min}$  | CoKernelFC (ours)  | CoKernelFC-IndAlloc  | IndRAGE  |  IndALBA | IndPolyALBA  |
> | :------------: | :------------: | :------------: | :------------: | :------------: | :------------: |
> | 0.15  |  9739.28 | 20281.00  | 50185.04  |  3065285.00 |  4161535.32 |
> | 0.2  | 6923.42  | 10140.90  |  34881.62 | 3204695.00  | 3390960.00  |
> |  0.3 |  2109.40 | 4885.00  | 11528.06  | 672555.00  | 881190.00  |
> | 0.4  | 1245.20  | 4885.00 |  6553.80  | 672555.00   |  881190.00 |
> |  0.5 |  1036.60 |  4885.00 |  5153.20 | 251573.76  | 449422.04  |
> |  0.6 | 977.00  | 4885.00  | 4885.00  |  97665.00 |  285120.00 |
>
> **Table 2. $rank(\mathcal{Z}) \in (1,V)$, i.e., tasks are similar.**
>
> | $\Delta_{min}$  | CoKernelFC (ours)  | CoKernelFC-IndAlloc  | IndRAGE  |  IndALBA | IndPolyALBA  |
> | :------------: | :------------: | :------------: | :------------: | :------------: | :------------: |
> | 0.15  |  20529.76 | 29666.72  | 51920.66  |  6207575.00 | 7514873.80  |
> | 0.2  |  11647.86 |  21923.64 | 33481.30  | 6451960.00  |  6650110.00 |
> |  0.3 |  4417.44 |  7808.32 | 11572.34  | 1364845.00  |  1619735.00 |
> | 0.4  |  2815.82 | 4885.00 |  7179.60 |  1364845.00  |  1619735.00 |
> |  0.5 |  2049.00 |  4885.00 |  5123.40 | 553702.32  |  770530.44 |
> |  0.6 |  1900.00 |  4885.00 |  4914.80 | 211170.00  |  425285.00 |

---

> > ### Author Response · Authors · 2022-11-15
> > **Response to Reviewer 3Dpt - Part 2/2**
> >
> > **Table 3. $rank(\mathcal{Z}) = V$, i.e., tasks are totally different.**
> >
> > | $\Delta_{min}$  | CoKernelFC (ours)  | CoKernelFC-IndAlloc  | IndRAGE  |  IndALBA | IndPolyALBA  |
> > | :------------: | :------------: | :------------: | :------------: | :------------: | :------------: |
> > | 0.15  | 51407.42  | 70460.00 | 50676.80 |  16312385.00 |  17558480.32 |
> > | 0.2  |  34374.84 | 41445.72  | 34252.42  | 16267755.00 | 16479720.00  |
> > |  0.3 |  11208.60 |  14166.10 |  11078.14 |  3454540.00  | 3828730.00  |
> > | 0.4  |  7242.20 | 7060.40 |  7328.60 |   3454540.00  |   3828730.00 |
> > |  0.5 | 4885.60 |  5123.40 | 5332.00  | 1374857.30 | 1785804.80 |
> > |  0.6 | 4408.80 | 4885.00  |  4885.00 | 548700.00  | 823350.00 |
> >
> >
> >
> > The experimental results show that, our  CoKernelFC achieves the best sample complexity when tasks are the same or similar (in which case cooperation among agents can bring benefits). This demonstrates the superiority of our globally optimal sampling strategy and the effectiveness of our cooperation scheme.
> >
> > Moreover, the experimental results reflect that, the more the tasks are similar, the higher learning speedup the agents can attain.
> > Specifically, in the $rank(K_{\mathcal{Z}})=1$ case (Table 1), the sample complexity of CoKernelFC is about $\frac{1}{V}$ of that of the best single-agent adaption IndRAGE, which shows that CoKernelFC achieves a $V$-speedup.
> > In the $rank(K_{\mathcal{Z}})\in(1,V)$ case (Table 2), the ratio of the sample complexity of CoKernelFC over that of IndRAGE is larger than $\frac{1}{V}$, which implies that CoKernelFC achieves a speedup lower than $V$.
> > In the $rank(K_{\mathcal{Z}})=V$ case (Table 3), CoKernelFC performs similar to IndRAGE, since information sharing among agents brings no advantage in this case.
> > These empirical results  match and validate our theoretical analysis.
> >
> > ---
> >
> > References:
> > Tanner Fiez, Lalit Jain, Kevin G. Jamieson,  Lillian Ratliff. Sequential experimental design
> > for transductive linear bandits. NeurIPS, 2019.
> > Chao Tao, Saúl Blanco, Yuan Zhou. Best arm identification in linear bandits with linear dimension dependency. ICML, 2018.
> > Yihan Du, Yuko Kuroki, Wei Chen. Combinatorial pure exploration with full-bandit or partial
> > linear feedback. AAAI, 2021.

---

### Official Review · Reviewer_cGaa · 2022-10-25

**Confidence:** 2
**Correctness:** 4
**Technical Novelty And Significance:** 3
**Empirical Novelty And Significance:** Not applicable
**Recommendation:** 6

**Clarity, Quality, Novelty And Reproducibility:**

The work and theorems are high quality and clear. Code is provided for reproducibility.

**Strength And Weaknesses:**

Strengths
- Model is useful for neural architecture search problems.
- Algorithms achieve communication optimality.

Weaknesses
- Lack of experiments for this model in the main paper. Would be great to see applications in the main work.

**Summary Of The Paper:**

The authors prefer a new kernel bandit model where multiple agents collaborate to complete different but related tasks with limited communication. The authors devise a novel communication scheme and design algorithms for fixed-confidence and fixed-budget objectives.

**Summary Of The Review:**

This paper provides novel algorithms for a well-founded and novel problem setting, which provide a clear contribution. While experiments are not the primary focus of the paper, it would be nice to see more experimental evaluation in the main paper.

---

> ### Author Response · Authors · 2022-11-15
> **Response to Reviewer cGaa - Part 1/2**
>
> Thank you very much for your insightful comments! We have revised our paper in the rebuttal version according to your suggestions.
>
> **1. Experiments.**
>
> Following the reviewer's suggestion, we have included the experiments in the main text (Section 6) of our revision. The code and implementation instructions have been provided in our supplementary materials for reproducibility (we have uploaded them when we submitted this work).
>
> For ease of the reviewer's reading, we also present our experimental results here.
>
> In our experiments, we set $V = 5$, $n = 6$, $\delta = 0.005$ and $\mathcal{H}=\mathbb{R}^{d}$, where $d$ is a dimension parameter that will be specified later. We consider three
> different cases of task similarities in Tables 1-3, i.e., $rank(K_{\mathcal{Z}}) = 1$ (all tasks are the same), $rank(K_{\mathcal{Z}}) \in (1,V)$ (tasks are similar), and $rank(K_{\mathcal{Z}}) = V$ (tasks are totally different),
> to show how task similarities impact learning performance in practice.
>
> For the $rank(K_{\mathcal{Z}})=1$ case (Table 1), we set
> $d=4$. For any $v \in [V]$,  $\{\phi(\tilde{x})\}_{\tilde{x} \in \tilde{\mathcal{X}}_v}$ is the set of all $\binom{4}{2}$ vectors in $\mathbb{R}^{4}$, where each vector has two entries $0$ and two entries $1$.
>
> For the $rank(K_{\mathcal{Z}})\in(1,V)$ case (Table 2), we set $d=8$. For any $v \in \\{1,2\\}$, $\{\phi(\tilde{x})\}_{\tilde{x} \in \tilde{\mathcal{X}}_v}$ is the set of all $\binom{4}{2}$ vectors in the first subspaces $\mathbb{R}^{4}$ of the whole space $\mathbb{R}^{8}$, where each vector has two entries $0$ and two entries $1$.
>
> For any $v \in \\{3,4,5\\}$, $\{\phi(\tilde{x})\}_{\tilde{x} \in \tilde{\mathcal{X}}_v}$ is the set of all $\binom{4}{2}$ vectors in the second subspaces $\mathbb{R}^{4}$ of the whole space $\mathbb{R}^{8}$, where each vector has two entries $0$ and two entries $1$.
>
> For the $rank(K_{\mathcal{Z}})=V$ case (Table 3), we set $d=20$. For any $v \in [V]$, $\{\phi(\tilde{x})\}_{\tilde{x} \in \tilde{\mathcal{X}}_v}$ is the set of all $\binom{4}{2}$ vectors in the $v$-th subspace $\mathbb{R}^{4}$ of the whole space $\mathbb{R}^{20}$, where each vector has two entries $0$ and two entries $1$.
>
>
> For all cases, we set $\theta^*=[0.1,0.1+\Delta_{min},\dots,0.1+(d-1)\Delta_{min}]^\top \in \mathbb{R}^{d}$, where $\Delta_{min}$ is a reward gap parameter.
> We change $\Delta_{min} \in [0.15,0.6]$ to generate different instances, and show the sample complexities of algorithms on these different instances. We perform $50$ independent runs, and report the average sample complexities across runs.
>
> We compare  our algorithm CoKernelFC with four baseline algorithms, i.e., CoKernel-IndAlloc, IndRAGE, IndALBA and IndPolyALBA. CoKernel-IndAlloc is an ablation variant of CoKernelFC, where agents use locally optimal sample allocations, instead of a globally optimal sample allocation. IndRAGE, IndALBA and IndPolyALBA are adaptions of single-agent algorithms, which use $V$ copies of existing single-agent algorithms RAGE [Fiez et al., 2019], ALBA [Tao et al., 2018] and PolyALBA [Du et al., 2021] to independently solve the $V$ tasks, respectively. Since all these algorithms determine the number of required samples at the beginning of each round, the sample complexities for different instances can be the same (i.e., use the same number of rounds).
>
>
> **Table 1. $rank(\mathcal{Z})=1$, i.e., all tasks are the same.**
>
> | $\Delta_{min}$  | CoKernelFC (ours)  | CoKernelFC-IndAlloc  | IndRAGE  |  IndALBA | IndPolyALBA  |
> | :------------: | :------------: | :------------: | :------------: | :------------: | :------------: |
> | 0.15  |  9739.28 | 20281.00  | 50185.04  |  3065285.00 |  4161535.32 |
> | 0.2  | 6923.42  | 10140.90  |  34881.62 | 3204695.00  | 3390960.00  |
> |  0.3 |  2109.40 | 4885.00  | 11528.06  | 672555.00  | 881190.00  |
> | 0.4  | 1245.20  | 4885.00 |  6553.80  | 672555.00   |  881190.00 |
> |  0.5 |  1036.60 |  4885.00 |  5153.20 | 251573.76  | 449422.04  |
> |  0.6 | 977.00  | 4885.00  | 4885.00  |  97665.00 |  285120.00 |
>
> **Table 2. $rank(\mathcal{Z}) \in (1,V)$, i.e., tasks are similar.**
>
> | $\Delta_{min}$  | CoKernelFC (ours)  | CoKernelFC-IndAlloc  | IndRAGE  |  IndALBA | IndPolyALBA  |
> | :------------: | :------------: | :------------: | :------------: | :------------: | :------------: |
> | 0.15  |  20529.76 | 29666.72  | 51920.66  |  6207575.00 | 7514873.80  |
> | 0.2  |  11647.86 |  21923.64 | 33481.30  | 6451960.00  |  6650110.00 |
> |  0.3 |  4417.44 |  7808.32 | 11572.34  | 1364845.00  |  1619735.00 |
> | 0.4  |  2815.82 | 4885.00 |  7179.60 |  1364845.00  |  1619735.00 |
> |  0.5 |  2049.00 |  4885.00 |  5123.40 | 553702.32  |  770530.44 |
> |  0.6 |  1900.00 |  4885.00 |  4914.80 | 211170.00  |  425285.00 |

---

> > ### Author Response · Authors · 2022-11-15
> > **Response to Reviewer cGaa - Part 2/2**
> >
> > **Table 3. $rank(\mathcal{Z}) = V$, i.e., tasks are totally different.**
> >
> > | $\Delta_{min}$  | CoKernelFC (ours)  | CoKernelFC-IndAlloc  | IndRAGE  |  IndALBA | IndPolyALBA  |
> > | :------------: | :------------: | :------------: | :------------: | :------------: | :------------: |
> > | 0.15  | 51407.42  | 70460.00 | 50676.80 |  16312385.00 |  17558480.32 |
> > | 0.2  |  34374.84 | 41445.72  | 34252.42  | 16267755.00 | 16479720.00  |
> > |  0.3 |  11208.60 |  14166.10 |  11078.14 |  3454540.00  | 3828730.00  |
> > | 0.4  |  7242.20 | 7060.40 |  7328.60 |   3454540.00  |   3828730.00 |
> > |  0.5 | 4885.60 |  5123.40 | 5332.00  | 1374857.30 | 1785804.80 |
> > |  0.6 | 4408.80 | 4885.00  |  4885.00 | 548700.00  | 823350.00 |
> >
> >
> >
> > The experimental results show that, our  CoKernelFC achieves the best sample complexity when tasks are the same or similar (in which case cooperation among agents can bring benefits). This demonstrates the superiority of our globally optimal sampling strategy and the effectiveness of our cooperation scheme.
> >
> > Moreover, the experimental results reflect that, the more the tasks are similar, the higher learning speedup the agents can attain.
> > Specifically, in the $rank(K_{\mathcal{Z}})=1$ case (Table 1), the sample complexity of CoKernelFC is about $\frac{1}{V}$ of that of the best single-agent adaption IndRAGE, which shows that CoKernelFC achieves a $V$-speedup.
> > In the $rank(K_{\mathcal{Z}})\in(1,V)$ case (Table 2), the ratio of the sample complexity of CoKernelFC over that of IndRAGE is larger than $\frac{1}{V}$, which implies that CoKernelFC achieves a speedup lower than $V$.
> > In the $rank(K_{\mathcal{Z}})=V$ case (Table 3), CoKernelFC performs similar to IndRAGE, since information sharing among agents brings no advantage in this case.
> > These empirical results  match and validate our theoretical analysis.
> >
> >
> >
> >
> > References:
> > Tanner Fiez, Lalit Jain, Kevin G. Jamieson,  Lillian Ratliff. Sequential experimental design
> > for transductive linear bandits. NeurIPS, 2019.
> > Chao Tao, Saúl Blanco, Yuan Zhou. Best arm identification in linear bandits with linear dimension dependency. ICML, 2018.
> > Yihan Du, Yuko Kuroki, Wei Chen. Combinatorial pure exploration with full-bandit or partial
> > linear feedback. AAAI, 2021.
> >
> > ---
> >
> > Finally, we want to thank the reviewer again for the helpful comments for our work! If our response resolves your concerns at a satisfactory level, we hope that the reviewer can kindly consider raising the rating of our paper. Certainly, we are more than happy to address any further questions you may have during the discussion period.

---

> > > ### Author Response · Authors · 2022-11-29
> > > **Dear Reviewer cGaa, We Are Wondering If Your Concerns Are Addressed**
> > >
> > > Dear Reviewer cGaa,
> > >
> > > We are wondering if you have gotten a chance to look over our responses and revisions and if your concerns are resolved.
> > >
> > > If our response resolves your concerns at a satisfactory level, we would like to ask you to kindly consider raising the score for our work. Certainly, we are more than happy to answer any further questions you may have during the discussion period.
> > >
> > > Thank you very much!
> > >
> > > Best regards,
> > > Authors

---

### Official Review · Reviewer_j71S · 2022-10-25

**Confidence:** 4
**Correctness:** 4
**Technical Novelty And Significance:** 3
**Empirical Novelty And Significance:** 2
**Recommendation:** 6

**Clarity, Quality, Novelty And Reproducibility:**

This paper is written well and the theory established looks comprehensive and strict. It is the first one trying to identify the best arm in kernel bandits collaboratively using the communicaiton model formulated in CoPE.

**Details Of Ethics Concerns:**

I have no concerns.

**Strength And Weaknesses:**

Strength
*  This paper is a valuable extention of previous CoPE formulation. Each agent may experience a varied environment (reward functions) because of the characteristics of kernel bandits.
* Strong and comprehensive theoretical analysis is provided.

Weaknesses
* In contrast to CoPE, where the sample complexity is defined as the greatest number of samples used per agent, the purpose specified in this study is a little bit different. The sample complexity is defined in this study as the average number of samples across all agents, which makes the speed-up definition less strong than that in CoPE. Since the running duration is determined by the one running the longest, I believe the definition in CoPE is more reasonable practically.
* It is better if the authors could put the experiment results in the main paper

**Summary Of The Paper:**

This paper looks into the problem of cooperatively determining the best arm in kernel bandits. The study builds on the prior CoPE formulation, which emphasizes the conventional MAB model, while retaining the fundamental communication concept. This paper makes two contributions: 1) it generalizes earlier work to the kernel bandit problem space, and 2) it produces nearly-optimal algorithms for both fixed-confidence and fixed-budget scenarios.

Minor:

It is better if the author(s) could provide line numbers in the manuscript.

**Summary Of The Review:**

This paper is a valuable extention of the previous work CoPE and it is the first one trying to identify the best arm in kernel bandits collaboratively using the communicaiton model formulated in CoPE. Additionally, it provides nearly-optimal algorithms. Overall I think it is a nice paper. However, I am a little bit worried about the goal of the problem, which tries to optimize the average number of samples per agent instead of the maximum one. I am happy to raise my score if the author(s) could provide practical examples to motivate this goal.

After rebuttal
--------------------------
The author(s) have promised to include the discussions regarding the difference between the goal of this paper and that of CoPE, hence I would like to increase my score.

---

> ### Author Response · Authors · 2022-11-15
> **Reponse to Reviewer j71S - Part 1/3**
>
> Thank you very much for your time and efforts in reviewing our paper! We have revised our paper according to your suggestions in the rebuttal version.
>
> **1. Motivation of Minimizing the Average Number of Samples.**
>
> Our goal of minimizing the average number of samples per agent is most suitable for the applications where **obtaining a sample is expensive**, and we want to save the total (average) number of samples used, such as clinical trials and car crash test.
>
>
> For example, consider a clinical trial scenario [Leon et al., 2019], where a medical institution wants to identify the best treatments for different and related tasks (e.g., find the best treatments for patients in different ages).
> This institution conducts multiple clinical trials for different tasks in parallel, and share the results obtained from different tasks, in order to expedite the process of best treatment identification.
> In this case, conducting a trial and obtaining a sample often consumes significant medical resources, and is costly (e.g.,  convalescent plasma treatments for COVID-19 [Zame et al., 2020] and organ transplant surgeries [Cornor et al., 2021]). Therefore, in order to save the medical resources and funds, the medical institution hopes to use as few trials as possible to complete  learning. In such scenarios, minimizing the total (average) number of samples used can be more desirable than minimizing the largest number of samples among all the tasks (which can cost more samples in total).
>
> As another example, imagine that a car company performs multiple groups of crash tests in order to select the best materials for different types of cars (i.e., different but related tasks) [Bohn et al., 2013]. The company shares the data collected from different groups of crash tests to facilitate the material selection process.
> Since performing a crash test is expensive, the car company wants to reduce the total number of tests conducted, in order to save the development cost.
> In this case, the company may prefer to minimize the total (average) number of samples required, rather than minimizing the largest number of samples among all the tasks.
>
> We have included these discussions in Sections 3 of our revision.
>
> ---
>
> References:
> Weninger, Leon, Qianyu Liu, Dorit Merhof. Multi-task learning for brain tumor segmentation. International MICCAI Brainlesion Workshop, 2019.
> William R. Zame, Ioana Bica, Cong Shen, Alicia Curth, Hyun-Suk Lee, Stuart Bailey, James Weatherall, David Wright, Frank Bretz, Mihaela van der Schaar. Machine learning for clinical trials in the era of COVID-19. Statistics in Biopharmaceutical Research, 2020.
> Katie Connor, Eoin D. O'Sullivan, Lorna P Marson, Stephen J Wigmore, Ewen M Harrison. The future role of machine learning in clinical transplantation. Transplantation. 2021.
> Bastian Bohn, Jochen Garcke, Rodrigo Iza-Teran, Alexander Paprotny, Benjamin Peherstorfer, Ulf Schepsmeier, Clemens-August Thole. Analysis of car crash simulation data with nonlinear machine learning methods. Procedia Computer Science, 2013.

---

> > ### Comment · Reviewer_j71S · 2022-11-17
> > **The examples given are not convincing**
> >
> > I would like to thank the author(s) for adding two particular examples. The instances offered, nevertheless, are unclear to me.
> >
> > For the first example, I feel that the key of performing a clinical trial is not the cost but rather the time needed to collect the result. I don't think it's feasible to do many crash tests in the second scenario, and the difference between the various approaches utilizing a small number of tests might not be substantial. It would be preferred if the author(s) could provide more evidence to motivate the problem setup.

---

> > > ### Author Response · Authors · 2022-11-19
> > > **Additional Response to Reviewer j71S**
> > >
> > > Thank you very much for your reply!
> > >
> > > First, we would like to emphasize that when all tasks are the same (i.e., the setting considered in prior CoPE works [Tao et al., 2019; Karpov et al., 2020]), our algorithm CoKernelFC can also achieve a $V$-speedup under the speedup definition based on the maximum number of samples among agents, since in this case each agent has the same sample complexity.
> > > In other words, when our problem reduces to the same setting and uses the same speedup definition as in prior CoPE works [Tao et al., 2019; Karpov et al., 2020], our sample complexity and learning speedup results **match (and are no weaker than)** existing results.
> > >
> > > Next, we would like to justify that while saving time is an important objective in ordinary clinical trials, there are other resource-limited clinical trials where **obtaining a sample consumes significant medical resources**, and saving the total number of samples (medical resources) is more desirable.
> > > For example, in organ transplant trials [Connor et al., 2021], available organs are limited and precious. Hence, the medical institute prefers to minimize the total number of trials (organs) required, and may be willing to wait for more time to achieve this objective.
> > >
> > > Besides clinical trials, our objective of minimizing the average (total) number of samples is also very useful for other scenarios where **obtaining a sample consumes scarce  resources**, and we want to save the total number of resources consumed.
> > > For example, consider a nuclear industry development scenario [Murty \& Charit, 2013], where a nuclear energy institute wants to conduct multiple  experiments to select the best materials for different types of nuclear reactors (i.e., different tasks), and shared the obtained data to facilitate the development.
> > > In this scenario, conducting a nuclear reaction experiment consumes a lot of rare metal resources which are limited in the nature and expensive to extract, and  also produces nuclear wastes which are difficult to dispose. Therefore, in order to save irreversible rare metal resources and reduce nuclear wastes, this institute prefers to minimize the total number of  nuclear reaction experiments required, than minimizing the maximum number of experiments among tasks (duration time).
> > >
> > >
> > > ---
> > >
> > > References:
> > > Chao Tao, Qin Zhang, and Yuan Zhou. Collaborative learning with limited interaction: Tight bounds for distributed exploration in multi-armed bandits. FOCS, 2019.
> > > Nikolai Karpov, Qin Zhang, and Yuan Zhou. Collaborative top distribution identifications with limited interaction. FOCS, 2020.
> > > Katie Connor, Eoin D. O'Sullivan, Lorna P Marson, Stephen J Wigmore, Ewen M Harrison. The future role of machine learning in clinical transplantation. Transplantation. 2021.
> > > K. Linga Murty, and Indrajit Charit. An introduction to nuclear materials: fundamentals and applications. John Wiley \& Sons, 2013.

---

> > > > ### Comment · Reviewer_j71S · 2022-11-25
> > > > **Response after rebuttal**
> > > >
> > > > I appreciate the authors' thorough response. I hope these discussions are included in the paper's amended edition. Given the circumstances, I'll maintain my score.

---

> > > > > ### Author Response · Authors · 2022-11-26
> > > > > **Thank You for Your Appreciation! We Sincerely Hope that You Could Kindly Consider Raising the Score**
> > > > >
> > > > > Thank you very much for your reply and appreciation!
> > > > >
> > > > > We have included the motivation of minimizing the average (total) number of samples in Section 3 of our rebuttal version, and will certainly incorporated more discussions in our further revision (since the rebuttal version is not allowed to update now).
> > > > >
> > > > > Since the reviewer also appreciates our response, if our response addresses your concerns to a satisfactory level, we sincerely hope that the reviewer could kindly consider raising the score for our paper. In addition, we are more than happy to answer any further questions you may have during the discussion period.
> > > > >
> > > > > Thank you again for your time in reviewing our paper!

---

> > > > > > ### Comment · Reviewer_j71S · 2022-11-29
> > > > > > **Thanks for the response**
> > > > > >
> > > > > > I would like to thank the author(s) for the effort to include more discussions. Given the promise, I will increase the score.

---

> > > > > > > ### Author Response · Authors · 2022-11-30
> > > > > > > **Thank You Very Much for Raising Your Score for Our Work**
> > > > > > >
> > > > > > > Thank you very much for raising your score and your time in reviewing our paper! Certainly, we will incorporate more discussions on the differences of goals in our revision according to your comments.

---

> ### Author Response · Authors · 2022-11-15
> **Response to Reviewer j71S - Part 2/3**
>
> **2. Experiments.**
>
>
> Following the reviewer's suggestion, we have included the experiments in the main text (Section 6) of our revision. The code and implementation instructions have been provided in our supplementary materials for reproducibility (we have uploaded them when we submitted this work).
>
> For ease of the reviewer's reading, we also present our experimental results here.
>
> In our experiments, we set $V = 5$, $n = 6$, $\delta = 0.005$ and $\mathcal{H}=\mathbb{R}^{d}$, where $d$ is a dimension parameter that will be specified later. We consider three
> different cases of task similarities in Tables 1-3, i.e., $rank(K_{\mathcal{Z}}) = 1$ (all tasks are the same), $rank(K_{\mathcal{Z}}) \in (1,V)$ (tasks are similar), and $rank(K_{\mathcal{Z}}) = V$ (tasks are totally different),
> to show how task similarities impact learning performance in practice.
>
> For the $rank(K_{\mathcal{Z}})=1$ case (Table 1), we set
> $d=4$. For any $v \in [V]$,  $\{\phi(\tilde{x})\}_{\tilde{x} \in \tilde{\mathcal{X}}_v}$ is the set of all $\binom{4}{2}$ vectors in $\mathbb{R}^{4}$, where each vector has two entries $0$ and two entries $1$.
>
> For the $rank(K_{\mathcal{Z}})\in(1,V)$ case (Table 2), we set $d=8$. For any $v \in \\{1,2\\}$, $\{\phi(\tilde{x})\}_{\tilde{x} \in \tilde{\mathcal{X}}_v}$ is the set of all $\binom{4}{2}$ vectors in the first subspaces $\mathbb{R}^{4}$ of the whole space $\mathbb{R}^{8}$, where each vector has two entries $0$ and two entries $1$.
>
> For any $v \in \\{3,4,5\\}$, $\{\phi(\tilde{x})\}_{\tilde{x} \in \tilde{\mathcal{X}}_v}$ is the set of all $\binom{4}{2}$ vectors in the second subspaces $\mathbb{R}^{4}$ of the whole space $\mathbb{R}^{8}$, where each vector has two entries $0$ and two entries $1$.
>
> For the $rank(K_{\mathcal{Z}})=V$ case (Table 3), we set $d=20$. For any $v \in [V]$, $\{\phi(\tilde{x})\}_{\tilde{x} \in \tilde{\mathcal{X}}_v}$ is the set of all $\binom{4}{2}$ vectors in the $v$-th subspace $\mathbb{R}^{4}$ of the whole space $\mathbb{R}^{20}$, where each vector has two entries $0$ and two entries $1$.
>
>
> For all cases, we set $\theta^*=[0.1,0.1+\Delta_{min},\dots,0.1+(d-1)\Delta_{min}]^\top \in \mathbb{R}^{d}$, where $\Delta_{min}$ is a reward gap parameter.
> We change $\Delta_{min} \in [0.15,0.6]$ to generate different instances, and show the sample complexities of algorithms on these different instances. We perform $50$ independent runs, and report the average sample complexities across runs.
>
> We compare  our algorithm CoKernelFC with four baseline algorithms, i.e., CoKernel-IndAlloc, IndRAGE, IndALBA and IndPolyALBA. CoKernel-IndAlloc is an ablation variant of CoKernelFC, where agents use locally optimal sample allocations, instead of a globally optimal sample allocation. IndRAGE, IndALBA and IndPolyALBA are adaptions of single-agent algorithms, which use $V$ copies of existing single-agent algorithms RAGE [Fiez et al., 2019], ALBA [Tao et al., 2018] and PolyALBA [Du et al., 2021] to independently solve the $V$ tasks, respectively. Since all these algorithms determine the number of required samples at the beginning of each round, the sample complexities for different instances can be the same (i.e., use the same number of rounds).
>
>
> **Table 1. $rank(\mathcal{Z})=1$, i.e., all tasks are the same.**
>
> | $\Delta_{min}$  | CoKernelFC (ours)  | CoKernelFC-IndAlloc  | IndRAGE  |  IndALBA | IndPolyALBA  |
> | :------------: | :------------: | :------------: | :------------: | :------------: | :------------: |
> | 0.15  |  9739.28 | 20281.00  | 50185.04  |  3065285.00 |  4161535.32 |
> | 0.2  | 6923.42  | 10140.90  |  34881.62 | 3204695.00  | 3390960.00  |
> |  0.3 |  2109.40 | 4885.00  | 11528.06  | 672555.00  | 881190.00  |
> | 0.4  | 1245.20  | 4885.00 |  6553.80  | 672555.00   |  881190.00 |
> |  0.5 |  1036.60 |  4885.00 |  5153.20 | 251573.76  | 449422.04  |
> |  0.6 | 977.00  | 4885.00  | 4885.00  |  97665.00 |  285120.00 |
>
> **Table 2. $rank(\mathcal{Z}) \in (1,V)$, i.e., tasks are similar.**
>
> | $\Delta_{min}$  | CoKernelFC (ours)  | CoKernelFC-IndAlloc  | IndRAGE  |  IndALBA | IndPolyALBA  |
> | :------------: | :------------: | :------------: | :------------: | :------------: | :------------: |
> | 0.15  |  20529.76 | 29666.72  | 51920.66  |  6207575.00 | 7514873.80  |
> | 0.2  |  11647.86 |  21923.64 | 33481.30  | 6451960.00  |  6650110.00 |
> |  0.3 |  4417.44 |  7808.32 | 11572.34  | 1364845.00  |  1619735.00 |
> | 0.4  |  2815.82 | 4885.00 |  7179.60 |  1364845.00  |  1619735.00 |
> |  0.5 |  2049.00 |  4885.00 |  5123.40 | 553702.32  |  770530.44 |
> |  0.6 |  1900.00 |  4885.00 |  4914.80 | 211170.00  |  425285.00 |

---

> ### Author Response · Authors · 2022-11-15
> **Response to Reviewer j71S - Part 3/3**
>
> **Table 3. $rank(\mathcal{Z}) = V$, i.e., tasks are totally different.**
>
> | $\Delta_{min}$  | CoKernelFC (ours)  | CoKernelFC-IndAlloc  | IndRAGE  |  IndALBA | IndPolyALBA  |
> | :------------: | :------------: | :------------: | :------------: | :------------: | :------------: |
> | 0.15  | 51407.42  | 70460.00 | 50676.80 |  16312385.00 |  17558480.32 |
> | 0.2  |  34374.84 | 41445.72  | 34252.42  | 16267755.00 | 16479720.00  |
> |  0.3 |  11208.60 |  14166.10 |  11078.14 |  3454540.00  | 3828730.00  |
> | 0.4  |  7242.20 | 7060.40 |  7328.60 |   3454540.00  |   3828730.00 |
> |  0.5 | 4885.60 |  5123.40 | 5332.00  | 1374857.30 | 1785804.80 |
> |  0.6 | 4408.80 | 4885.00  |  4885.00 | 548700.00  | 823350.00 |
>
>
>
> The experimental results show that, our  CoKernelFC achieves the best sample complexity when tasks are the same or similar (in which case cooperation among agents can bring benefits). This demonstrates the superiority of our globally optimal sampling strategy and the effectiveness of our cooperation scheme.
>
> Moreover, the experimental results reflect that, the more the tasks are similar, the higher learning speedup the agents can attain. Specifically, in the $rank(K_{\mathcal{Z}})=1$ case (Table 1), the sample complexity of CoKernelFC is about $\frac{1}{V}$ of that of the best single-agent adaption IndRAGE, which shows that CoKernelFC achieves a $V$-speedup.
> In the $rank(K_{\mathcal{Z}})\in(1,V)$ case (Table 2), the ratio of the sample complexity of CoKernelFC over that of IndRAGE is larger than $\frac{1}{V}$, which implies that CoKernelFC achieves a speedup lower than $V$.
> In the $rank(K_{\mathcal{Z}})=V$ case (Table 3), CoKernelFC performs similar to IndRAGE, since information sharing among agents brings no advantage in this case.
> These empirical results  match and validate our theoretical analysis.
>
> ---
>
> References:
> Tanner Fiez, Lalit Jain, Kevin G. Jamieson,  Lillian Ratliff. Sequential experimental design
> for transductive linear bandits. NeurIPS, 2019.
> Chao Tao, Saúl Blanco, Yuan Zhou. Best arm identification in linear bandits with linear dimension dependency. ICML, 2018.
> Yihan Du, Yuko Kuroki, Wei Chen. Combinatorial pure exploration with full-bandit or partial
> linear feedback. AAAI, 2021.

---

### Comment · Area_Chair_KfZG · 2022-11-18
**Additional references on distributed/federated linear bandits**

Thank you for addressing the reviewers' comments. For distributed/federated linear bandits, the authors also need to comment on the following two references. [1] studied synchronous federated linear bandits with full participation. [2] studied asynchronous federated linear bandits which supports partial participation.

[1] Wang et al.  Distributed Bandit Learning: Near-Optimal Regret with Efficient Communication, ICLR2020
[2] He et al. A Simple and Provably Efficient Algorithm for Asynchronous Federated Contextual Linear Bandits, NeurIPS 2022

---

> ### Author Response · Authors · 2022-11-18
> **Thank Area Chair! We have cited and discussed the suggested references in our revision**
>
> Thank you very much for providing these two important references! We have cited them and added discussion in Section 2 and Appendix B of our revision.
>
> [Wang et al., 2020] study distributed multi-armed and linear bandit problems, and [He et al., 2022] investigate federated linear bandits with asynchronous communication. These two works both consider the regret minimization objective, which is different from our pure exploration goal and does not involve our communication round-speedup trade-off analysis. In addition, their algorithms are mainly designed for the linear bandit setting, where agents directly communicate the estimated reward parameters. These linear bandit algorithms cannot be directly applied to solve our CoPE-KB problem, because under kernel representation, the reward parameter is high-dimensional and expensive to explicitly transmit.

---

### Decision · Program_Chairs · 2023-01-20

**Decision:**

Accept: poster

**Justification For Why Not Higher Score:**

This is mostly a theory paper, and its audience might be limited in ICLR.

**Justification For Why Not Lower Score:**

There is unanimous support to accept this paper after the author's response.

**Metareview: Summary, Strengths And Weaknesses:**

This paper studies collaborative pure exploration in kernel bandits and proposes two algorithms under fixed confidence and fixed budget respectively. It also provides theoretical analysis for the proposed algorithms.

Strengths:
Nearly matching upper and lower bounds in both sample complexity and communication complexity are established.

Weaknesses:
Lack of thorough experiments in the main paper (before revision).


**Note From Pc:**

if the above contains the word "oral" or "spotlight" please see: "oral" presentation means -> notable-top-5% and "spotlight" means -> notable-top-25%. As stated in our emails, we are disassociating presentation type from AC recommendations